# On the sample complexity of semi-supervised multi-objective learning

**Tobias Wegel**[1]     **Geelon So**[2]     **Junhyung Park**[1]     **Fanny Yang**[1]

[1]Department of Computer Science, ETH Zurich
[2]Department of Computer Science and Engineering, UC San Diego

## Abstract

In multi-objective learning (MOL), several possibly competing prediction tasks must be solved jointly by a single model. Achieving good trade-offs may require a model class $\mathcal{G}$ with larger capacity than what is necessary for solving the individual tasks. This, in turn, increases the statistical cost, as reflected in known MOL bounds that depend on the complexity of $\mathcal{G}$. We show that this cost is unavoidable for some losses, even in an idealized semi-supervised setting, where the learner has access to the Bayes-optimal solutions for the individual tasks as well as the marginal distributions over the covariates. On the other hand, for objectives defined with Bregman losses, we prove that the complexity of $\mathcal{G}$ may come into play only in terms of *unlabeled* data. Concretely, we establish sample complexity upper bounds, showing precisely when and how unlabeled data can significantly alleviate the need for labeled data. This is achieved by a simple pseudo-labeling algorithm.

## 1   Introduction

The *multi-objective learning* (MOL) paradigm has recently emerged to extend the classical problem of risk minimization from statistical learning to settings with multiple notions of risk [32, 19, 59, 27]. Multi-objective learning problems are ubiquitous in practice, as it often matters how our models behave with respect to multiple metrics and across different populations. For example, consider designing a policy for a self-driving car: the risks could measure different notions of safety (e.g., safety of passengers or pedestrians), or safety under various conditions (e.g., different locations).

More formally, we study the MOL setting with $K$ population risk functionals $\mathcal{R}_1, \ldots, \mathcal{R}_K$, each quantifying an average, possibly different, loss $\ell_k$ incurred by a prediction model $g$ over the data distribution $P^k$. The aim is to learn models from a class $\mathcal{G}$ that minimize all $K$ *excess risks* $\mathcal{E}_k(g) := \mathcal{R}_k(g) - \inf \mathcal{R}_k$ jointly, using only finite-sample access to the distributions. Here, $\inf \mathcal{R}_k$ is the Bayes risk of the $k$th task, which is the smallest achievable risk over all measurable functions. Specifically, we study the sample complexity of learning the set of Pareto optimal models in $\mathcal{G}$. Recall that a model is *Pareto optimal* in $\mathcal{G}$ if any alternative model in $\mathcal{G}$ that reduces one risk necessarily increases another (see Definition 1); we often simply say that such a model makes an optimal trade-off. Under mild conditions, the set of Pareto optimal models can be recovered by minimizing a family of *scalarized* objectives $\mathcal{T}_s$ that we call the *s-trade-offs*:[1]

$$\min_{g \in \mathcal{G}} \quad \underbrace{\mathcal{T}_s(g) := s\big(\mathcal{E}_1(g), \ldots, \mathcal{E}_K(g)\big)}_{\text{the } s\text{-trade-off achieved by } g}, \qquad s \in \mathcal{S} \tag{1}$$

where the map $s : \mathbb{R}^K \to \mathbb{R}$ is from some family $\mathcal{S}$ of *scalarization* functions that aggregates the excess risks into a single statistic. Notice that if the excess risks were known, the problem in Eq. (1) would reduce to a family of classical (multi-objective) optimization problems [44, 54, 38]. However, because the objectives in Eq. (1) depend on distributional quantities that are unknown, we need to

39th Conference on Neural Information Processing Systems (NeurIPS 2025).

learn the solutions from data. Specifically, we study the sample complexity of achieving Eq. (1) up to errors $\varepsilon_s > 0$, a problem we call $\mathcal{S}$-*multi-objective learning*, $\mathcal{S}$-MOL for short (see Definition 2).

Two main lines of work have studied the sample complexity of MOL, both predominantly in the *supervised* framework. In the multi-distribution learning (MDL) literature [27, 4, 51, 73], the goal of the learner is to recover a solution to Eq. (1) for one specific $s$-trade-off induced by the scalarization $s(\boldsymbol{v}) = \max_{k \in [K]} v_k$. This yields the familiar min-max formulation of MOL.[2] And in the literature for the general $\mathcal{S}$-MOL setting [19, 59], the learner aims to solve Eq. (1) for multiple scalarizations. Both lines establish sample complexity bounds in terms of capacity measures of $\mathcal{G}$, which can be shown to be tight in the worst case. However, solutions with good trade-offs may only be found in a complex model class $\mathcal{G}$ even when individual tasks are easy to solve in smaller classes $\mathcal{H}_k$. In such cases, previous worst-case results do not address whether it is really necessary to pay the full, supervised statistical cost of $\mathcal{S}$-MOL over $\mathcal{G}$. This motivates our consideration of *semi-supervised* multi-objective learning, in which the learner has access to both *labeled* and (cheaper) *unlabeled* data for each of the $K$ tasks. In the single-objective setting, it is well-known that access to unlabeled data can, at times, significantly reduce the amount of labeled data required [17, 70]. But for multi-objective learning, the sample complexity in a semi-supervised setting is largely unexplored, with only a few exceptions [5, 65] that rely on additional assumptions for unlabeled data to be helpful (see Appendix A for a discussion of related works). Thus, the question we aim to address in this paper is

> *Given that each task $k \in [K]$ is solvable in a hypothesis class $\mathcal{H}_k$, how much labeled and unlabeled data is needed to achieve trade-offs available in a larger function class $\mathcal{G}$?*

In this paper, we give a holistic characterization of the conditions when unlabeled data can help and by how much. In terms of sample complexity upper bounds, we show that for a large class of losses, the capacity of $\mathcal{G}$ comes into play only in the amount of unlabeled data required, while the amount of labeled data merely depends on that of $\mathcal{H}_1, \ldots, \mathcal{H}_K$. Moreover, we show that these rates are achieved by a simple, pseudo-labeling-based algorithm. Concretely, our contributions are as follows:

- We first show hardness of $\mathcal{S}$-MOL under *uninformative losses* via a minimax sample complexity lower bound that holds even when the learner knows the Bayes-optimal models for each task and has access to the marginal distributions over unlabeled data, i.e., infinite unlabeled data (Section 3.1).

- We then prove that risks induced by *Bregman divergence losses*—which include the square and cross-entropy losses—effectively disentangle the multi-objective learning problem. For Bregman losses, information about individual risk minimizers can significantly reduce labeled sample complexity in the semi-supervised setting via a simple *pseudo-labeling* algorithm (Section 3.2).

- Specifically, for $\mathcal{S}$-MOL with Bregman losses, we first provide a uniform bound over the excess $s$-trade-offs of the pseudo-labeling algorithm for bounded, Lipschitz losses via uniform convergence (Section 4.1). Our major technical contribution then lies in proving *localized rates* that are distribution-specific under stronger assumptions (Section 4.2). Crucially, the labeled sample complexity in both bounds only depends on the classes $\{\mathcal{H}_k\}_{k=1}^K$, while $\mathcal{G}$ only appears in the unlabeled sample complexity.

Our analysis reveals an interesting insight: even though the pseudo-labeling algorithm is reminiscent of single-objective semi-supervised learning procedures, the reason behind the benefits of unlabeled data turns out to be fundamentally different. In single-objective learning, unlabeled data can only help if, roughly speaking, the marginal carries information about the labels [13, 26, 76]. Our results, in contrast, hold without any such assumptions. In multi-objective learning, unlabeled data helps the learner determine the relative importance of each test instance to each task: if the likelihood of an input is higher under one task than another, a model can accordingly prioritize the more relevant risk to achieve better trade-offs. This may be completely independent of the labels assigned by each task.

---

[1]We scalarize the excess risks $\mathcal{E}_k$ because they each capture the *suboptimality* with respect to what is theoretically attainable when optimizing $\mathcal{R}_k$ without consideration for other risks. Notice however that the Pareto set of the excess risks $\mathcal{E}_k$ is the same as the Pareto set of the risks $\mathcal{R}_k$, as they are equal up to the constants $\inf \mathcal{R}_k$. We further motivate this in Section 2.2 and Fig. 1a.

[2]Contrary to Eq. (1), in the MDL literature, the risks are usually directly scalarized.

## 2 Semi-supervised multi-objective learning

In this section, we formally introduce the semi-supervised $\mathcal{S}$-multi-objective learning problem. For ease of reference, an overview of notation is provided in Table 2 of Appendix F.

### 2.1 Preliminaries and the individual tasks

Let $\mathcal{X}$ be the *feature space*, and $\mathcal{Y} \subseteq \mathbb{R}^q$ the *label space*. We are interested in $K$ prediction tasks, indexed by $k \in [K] := \{1, \ldots, K\}$, over joint distributions $P^k$ of $(X^k, Y^k)$ on the product space $\mathcal{X} \times \mathcal{Y}$. We denote the underlying joint probability measure by $\mathbb{P}$. From each task, we observe $n_k$ i.i.d. *labeled samples* $\{(X_i^k, Y_i^k)\}_{i=1}^{n_k}$ from $P^k$, and $N_k$ i.i.d. *unlabeled samples* $\{\widetilde{X}_i^k\}_{i=1}^{N_k}$ from the marginal of $P^k$ on $\mathcal{X}$, denoted $P_X^k$. Let $\mathcal{D}$ denote the combined dataset of both labeled and unlabeled data. For each task $k \in [K]$, we define the population and empirical risks of a function $f : \mathcal{X} \to \mathcal{Y}$ as

$$\mathcal{R}_k(f) := \mathbb{E}\left[\ell_k(Y^k, f(X^k))\right] \qquad \text{and} \qquad \widehat{\mathcal{R}}_k(f) := \frac{1}{n_k} \sum_{i=1}^{n_k} \ell_k(Y_i^k, f(X_i^k)), \tag{2}$$

where $\ell_k : \mathcal{Y} \times \mathcal{Y} \to \mathbb{R}$ is a (not necessarily symmetric) loss function with $\ell_k(y, \widehat{y})$ being the loss incurred by predicting $\widehat{y}$ when the true label is $y$. Further, we write $\mathcal{F}_{\text{all}}$ for the set of all functions $f : \mathcal{X} \to \mathcal{Y}$ for which all integrals in this paper are well-defined. For each $k \in [K]$, we assume access to a function class $\mathcal{H}_k \subseteq \mathcal{F}_{\text{all}}$ that contains a population risk minimizer of $\mathcal{R}_k$, that is,

$$\exists f^\star \in \mathcal{F}_{\text{all}} \qquad \text{such that} \qquad f_k^\star \in \arg\min_{f \in \mathcal{F}_{\text{all}}} \mathcal{R}_k(f) \qquad \text{and} \qquad f_k^\star \in \mathcal{H}_k. \tag{3}$$

The risk that any model $f : \mathcal{X} \to \mathcal{Y}$ incurs is at least $\mathcal{R}_k(f_k^\star)$, so we focus our attention on achieving small *excess risk* with respect to the Bayes optimal predictor, defined as $\mathcal{E}_k(f) := \mathcal{R}_k(f) - \mathcal{R}_k(f_k^\star)$.

### 2.2 Pareto optimality and scalarization

In multi-objective learning, our aim is to learn models $g$ from some function class $\mathcal{G}$ that, ideally, achieve low excess risk on all tasks *simultaneously*. Since, by assumption, the individual tasks are optimally solved in $\mathcal{H}_k$, we only consider hypothesis classes $\mathcal{G} \subset \mathcal{F}_{\text{all}}$ that satisfy $\mathcal{G} \supseteq \bigcup_{k \in [K]} \mathcal{H}_k$. But even if $\mathcal{G}$ is very large, minimizing all excess risks may not be possible. In particular, in this work we do *not* assume that there exists one $f : \mathcal{X} \to \mathcal{Y}$ that performs well across objectives (as opposed to the collaborative learning framework [14] or the setting in [5]). Instead, the aim is to recover the set of *Pareto optimal* solutions in the class $\mathcal{G}$ for the $K$ objectives, formally defined as follows.

**Definition 1** (Pareto optimality). Let $\mathcal{E}_1, \ldots, \mathcal{E}_K$ be a collection of excess risk functionals. We say that a function $g \in \mathcal{G}$ is *Pareto optimal* in $\mathcal{G}$ if there is no other $g' \in \mathcal{G}$ such that

$$\exists k \in [K] \text{ s.t. } \mathcal{E}_k(g') < \mathcal{E}_k(g) \qquad \text{and} \qquad \forall j \in [K], \mathcal{E}_j(g') \leq \mathcal{E}_j(g).$$

The subset of $\mathcal{G}$ containing all Pareto optimal functions is called the *Pareto set*. The subset of $\mathbb{R}^K$ containing the excess risk vectors of the Pareto set is called the *Pareto front*, defined as

$$\mathfrak{F}(\mathcal{G}) := \left\{ \left( \mathcal{E}_1(g), \ldots, \mathcal{E}_K(g) \right) : g \text{ is in the Pareto set of } \mathcal{G} \right\} \subseteq \mathbb{R}^K.$$

In words, any model in $\mathcal{G}$ that reduces one risk over a Pareto optimal model must increase another risk. Every Pareto-optimal model corresponds to a distinct "preference" or "trade-off", all of which are equally valid from a decision-theoretic perspective [29]. We can quantify such trade-offs using scalarization functions $s : \mathbb{R}^K \to \mathbb{R}$ that, for all $f \in \mathcal{F}_{\text{all}}$, map the excess risks into a scalar objective

$$\mathcal{T}_s(f) = s\left( \mathcal{E}_1(f), \ldots, \mathcal{E}_K(f) \right) \qquad \text{with} \qquad g_s \in \arg\min_{g \in \mathcal{G}} \mathcal{T}_s(g), \tag{4}$$

see also Eq. (1). We call $\mathcal{T}_s(f)$ the *s-trade-off* achieved by $f$. It has a natural interpretation: recall that $\mathcal{E}_k(g_s)$ is the cost incurred by the $k$-th task to make this particular type of trade-off over myopically optimizing $\mathcal{E}_k$. Then, $\mathcal{T}_s(g_s)$ aggregates these costs (see Fig. 1). This interpretation also further motivates scalarizing the *excess* risks instead of the risks: if one task were to have much higher Bayes risk than another, scalarizing the risks would not aggregate the *additional* cost, cf. Fig. 1a and [1].

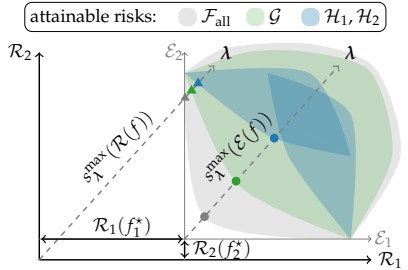 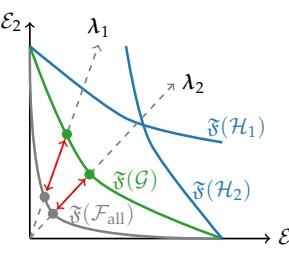 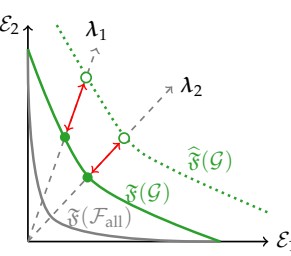

(a) The attainable risks in each function class and one Tchebycheff scalarization of the risks and of the excess risks.

(b) The larger $\mathcal{G}$, the smaller the minimal $s$-trade-off within $\mathcal{G}$ due to low "bias".

(c) The larger $\mathcal{G}$, the larger the *excess* $s$-trade-off on the function class $\mathcal{G}$ due to high "variance".

Figure 1: (a) All attainable risks for the function classes $\mathcal{F}_{\text{all}}, \mathcal{G}, \mathcal{H}_1, \mathcal{H}_2$. Using scalarizations on the risks directly can be misleading if the Bayes risk of one task is much larger than that of another: even if both tasks have equal weight, the Tchebycheff scalarization (5) may inadvertently only solve one task (triangles); cf. [1]. Scalarizing the excess risks avoids this (dots). (b) The Pareto fronts for the classes $\mathcal{H}_1, \mathcal{H}_2, \mathcal{G}, \mathcal{F}_{\text{all}}$ in the space of *excess* risks, and two Tchebycheff scalarizations $s_{\boldsymbol{\lambda}}^{\max}$ (5) with different $\boldsymbol{\lambda}$ (gray dashed line), with the corresponding trade-off minimizers (dots), and the gap to the minimizers in $\mathcal{F}_{\text{all}}$ (bi-directed red arrows). (c) The population and empirical Pareto fronts $\mathfrak{F}(\mathcal{G})$ and $\widehat{\mathfrak{F}}(\mathcal{G})$, and the excess $s$-trade-off of the estimated Pareto points (bi-directed red arrows) on the same two Tchebycheff scalarizations.

Two popular examples of scalarization families are *Tchebycheff* and *linear* scalarizations, defined as

$$\mathcal{S}_{\max} = \left\{ s_{\boldsymbol{\lambda}}^{\max}(\boldsymbol{v}) = \max_{k \in [K]} \lambda_k v_k \mid \boldsymbol{\lambda} \in \Delta^{K-1} \right\}, \quad \mathcal{S}_{\text{lin}} = \left\{ s_{\boldsymbol{\lambda}}^{\text{lin}}(\boldsymbol{v}) = \sum_{k \in [K]} \lambda_k v_k \mid \boldsymbol{\lambda} \in \Delta^{K-1} \right\},$$

(5)

where $\Delta^{K-1}$ is the $(K-1)$-probability simplex. They represent the worst-case and averaged notions of excess risks, respectively (see Fig. 1a for a visualization of the Tchebycheff scalarization). Minimizing these families of scalarizations recovers the Pareto set under some conditions (e.g., convexity for linear scalarization), see the detailed discussions in [47, 23, 44]. But of course, other scalarizations also exist [23]. Our most general result (Section 4.1) holds for monotonic scalarizations that satisfy the reverse triangle inequality and positive homogeneity, defined as

$$\begin{aligned}
\forall \boldsymbol{v}, \boldsymbol{w} \in [0, \infty)^K : && (\forall k \in [K], \, v_k \leq w_k) &\implies s(\boldsymbol{v}) \leq s(\boldsymbol{w}), \\
\forall \boldsymbol{v}, \boldsymbol{w} \in \mathbb{R}^K : && |s(\boldsymbol{v}) - s(\boldsymbol{w})| &\leq s(|v_1 - w_1|, \ldots, |v_K - w_K|), \\
\forall \boldsymbol{v} \in \mathbb{R}^K, \alpha > 0 : && s(\alpha \boldsymbol{v}) &= \alpha \cdot s(\boldsymbol{v}).
\end{aligned}$$

(6)

Both the linear and Tchebycheff scalarizations from Eq. (5) satisfy the properties in Eq. (6).

## 2.3 Multi-objective learning

Because $\inf_{g \in \mathcal{G}} \mathcal{T}_s(g)$ may be arbitrarily large, we evaluate our empirical estimates of $g_s$ using the *excess $s$-trade-off*, defined, for $f \in \mathcal{F}_{\text{all}}$, as $\mathcal{T}_s(f) - \inf_{g \in \mathcal{G}} \mathcal{T}_s(g)$. The $\mathcal{S}$-MOL problem is then to achieve small excess $s$-trade-off across scalarizations with high probability.

**Definition 2** ($\mathcal{S}$-MOL). Let $\mathcal{S}$ be a family of scalarizations $s : \mathbb{R}^K \to \mathbb{R}$, $(\varepsilon_s)_{s \in \mathcal{S}}$ a family of positive real numbers, and $\delta \in (0, 1)$. Let $\mathcal{A}$ be an algorithm that, provided with a dataset $\mathcal{D}$ and the function classes $\{\mathcal{H}_k\}_{k=1}^K$ and $\mathcal{G}$, returns a family of functions $\{\widehat{g}_s : s \in \mathcal{S}\} \subset \mathcal{G}$. Then $\mathcal{A}$ solves the *$\mathcal{S}$-multi-objective learning ($\mathcal{S}$-MOL)* problem with parameters $((\varepsilon_s)_{s \in \mathcal{S}}, \delta)$, if

$$\mathbb{P}\left( \forall s \in \mathcal{S} : \mathcal{T}_s(\widehat{g}_s) - \inf_{g \in \mathcal{G}} \mathcal{T}_s(g) \leq \varepsilon_s \right) \geq 1 - \delta, \tag{$\mathcal{S}$-MOL}$$

where the probability is taken with respect to draws of the training dataset $\mathcal{D}$.

From the population-level optimization perspective in Eq. (1), better trade-offs become possible as the class $\mathcal{G}$ grows. This is visualized in Fig. 1b, showing the Pareto fronts achieved by the function classes $\mathcal{F}_{\text{all}}, \mathcal{G}, \mathcal{H}_1, \mathcal{H}_2$ in a two-objective setting. The separation between the Pareto front $\mathfrak{F}(\mathcal{G})$ and the theoretical optimum $\mathfrak{F}(\mathcal{F}_{\text{all}})$ can be seen as the "multi-objective bias" incurred in $\mathcal{S}$-MOL

due to a conservative choice of $\mathcal{G}$. For two Tchebycheff scalarizations, the red bi-directed arrows in Fig. 1b reflect this point-wise "bias". However, because the Pareto front needs to be learned from finite samples, we would also expect from classical learning theory that as $\mathcal{G}$ grows, so does the "multi-objective variance" of an empirical Pareto front $\widehat{\mathfrak{F}}(\mathcal{G})$. Fig. 1c illustrates this by the gap between $\mathfrak{F}(\mathcal{G})$ and $\widehat{\mathfrak{F}}(\mathcal{G})$, and for the same two Tchebycheff scalarizations, the red bi-directed arrows reflect the excess $s$-trade-off. In the next section we first address how much excess trade-off any algorithm necessarily incurs when learning $\mathfrak{F}(\mathcal{G})$ from data.

## 3 Motivating Bregman losses: A hardness result

To answer this in the context of a semi-supervised setting, we now argue that for the unlabeled data to help solve $\mathcal{S}$-MOL, the structure of the loss functions is key.

### 3.1 A sample complexity lower bound for ideal semi-supervised $\mathcal{S}$-MOL

Let us consider the class of PAC-learners for $\mathcal{S}$-MOL, which are learners that achieve $\mathcal{S}$-MOL for all distributions over $\mathcal{X} \times \mathcal{Y}$. For concreteness, consider multi-objective binary classification with zero-one loss, where $\mathcal{S}$ is the entire family of linear scalarizations $\mathcal{S}_{\text{lin}}$:

**Definition 3** (Binary classification). *Let $\mathcal{G}$ be a hypothesis class with VC dimension $d_{\mathcal{G}} \in \mathbb{N}$ on a data domain $\mathcal{X} \times \mathcal{Y}$ where $\mathcal{Y} = \{0, 1\}$. For each task $k \in [K]$, define $\ell_k(y, \widehat{y}) = \mathbf{1}\{y \neq \widehat{y}\}$.*

For supervised $\mathcal{S}$-MOL with $\varepsilon_s \equiv \varepsilon$ for all $s \in \mathcal{S}_{\text{lin}}$, prior works achieve a sample complexity upper bound of $O(Kd_{\mathcal{G}}/\varepsilon^2)$, up to logarithmic terms, see [19, 59] and Corollary A.1. In fact, a matching lower bound of $\Omega(Kd_{\mathcal{G}}/\varepsilon^2)$ holds as well: after all, the set of $s$-trade-offs $\{\mathcal{T}_s : s \in \mathcal{S}_{\text{lin}}\}$ contains the individual excess risk functionals $\mathcal{E}_k$, and hence solving $\mathcal{S}$-MOL requires the learner to solve the $K$ original tasks as well. The lower bound then follows from standard agnostic PAC-learning [57, Theorem 6.8]. In short, previous upper bounds are tight and the sample complexity of supervised $\mathcal{S}$-MOL is $\Theta(Kd_{\mathcal{G}}/\varepsilon^2)$, which also coincides with the sample complexity of MDL under non-adaptive sampling [73]. In the semi-supervised $\mathcal{S}$-MOL setting, the question now becomes: can the unlabeled data reduce the label complexity of this problem? Perhaps surprisingly, we now show that the same lower bound holds, even if the learner has additional access to Bayes optimal classifiers $f_k^{\star}$ and the marginal distributions $P_X^k$.

**Proposition 1** (Hardness of semi-supervised multi-objective binary classification). *Fix any $K > 1$ and any $\varepsilon \in (0, 1/12)$. For a given tuple $(P^1, \ldots, P^K)$, denote by $S^k$ a labeled dataset consisting of i.i.d. samples from $P^k$, let $f_k^{\star}$ be a Bayes optimal classifier of $P^k$, and let $P_X^k$ be the marginal distribution on $\mathcal{X}$. Denote by $\mathcal{A}$ any algorithm that, given $\{S^k, f_k^{\star}, P_X^k\}_{k=1}^K$, returns a set of classifiers $\{\widehat{g}_s \in \mathcal{G} : s \in \mathcal{S}_{\text{lin}}\}$. If $\mathcal{A}$ achieves ($\mathcal{S}$-MOL) with $\varepsilon_s \equiv \varepsilon$ for all linear scalarizations $s \in \mathcal{S}_{\text{lin}}$, $\delta \leq 1/6$, and for all distributions $(P^1, \ldots, P^K)$ in the multi-objective binary classification setting (Definition 3), then the total number of labeled samples it requires is at least $|S^1| + \cdots + |S^K| \geq CKd_{\mathcal{G}}/\varepsilon^2$ where $C > 0$ is a universal constant.*

See Appendix D.1 for the proof. Proposition 1 shows that the label sample complexity lower bounds for supervised $\mathcal{S}$-MOL cannot be improved for the problem in Definition 3—even in an idealized semi-supervised $\mathcal{S}$-MOL setting where the learner has infinite unlabeled data and can perfectly solve the individual learning tasks. This effect is due to the zero-one loss not being a *proper scoring rule*, which is necessary for weighing the risks of two different tasks against each other. And indeed, other losses, such as the hinge or absolute deviation loss, suffer from the same problem. See also [62] for a discussion of *calibration* in multi-objective learning. In our main results, we show that this lower bound can be circumvented in learning settings where the loss functions are proper in this sense.

### 3.2 Bregman divergence losses and a pseudo-labeling algorithm

In this section, we introduce *Bregman losses* and their key property that allows us to leverage unlabeled data and alleviate labeled sample complexity via a pseudo-labeling algorithm.

**Definition 4** (Bregman loss). *Let $\mathcal{Y}$ be convex. A loss $\ell : \mathcal{Y} \times \mathcal{Y} \to [0, \infty]$ is called a *Bregman loss* if there is a strictly convex and differentiable potential $\phi : \mathcal{Y} \to \mathbb{R}$ such that $\ell(y, \widehat{y}) = \phi(y) - \phi(\widehat{y}) - \langle \nabla \phi(\widehat{y}), y - \widehat{y} \rangle$.*

Many standard prediction losses are Bregman losses. For example, the squared loss can be obtained by setting $\phi(y) = \|y\|_2^2$, the logistic loss by choosing $\phi(y) = y \log y + (1 - y) \log(1 - y)$, and the Kullback-Leibler divergence using $\phi(y) = \sum_{k=1}^q y_j \log y_j$. As we now show, an important fact that we will leverage about learning with a Bregman loss is that the associated excess risk functional can be expressed in terms of its minimizer. To state it precisely, we introduce the notions of *population and empirical risk discrepancies* of a function $f \in \mathcal{F}_{\text{all}}$ from some $h \in \mathcal{H}_k$, defined as:

$$d_k(f;h) := \mathbb{E}\left[\ell_k(h(X^k), f(X^k))\right], \qquad \widehat{d}_k(f;h) := \frac{1}{N_k} \sum_{i=1}^{N_k} \ell_k(h(\widetilde{X}_i^k), f(\widetilde{X}_i^k)). \tag{7}$$

We further define for some $h_1 \in \mathcal{H}_1, \dots, h_K \in \mathcal{H}_K$ and $\boldsymbol{h} = (h_1, \dots, h_K)$ the population and empirical *scalarized* risk discrepancies of a function $f \in \mathcal{F}_{\text{all}}$ from $\boldsymbol{h}$ as

$$d_s(f;\boldsymbol{h}) = s(d_1(f;h_1), \dots, d_K(f;h_K)), \qquad \widehat{d}_s(f;\boldsymbol{h}) = s(\widehat{d}_1(f;h_1), ..., \widehat{d}_K(f;h_K)). \tag{8}$$

We are now ready to state Lemma 1, proved in Appendix D.2.

**Lemma 1** (Properties of Bregman losses, based on [7]). *For each $k \in [K]$, let $\ell_k$ be a Bregman loss with potential $\phi_k$. If both $\mathbb{E}[Y^k]$ and $\mathbb{E}[\phi_k(Y^k)]$ are finite, then up to almost sure equivalence,*

$$f_k^\star(\cdot) := \underset{f \in \mathcal{F}_{\text{all}}}{\arg\min} \, \mathcal{R}_k(f) = \mathbb{E}\left[Y^k | X^k = \cdot\right] \quad \text{and} \quad \forall f \in \mathcal{F}_{\text{all}}, \; \mathcal{E}_k(f) = d_k(f; f_k^\star).$$

Along with Eqs. (4) and (8), Lemma 1 implies $\mathcal{T}_s(f) = d_s(f; \boldsymbol{f}^\star)$ for $\boldsymbol{f}^\star = (f_1^\star, \dots, f_K^\star)$. Note that the second part of Lemma 1 decomposes the risk into a task-specific intrinsic noise and a discrepancy term; $\mathcal{R}_k(f) = \mathcal{R}_k(f_k^\star) + d_k(f; f_k^\star)$. It turns out that Bregman divergences are, up to transformation of the label space, the only loss functions that enjoy such a decomposition (see [28] and Appendix B). This decomposition helps justify the following *pseudo-labeling multi-objective learning* algorithm (Algorithm 1).

---
**Algorithm 1** Pseudo-labeling (PL-MOL)
---
1: **for** $k \in [K]$ **do**
2:     Compute $\widehat{h}_k = \arg\min_{h \in \mathcal{H}_k} \widehat{\mathcal{R}}_k(h)$
3: **end for**
4: **for** $s \in \mathcal{S}$ **do**
5:     Compute $\widehat{g}_s = \arg\min_{g \in \mathcal{G}} \widehat{d}_s(g; \widehat{\boldsymbol{h}})$
6: **end for**
7: Return $\{\widehat{g}_s : s \in \mathcal{S}\}$.
---

First, we minimize the individual empirical risks $\widehat{\mathcal{R}}_k$ over $\mathcal{H}_k$ to obtain $\widehat{\boldsymbol{h}} = (\widehat{h}_1, \dots, \widehat{h}_K)$, the set of empirical risk minimizers; thus, we estimate the task-wise Bayes-optimal models $f_k^\star$ (Line 2). Given Lemma 1, we can then approximate the excess risks $\mathcal{E}_k$ via the empirical risk discrepancies $\widehat{d}_k(\cdot; \widehat{h}_k)$ using unlabeled data. And so, the $s$-trade-off $\mathcal{T}_s(\cdot)$ can accordingly be approximated by $\widehat{d}_s(\cdot; \widehat{\boldsymbol{h}})$. The empirical estimate of the Pareto set in $\mathcal{G}$ is then given as the minimizer of $\widehat{d}_s(\cdot; \widehat{\boldsymbol{h}})$ in $\mathcal{G}$ (Line 5). Note that reusing the covariates of the labeled data in this second step would yield at most a constant gain.

Notice that the second step (Line 5) is equivalent to first pseudo-labeling all unlabeled data using the ERMs, and then passing it to the supervised $\mathcal{S}$-MOL algorithm from [59] ("ERM-MOL", Algorithm 2 discussed in Appendix A.1). Finally, note that from a computational perspective, even if $\mathcal{S}$ is not finite, Algorithm 1 can be implemented, e.g., using hypernetworks, see Appendix A.2.

### 3.3 Characterization of models with optimal trade-offs: A variational inequality

The pseudo-labeling method illustrates how one can estimate the $s$-trade-off solutions $g_s \in \mathcal{G}$ from both labeled and unlabeled data when the losses are Bregman divergences. We now show that Bregman losses also enable characterizing the minimizers $g_s$ via a variational inequality in some cases. From this inequality, in turn, we can derive conditions for $g_s$ to have a particularly simple representation which sheds some light on why unlabeled data can help. Specifically, under linear scalarizations and convexity, we can show the following result.

**Lemma 2** (Variational characterization of minimizers). *For each $k \in [K]$, let $\ell_k$ be a Bregman loss with potential $\phi_k$. Suppose $s = s_{\boldsymbol{\lambda}}^{\text{lin}}$ is linear with weights $\boldsymbol{\lambda}$, and $\mu_s := \sum_{k=1}^K \lambda_k P_X^k$. Denote by $\langle \cdot, \cdot \rangle_s$ the inner product defined as $\langle f, f' \rangle_s = \int \langle f(x), f'(x) \rangle \, d\mu_s(x)$. Then for every non-empty, convex and closed set $\mathcal{G} \subseteq \mathcal{F}_{\text{all}}$ and convex $g \mapsto \mathcal{T}_s(g)$,*

$$\mathcal{T}_s(g) = \inf_{g' \in \mathcal{G}} \mathcal{T}_s(g') \quad \text{if and only if} \quad \forall g' \in \mathcal{G}: \quad \left\langle \sum_{k=1}^K \lambda_k \frac{dP_X^k}{d\mu_s} \nabla^2 \phi_k(g)(g - f_k^\star), g' - g \right\rangle_s \geq 0$$

*where $\nabla^2 \phi_k(g)$ denotes the function $x \mapsto \nabla^2 \phi_k(g(x))$. If $\mathcal{G}$ is bounded, such a $g \in \mathcal{G}$ exists.*

Lemma 2 is a direct consequence of Lemma D.6 and Theorem 46 in [69]. From this lemma, we can derive the $s$-trade-off solutions $g_s$ analytically in the special case where $\mathcal{G} = \mathcal{F}_{\text{all}}$ and all potentials are shared $\phi_k = \phi$. In that case, since the set of feasible models is unconstrained, the variational inequality in Lemma 2 holds with equality. In particular, the first argument of $\langle \cdot, \cdot \rangle_s$ must vanish, up to $\mu_s$-equivalence. Thus, we can deduce that the $s$-trade-off solution is $\mu_s$-a.s. of the form

$$g_s(x) = \sum_{k \in [K]} w_k(x) f_k^\star(x), \qquad \text{where} \quad w_k(x) = \lambda_k \frac{dP_X^k}{d\mu_s}(x) \qquad (9)$$

so that $x \mapsto w_k(x)$ is non-negative and $\sum_{k \in [K]} w_k(x) \equiv 1$. In short, the optimal prediction with respect to the $s$-trade-off on the instance $x \in \mathcal{X}$ is a convex combination of the individual Bayes optimal labels, cf. [40]. Additionally, if the marginals are shared $P_X^k = P_X$, then each $dP_X^k/d\mu_s = 1$, so the weights $w_k$ are independent of $x$. However, these are specific settings, and $g_s$ *does not generally need to take this form*. We will later make use of this specific form in Section 4.2.

# 4 Sample complexity upper bounds for pseudo-labeling

We now present uniform and localized upper bounds for Algorithm 1 for Bregman losses in terms of *Rademacher complexities*. Specifically, we use the coordinate-wise Rademacher complexity of a $\mathcal{Y}$-valued function class $\mathcal{H} \subseteq \mathcal{F}_{\text{all}}$ under distribution $P_X^k$ with $n$ samples, which is defined as

$$\mathfrak{R}_n^k(\mathcal{H}) := \mathop{\mathbb{E}}_{\substack{X_1^k, \ldots, X_n^k \sim P_X^k \\ \sigma_{11}, \sigma_{12}, \ldots, \sigma_{nq} \sim \text{Rad}}} \left[ \sup_{h \in \mathcal{H}} \left| \frac{1}{n} \sum_{i=1}^{n} \sum_{j=1}^{q} \sigma_{ij} h_j(X_i^k) \right| \right].$$

We discuss the choice and properties of this Rademacher complexity in Appendix E.2.

## 4.1 A uniform learning bound

We start with some assumptions on the loss functions $\ell_k$ that we require for our bounds.

**Assumption 1** (Regularity of the losses). For each $k \in [K]$, let $\ell_k$ be a Bregman loss satisfying:[3]

- Its associated potential function $\phi_k$ is $\mu_k$-strongly convex in $\mathcal{Y}$ with respect to $\ell_2$-norm, so that for all $y, y' \in \mathcal{Y}$, it holds that $\ell_k(y, y') = \phi_k(y) - \phi_k(y') - \langle \nabla \phi_k(y'), y - y' \rangle \geq \frac{\mu_k}{2} \|y - y'\|_2^2$.

- The loss is $L_k$-Lipschitz continuous in both arguments with $\ell_2$-norm in $\mathbb{R}^q$, that is, for all $y, y', y'' \in \mathcal{Y}$ it holds that $|\ell(y, y') - \ell(y, y'')| \leq L_k \|y' - y''\|_2$ and $|\ell(y', y) - \ell(y'', y)| \leq L_k \|y' - y''\|_2$.

- The loss is bounded by some constant $B_k < \infty$ as $\ell_k \leq B_k$.

The boundedness enables the concentration bounds used in our results and is a common assumption, and the strong convexity and Lipschitz continuity enable using a vector contraction inequality from [41], as well as establishing a uniform approximation of the excess risks. Most Bregman losses satisfy Assumption 1 on bounded domains $\mathcal{Y}$, while some (like the logistic loss) require careful treatment of Lipschitz continuity if the gradient is unbounded at the boundary of $\mathcal{Y}$ (cf. Corollary C.1 and Lemma E.1). We now state our first main result.

**Theorem 1.** *Suppose that Assumption 1 holds. Let $\mathcal{S}$ be any class of monotone scalarizations that satisfy the reverse triangle inequality and positive homogeneity in Eq. (6), and let $\{\widehat{g}_s : s \in \mathcal{S}\}$ be the class of solutions returned by Algorithm 1. Then ($\mathcal{S}$-MOL) holds for any $\delta \in (0, 1)$ and $\varepsilon_s = s(\varepsilon_1, \ldots, \varepsilon_K)$, where each $\varepsilon_k$ is bounded by*

$$\varepsilon_k \leq C_k \left( \mathfrak{R}_{N_k}^k(\mathcal{G}) + \left( \frac{\log(K/\delta)}{N_k} \right)^{1/2} + \sqrt{\mathfrak{R}_{n_k}^k(\mathcal{H}_k) + \left( \frac{\log(K/\delta)}{n_k} \right)^{1/2}} \right), \qquad (10)$$

*with $C_k = \max\{4L_k, \sqrt{2}B_k, L_k\sqrt{24L_k/\mu_k}, L_k\sqrt{6B_k/\mu_k}\}$.*

---

[3]The norm of the strong convexity and Lipschitz continuity in the first argument can be replaced by an arbitrary norm in Theorem 1. Replacing the norm of the Lipschitz continuity in the second argument in Theorem 1 entails using other vector Rademacher complexities, cf. Appendix E.2. They cannot be replaced in Theorem 2. Moreover, for our results it is sufficient for the Lipschitz property to hold on the range of $\mathcal{G}$.

Theorem 1 is proved in Appendix D.3. Using VC bounds on the Rademacher complexity (see Lemma E.6), Theorem 1 implies that for VC (subgraph) classes $\mathcal{G}$ and $\mathcal{H}_k = \mathcal{H}$ with VC dimensions $d_{\mathcal{G}}, d_{\mathcal{H}}$, only $\widetilde{O}(Kd_{\mathcal{H}}/\varepsilon^4)$ labeled and $\widetilde{O}(Kd_{\mathcal{G}}/\varepsilon^2)$ unlabeled samples are necessary to achieve $\varepsilon$-excess $s$-trade-off uniformly for all scalarizations. Comparing this with the sample complexity $\Theta(Kd_{\mathcal{G}}/\varepsilon^2)$ from Proposition 1, it is apparent that for Bregman losses, Algorithm 1 can alleviate the label complexity of $\mathcal{S}$-MOL significantly when $d_{\mathcal{G}} \gg d_{\mathcal{H}}$, and completely eradicates its dependence on $\mathcal{G}$. It shows that a large complexity of $\mathcal{G}$ can be compensated by a large amount of unlabeled data $N_k$, as long as $\mathfrak{R}_{N_k}^k(\mathcal{G}) \to 0$ for $N_k \to \infty$.

Also notice that, under Assumption 1, the map $g \mapsto \mathcal{E}_k(g)$ or its domain $\mathcal{G}$ can be non-convex, in which case non-linear scalarizations are necessary to reach the entire Pareto front. Theorem 1 applies to many such scalarizations, and in particular, the Tchebycheff scalarizations from Eq. (5).

## 4.2 A localized learning bound

The analysis in Theorem 1 is crude: it estimates the excess risks on all of $\mathcal{H}_k$ and $\mathcal{G}$, which is why the *global* Rademacher complexities appear in the bound and the unusual extra square-root appears. Such an analysis can be overly conservative, and a *localized* bound can provide much tighter statistical guarantees [9, 36]. To facilitate a localized analysis for Algorithm 1, we require some additional assumptions. First of all, we only consider linear scalarizations, that is, $\mathcal{S} \subseteq \mathcal{S}_{\mathrm{lin}}$ from Eq. (5), mostly for the following norms to be *Hilbert* norms: for all $k \in [K]$, $s \in \mathcal{S}_{\mathrm{lin}}$, and $f \in \mathcal{F}_{\mathrm{all}}$, define

$$\|f\|_k^2 := \mathbb{E}\|f(X^k)\|_2^2 \qquad \text{and} \qquad \|f\|_s^2 := s\left(\|f\|_1^2, \ldots, \|f\|_K^2\right).$$

We also require the following shape, strong convexity and smoothness assumptions.

**Assumption 2** (Shape, strong convexity and smoothness). Recall that $f_k^\star \in \arg\min_{f \in \mathcal{F}_{\mathrm{all}}} \mathcal{R}_k(f)$.

- For all $k \in [K]$, the function classes $\mathcal{H}_k - f_k^\star$ are star-shaped around the origin; for all $\alpha \in [0, 1]$, if $h \in \mathcal{H}_k - f_k^\star$, then $\alpha h \in \mathcal{H}_k - f_k^\star$. Moreover, the function class $\mathcal{G}$ is convex and closed.

- For $\gamma > 0$ and all $s \in \mathcal{S}$, $\boldsymbol{h} \in \mathcal{H}_1 \times \cdots \times \mathcal{H}_K$, the map $g \mapsto d_s(g; \boldsymbol{h}) - \gamma \|g\|_s^2$ is convex on $\mathcal{G}$.

- For some $\nu \in (0, \infty)$, the second and third derivatives of the potentials $\phi_k$ are bounded on $\mathcal{Y}$ as $\sup_{y \in \mathcal{Y}} \|\nabla^2 \phi_k(y)\|_2 \leq \nu$ and $\sup_{y \in \mathcal{Y}} \|\nabla^3 \phi_k(y)\|_2 \leq \nu$ in the $\ell_2$-operator norms.

The shape constraints are commonly used in local Rademacher complexity proofs [9], and the strong convexity acts as a "multi-objective Bernstein condition" [9, 35]. Moreover, the convexity of $\mathcal{G}$, strong convexity, and smoothness also enable a variational argument that is integral to the bound based on Lemma 2. To operationalize the smoothness, we assume a well-specified setting:

**Assumption 3.** The minimizer of $f \mapsto d_s(f; \boldsymbol{f}^\star)$ over $\mathcal{F}_{\mathrm{all}}$ is contained in $\mathcal{G}$ for all $s \in \mathcal{S}$.

Finally, for a refined version of our result, we also require the following norm-equivalence assumption that allows relating errors in $\|\cdot\|_s$-norm to errors in $\|\cdot\|_k$-norm.

**Assumption 4** (Norm equivalence). All covariate distributions $P_X^k$ are absolutely continuous with respect to the mixture distributions $\sum_{k=1}^K \lambda_k P_X^k$ for all $s_{\boldsymbol{\lambda}}^{\mathrm{lin}} \in \mathcal{S}$, and there is a constant $\eta$ so that

$$\forall k \in [K], s_{\boldsymbol{\lambda}}^{\mathrm{lin}} \in \mathcal{S}: \qquad \text{ess sup} \frac{dP_X^k}{d\left(\sum_{k=1}^K \lambda_k P_X^k\right)} \leq \eta^2 < \infty.$$

Specifically, as proved in Lemma D.7, Assumption 4 is equivalent to imposing $\|\cdot\|_k \leq \eta \|\cdot\|_s$ for all $k \in [K]$ and $s \in \mathcal{S}$. Sufficient conditions for Assumption 4 are that all weights of the scalarizations in $\mathcal{S}$ are bounded away from zero, or $P_X^k \ll P_X^j$ and $\text{ess sup} \, dP_X^k/dP_X^j \leq \eta^2$ for all $k, j \in [K]$.

We are now ready to state the localized bound. Recall that $f_k^\star$ is the Bayes model for the $k$th task, cf. Eq. (3), and for any $\boldsymbol{h} = (h_1, \ldots, h_K) \in \mathcal{H}_1 \times \cdots \times \mathcal{H}_K$, define $g_s^{\boldsymbol{h}} := \arg\min_{g \in \mathcal{G}} d_s(g; \boldsymbol{h})$, so that $g_s = g_s^{\boldsymbol{f}^\star}$, cf. Eq. (4). The result depends on the Rademacher complexities of the following sets of functions, defined using the balls $\mathcal{B}_{\|\cdot\|_k} = \{f \in \mathcal{F}_{\mathrm{all}} : \|f\|_k \leq 1\}$ as

$$\mathcal{H}_k(r) := (\mathcal{H}_k - f_k^\star) \cap r\mathcal{B}_{\|\cdot\|_k} \qquad \text{and} \qquad \mathcal{G}_k(r; \boldsymbol{h}) := \bigcup_{s \in \mathcal{S}} (\mathcal{G} - g_s^{\boldsymbol{h}}) \cap r\mathcal{B}_{\|\cdot\|_k}. \tag{11}$$

The excess $s$-trade-off is bounded in terms of the following *critical radii*, defined for each $k \in [K]$ as

$$\mathfrak{l}_k = \inf\left\{r \geq 0 : r^2 \geq \mathfrak{R}_{n_k}^k(\mathcal{H}_k(r))\right\} \quad \text{and} \quad \mathfrak{u}_k = \inf\left\{r \geq 0 : r^2 \geq \mathfrak{R}_{N_k}^k(\mathcal{G}_k(r; \boldsymbol{f}^\star))\right\}. \tag{12}$$

Critical radii like these are the key quantities of localized generalization bounds [9, 36]. They can be bounded using VC dimension (Lemma E.6) or with (generic) chaining [22, 61]. We define the *worst-case* critical radius in $\mathcal{G}$ by replacing $\boldsymbol{f}^\star$ in the definition of $\mathfrak{u}_k$ with a supremum over ground-truth functions $\boldsymbol{h}$, $\bar{\mathfrak{u}}_k := \sup_{\boldsymbol{h} \in \mathcal{H}_1 \times \cdots \times \mathcal{H}_K} \inf \left\{ r \geq 0 : r^2 \geq \mathfrak{R}_{N_k}^k \left( \mathcal{G}_k(r; \boldsymbol{h}) \right) \right\}$, and then clearly $\mathfrak{u}_k \leq \bar{\mathfrak{u}}_k$.

**Theorem 2.** *Let $\mathcal{S} \subseteq \mathcal{S}_{\mathrm{lin}}$ be a set of linear scalarizations, and let Assumptions 1 to 3 hold. Then, if $\delta > 0$ is sufficiently small, the output $\{\widehat{g}_s : s \in \mathcal{S}\}$ from Algorithm 1 satisfies ($\mathcal{S}$-MOL) with probability $1 - \delta$ and $\varepsilon_s = s(\varepsilon_1, \ldots, \varepsilon_K)$, where*

$$\varepsilon_k \lesssim C_k \left( \bar{\mathfrak{u}}_k^2 + \mathfrak{l}_k^2 + \left( N_k^{-1} + n_k^{-1} \right) \log(4K/\delta) \right) \tag{13}$$

*and $C_k = \left( \nu^3 (1 + \mathrm{diam}_{\|\cdot\|_2}(\mathcal{Y})) / \gamma^2 \right) \max\{ L_k^2/\gamma^2 + B_k/\gamma, \ L_k^2/\mu_k^2 + B_k/\mu_k \}$.[4] If additionally Assumption 4 holds, then for $\mathfrak{l}_{\mathcal{S}}^2 = \sup_{s \in \mathcal{S}} s(\mathfrak{l}_1^2, \ldots, \mathfrak{l}_K^2)$ and $n_{\mathcal{S}} = \left( \sup_{s \in \mathcal{S}} s(1/n_1, \ldots, 1/n_K) \right)^{-1}$ we have*

$$\varepsilon_k \lesssim \widetilde{C}_k \left( \mathfrak{u}_k^2 + \mathfrak{l}_{\mathcal{S}}^2 + (N_k^{-1} + n_{\mathcal{S}}^{-1}) \log(4K/\delta) \right), \tag{14}$$

*with $\widetilde{C}_k = C_k \cdot (\eta\nu/\gamma)^2 \max_{k \in [K]} \left( B_k/\mu_k + L_k^2/\mu_k^2 \right)$.*

The proof of Theorem 2 can be found in Appendix D.4. By comparing Eq. (10) with Eq. (13), we can see that, under the additional assumptions, Theorem 2 yields much better rates than Theorem 1, whenever the critical radii are (much) smaller than the global Rademacher complexities. Effectively, Theorem 1 provides a "slow rate" analysis, while Theorem 2 provides a "fast rate" analysis. Additionally, Theorem 2 avoids the "doubly slow rate" $\mathfrak{R}_{n_k}^k (\mathcal{H}_k)^{1/2}$ that appears in Theorem 1, and hence can potentially yield a speed-up of power 4 over Theorem 1; e.g., if $\mathcal{H}$ has VC (subgraph) dimension $d_{\mathcal{H}}$, the label complexity reduces to order $\widetilde{O}(K d_{\mathcal{H}}/\varepsilon)$ compared to the $\widetilde{O}(K d_{\mathcal{H}}/\varepsilon^4)$ from Theorem 1.

In the setting where the algorithm has access to the marginals $\{P_X^k\}_{k=1}^K$, called the *ideal semi-supervised setting* [70], the proof of Theorem 2 also yields a slightly tighter bound than (13) (by combining Eqs. (32) and (36)). Under Assumptions 1 and 2, we obtain $\varepsilon_s = s(\varepsilon_1, \ldots, \varepsilon_K)$ with $\varepsilon_k \lesssim C_k \left( \mathfrak{l}_k^2 + n_k^{-1} \log(2K/\delta) \right)$, where $C_k = \frac{\nu^3}{\gamma^2} \left( 1 + \mathrm{diam}_{\|\cdot\|_2}(\mathcal{Y}) \right) \left( B_k/\mu_k + L_k^2/\mu_k^2 \right)$.

**Adaptivity and weakening Assumption 4.** While Eq. (13) depends on the *worst* location of the true Pareto set $g_s$ in $\mathcal{G}$ (through $\bar{\mathfrak{u}}_k$), Eq. (14) refines this bound by also showing the *adaptivity* of the algorithm to the specific location of the true Pareto set $g_s$ in $\mathcal{G}$. Depending on the geometry of $\mathcal{G}$, this set may lie in a "low complexity region" of $\mathcal{G}$. If that is the case, then the radii $\mathfrak{u}_k$ can be smaller than $\bar{\mathfrak{u}}_k$, and the bound adapts to this low complexity. But this comes at a cost: to prove Eq. (14), we require the norm equivalence from Assumption 4, and have to replace $\mathfrak{l}_k$ by $\mathfrak{l}_{\mathcal{S}}$. Intuitively, the distance of $\widehat{g}_s$ to $g_s$ can only be controlled in the norm $\|\cdot\|_s$; in particular, if $\widehat{\lambda}_k = 0$, then there is no reason that $\widehat{g}_s$ should be close to $g_s$ in the norm $\|\cdot\|_k$. But $\mathfrak{u}_k$, defined through $\|\cdot\|_k$, has to bound the $k$th coordinate for *all* scalarizations $s \in \mathcal{S}$, making the norm equivalence from Assumption 4 necessary. For *finite* sets of scalarizations, on the other hand, this can be avoided (but replaced by a union bound), see Corollary A.2. Hence, there seems to be an inherent tension between controlling the error for all scalarizations simultaneously and proper adaptivity to the local complexity of the problem. It is interesting to explore this tension further.

## 5 Example: non-parametric regression with Lipschitz functions

We now exemplify the benefit of Theorem 2 in an example where the localized rates are much faster than unlocalized ones. More examples are presented in Appendix C.

Let $\mathcal{X} = [0, 1]$, $\mathcal{Y} = [0, 1]$ and let $\ell_k$ be the square loss. Define for $0 < L_{\mathcal{H}} < L_{\mathcal{G}}$ the function classes $\mathcal{H} = \{h : [0, 1] \to [0, 1] : h \text{ is } L_{\mathcal{H}}\text{-Lipschitz}\}$ and $\mathcal{G} = \{g : [0, 1] \to [0, 1] : g \text{ is } L_{\mathcal{G}}\text{-Lipschitz}\}$. Furthermore, let $K = 2$ and $P_X^k$ have a density $p_k$ on $[0, 1]$ with respect to the Lebesgue measure. For Eq. (3) to hold, assume that there exist two functions $f_1^\star, f_2^\star \in \mathcal{H}$ for which $\mathbb{E}[Y^k | X^k = x] = f_k^\star(x)$ for all $x \in [0, 1]$. We now apply Theorem 2 to obtain upper bounds for $\mathcal{S}$-MOL in this setting.

**Corollary 1.** *Let $\mathcal{S} \subseteq \mathcal{S}_{\mathrm{lin}}$ be a set of linear scalarizations and assume the functions from Eq. (9) are $L_{\mathcal{G}}$-Lipschitz. Then the output $\{\widehat{g}_s : s \in \mathcal{S}\}$ of Algorithm 1 satisfies ($\mathcal{S}$-MOL) with probability $0.99$ and $\varepsilon_s = s(\varepsilon_1, \ldots, \varepsilon_K)$ where $\varepsilon_k \lesssim \left( L_{\mathcal{H}}/n_k \right)^{2/3} + \left( L_{\mathcal{G}}/N_k \right)^{2/3}$ for all $s \in \mathcal{S}$.*

---

[4]We assume $\min \{L_k, \mu_k\} / \gamma \geq 1$ for all $k \in [K]$; otherwise, remove the squares on each ratio.

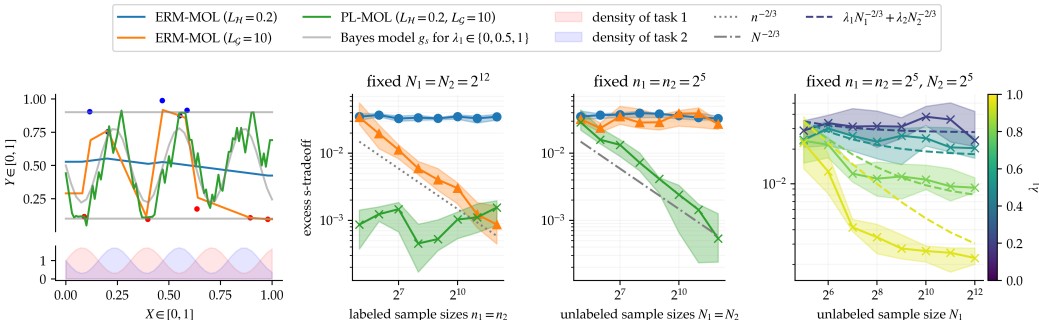

Figure 2: *On the left:* one fit of the methods on 5 labeled and 100 unlabeled samples with weights $\boldsymbol{\lambda} = (1/2, 1/2)$. *In the center*: excess $s$-trade-off as a function of labeled and unlabeled sample sizes for fixed weights $\boldsymbol{\lambda} = (1/2, 1/2)$. We fix the unlabeled and labeled sample sizes to $2^{12}$ and $2^5$, respectively. *On the right:* the excess $s$-trade-off of PL-MOL as a function of unlabeled sample size $N_1$ while $n_1 = n_2 = N_2 = 2^5$ are fixed, and for varying weights. We repeat each experiment 10 times and show median, 20% and 80% quantiles.

The proof of Corollary 1 can be found in Appendix C.3. Note that we recover the familiar minimax rate $n^{-2/3}$ of Lipschitz regression. In comparison, the crude, unlocalized bound from Theorem 1 would yield the potentially much slower rates $L_{\mathcal{H}}^{1/4} n_k^{-1/4} + L_{\mathcal{G}}^{1/2} N_k^{-1/2}$.

We illustrate Corollary 1 in Fig. 2 on the following example: Let $\mathcal{H}$ be a set of almost constant functions (that is, $L_{\mathcal{H}} = 0.2$), and let $f_1^\star \equiv a$ and $f_2^\star \equiv b$ for two constants $a, b \in [0, 1]$. Minimizing $\mathcal{T}_s(h)$ for the weights $\boldsymbol{\lambda} = (1/2, 1/2)$ over $\mathcal{H}$ yields the solution $h_s \approx (a + b)/2$ while for large enough $L_{\mathcal{G}}$, the solution in $\mathcal{G}$ becomes $g_s = (p_1 a + p_2 b)/(p_1 + p_2)$. On the left of Fig. 2, we show one data instance and the resulting models from Algorithms 1 and 2 when the densities are $p_1(x) = 0.7 \sin(20x) + 1$ and $p_2 = 2 - p_1$. In the center and on the right, we show the excess $s$-trade-off in this setting as a function of sample size. We can see the rates predicted by Corollary 1: when we fix the unlabeled sample sizes as large enough ($N_1 = N_2 = 2^{12}$), PL-MOL achieves a small excess $s$-trade-off already for small labeled sample sizes. Meanwhile, ERM-MOL requires a labeled sample size to be of the same order $2^{12}$ before it achieves a similar excess $s$-trade-off. At the same time, if we fix the labeled sample size sufficiently large to learn the almost constant functions in $\mathcal{H}$, only PL-MOL improves with an increasing number of unlabeled data. In both cases, the familiar $n^{-2/3}$-rate from Lipschitz regression is observable, as also predicted by Corollary 1. Finally, on the right of Fig. 2, we see that if we keep all sample sizes fixed—except for $N_1$—, then the rates are eventually bottlenecked by the harder task for all scalarizations; the risks stagnate at $\lambda_2 N_2^{-2/3} \asymp \lambda_2$.

## 6 Discussion

This work studies when it is possible to mitigate the statistical cost of multi-objective learning, in which we illuminate the roles of unlabeled data and of the loss functions. This need arises because the function classes that contain models achieving good trade-offs may need to be much larger than those that are well-suited for any one task. We show that for general losses, the label complexity of learning multiple trade-offs simultaneously in a class $\mathcal{G}$ is determined solely by the complexity of $\mathcal{G}$, even when the learner has full access to marginal distributions and the Bayes optimal models for each task (Proposition 1). But for Bregman losses, a simple pseudo-labeling algorithm can significantly reduce the label complexity (Theorem 1), where unlabeled data can fully absorb the statistical cost of the expressive model class. Our analysis with local Rademacher complexities further refines these bounds (Theorem 2) and shows adaptivity of the algorithm under some conditions.

The key property that the pseudo-labeling algorithm exploits is the risk decomposition from Lemma 1, which is unique to (generalized) Bregman losses [28]. Nevertheless, it is interesting to determine for exactly which losses the semi-supervised setting can improve upon the supervised one beyond Bregman losses. Under stronger assumptions, we provide a first result of this kind in Appendix B.

Future work may also investigate the tension between controlling the errors of all scalarizations and adaptive rates, and in this context, whether Assumption 4 is really necessary (see discussion in Section 4.2). Moreover, it would be interesting to relax structural assumptions in Theorem 2, e.g., by generalizing it to non-linear scalarizations, and to apply our framework to generative models.

## Acknowledgements

We thank Konstantin Donhauser for helpful discussions. TW was supported by SNSF Grant 204439 and JP by SNSF Grant 218343. GS was partially supported by the NSF award CCF-2112665 (TILOS), DARPA AIE program, the U.S. Department of Energy, Office of Science, the Facebook Research Award, as well as CDC-RFA-FT-23-0069 from the CDC's Center for Forecasting and Outbreak Analytics. This work was done in part while the authors were visiting the Simons Institute for the Theory of Computing.

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

# Appendix

## Table of Contents

## A  Related work

We review related work; for $\mathcal{S}$-MOL in Appendix A.1 and more adjacent literature in Appendix A.2.

**Semi-supervised learning.** Semi-supervised single-objective learning is a well-established field of research, and the question of *when and how* unlabeled data can help in a single learning problem is rather subtle [17, 26, 76, 6]. Interestingly, the reason that unlabeled data helps in our setting is quite different from how it can help in single-objective learning. Contrary to our setup, results that demonstrate a benefit of semi-supervised settings in single-objective learning usually require the marginals to carry some form of information about the labels (such as clusterability, manifold structure, low-density separation, smoothness, compatibility, etc.) [55, 52, 17], without which semi-supervised learners are no better than ones that discard the unlabeled data altogether [26, 13]. Our results, on the other hand, hold regardless of such assumptions: if the likelihood of a sample is higher under one task than another, a model with a good trade-off prioritizes that task, and unlabeled data enables (implicitly) estimating that likelihood. This is true, even if that likelihood carries no additional information about the labels.

---

**Algorithm 2** ERM for Multi-objective Learning (ERM-MOL)

---

**Input:** Labeled data $\left\{(X_i^k, Y_i^k)\right\}_{i=1}^{n_k}$, hypothesis space $\mathcal{G}$, scalarization set $\mathcal{S}$.

1: **for** $k \in [K]$ **do**
2:   Define the empirical risk functional: $\widehat{\mathcal{R}}_k(g) := \frac{1}{n_k} \sum_{i=1}^{n_k} \ell_k\left(Y_i^k, g(X_i^k)\right)$.
3: **end for**
4: **for** $s \in \mathcal{S}$ **do**
5:   Minimize the empirical $s$-trade-off: $\widehat{g}_s = \arg\min_{g \in \mathcal{G}} s\left(\widehat{\mathcal{R}}_1(g), \ldots, \widehat{\mathcal{R}}_K(g)\right)$.
6: **end for**
7: Return $\left\{\widehat{g}_s : s \in \mathcal{S}\right\}$.

---

**A priori and a posteriori decision making.** In multi-objective optimization, decision makers can be broadly categorized based on whether they have an *a priori* or an *a posteriori* preference over Pareto solutions [29, 33]. An *a priori* decision maker aims to recover a specific Pareto model, which is the solution to a trade-off $\mathcal{T}_s$ that is known beforehand. In contrast, an *a posteriori* decision maker will first recover the whole Pareto set. Recall from Section 2.3 or [23] that, under mild conditions, this entails solving a family of optimization problems $\forall s \in \mathcal{S}$, $\min_{g \in \mathcal{G}} \mathcal{T}_s(g)$. The preference of such a decision maker is then informed by the set of trade-offs that are possible.

The learning version of the problem has been studied for both types of decision makers, where the trade-off functionals need to be estimated from data. This leads to two types of algorithms and generalization bounds. The *a priori* approach has been especially developed in the context of learning with fairness or multi-group constraints [27, 4, 51, 73]. The *a posteriori* approach, to which our work mostly belongs, has been studied by [19, 59].

**Multi-distribution learning.** In the (supervised) multi-distribution learning (MDL) setting, the goal is to learn only one $s$-trade-off for the scalarization $s(\boldsymbol{v}) = \max v_k$, which belongs to the family of *a priori* decision making. Then, for VC-classes of dimension $d_\mathcal{G}$, the label complexity to achieve excess $s$-trade-off $\varepsilon > 0$ is $\widetilde{\Theta}((K + d_\mathcal{G})/\varepsilon^2)$ using an on-demand sampling framework, in which the algorithm is allowed to decide which distribution to sample from sequentially [27, 4, 51, 73]. Importantly, this adaptive sampling improves upon the "trivial" rate $\Theta(Kd_\mathcal{G}/\varepsilon^2)$ (see [73] and Appendix A.1) by removing the multiplicative dependence on the number of objectives. For non-adaptive sampling, the rate $\Theta(Kd_\mathcal{G}/\varepsilon^2)$ is tight, that is, the fact that the algorithm has to solve only one scalarization does not improve upon the sample complexity of solving all scalarizations, cf. Corollary A.1. Of course, the statistical complexity under adaptive sampling must fail to hold for $\mathcal{S}$-MOL with multiple scalarizations, because it includes all individual learning tasks. MDL is also related to collaborative (where the tasks are assumed to share a ground-truth), federated, and group DRO frameworks, for which we refer the readers to the discussions in [27, 73, 14].

In [5], the authors propose a semi-supervised framework for group DRO (a problem related to MDL). The underlying assumption in [5] is that for each label-scarce group, there exists a group with sufficiently much labeled data and which is "related enough" for cross-group pseudo-labeling to be effective (similar to the collaborative learning setup).

## A.1 A posteriori multi-objective learning

**Empirical risk minimization for $\mathcal{S}$-MOL.** Applying empirical risk minimization (ERM) on labeled data to solve $\mathcal{S}$-MOL was analyzed in [19, 59] and in [18] through algorithmic stability. ERM, or perhaps more aptly empirical trade-off minimization, is a natural approach to learning all Pareto solutions. The idea is simply to use labeled data sampled for each of the $K$ tasks to empirically estimate the $s$-trade-off functional $\mathcal{T}_s$ of any model. The Pareto set can then be found by minimizing the estimated trade-offs. This algorithm, that we call *empirical risk minimization for multi-objective learning* (ERM-MOL), is formalized in Algorithm 2.

Learning the Pareto set through ERM has been described and analyzed by [19], where $\mathcal{S}$ is a family of linear scalarizations. In particular, they provide a sample complexity upper bound that depends on the complexity of $\mathcal{S}$ through a covering number of the weights that appear in $\mathcal{S}$. Later, [59] extended the ERM framework to go beyond the empirical estimator of the risk functionals, allowing

for any "statistically valid" estimator based on uniform convergence. They further improve the sample complexity upper bound by removing dependency on $\mathcal{S}$ in [59, Theorem 2]. Their result can be used to derive bounds for ERM in $\mathcal{S}$-MOL: we now instantiate their bound in our setting (see Section 2.1), making the following assumption to enable comparison with our results:

**Assumption 5** (Regularity conditions for ERM-MOL). The risk and excess risk functionals are equal,
$$\forall k \in [K]: \qquad \inf_{f \in \mathcal{F}_{\text{all}}} \mathcal{R}_k(f) = 0.$$

**Proposition A.1** (Sample complexity of ERM-MOL). *Suppose that Assumption 5 holds and that the loss $\ell_k(\cdot, \cdot)$ is bounded by $B$ and $L_k$-Lipschitz continuous in the second argument for each $k \in [K]$. Let $\mathcal{S}$ be any class of monotone scalarizations satisfying reverse triangle inequality and positive homogeneity (6). Then, for any $\delta \in (0,1)$, the class of solutions returned by Algorithm 2, $\{\widehat{g}_s : s \in \mathcal{S}\}$, satisfies ($\mathcal{S}$-MOL) with probability $1 - \delta$ and $\varepsilon_s = s(\varepsilon_1, \ldots, \varepsilon_K)$, where for each $k \in [K]$, $\varepsilon_k$ is given by:*

$$\varepsilon_k = 6L_k \mathfrak{R}_{n_k}^k(\mathcal{G}) + 2B \left( \frac{2 \log(K/\delta)}{n_k} \right)^{1/2}. \tag{15}$$

We demonstrate the implications of this bound for a VC class and for linear scalarizations below, using the VC bound on the Rademacher complexity (Lemma E.6), see Appendix A.3 for proofs.

**Corollary A.1.** *Let $\mathcal{G}$ be any hypothesis class with VC dimension $d_{\mathcal{G}} \in \mathbb{N}$ on data domain $\mathcal{X} \times \mathcal{Y}$ where $\mathcal{Y} = \{0,1\}$. For each task $k \in [K]$, define $\ell_k(y, y') = \mathbf{1}\{y \neq y'\}$ be the zero-one loss (cf. Definition 3). Let $(P^1, \ldots, P^K)$ be any tuple of data distributions over $\mathcal{X} \times \mathcal{Y}$. Then, for any $\delta, \varepsilon > 0$, the output of Algorithm 2 $\{\widehat{g}_s : s \in \mathcal{S}_{\text{lin}}\}$ satisfies ($\mathcal{S}$-MOL) with probability $1 - \delta$ and $\varepsilon_s = \varepsilon$ for all $s \in \mathcal{S}_{\text{lin}}$ whenever the number of samples is at least $n_k = \Omega\left( \frac{d_{\mathcal{G}} + \log(K/\delta)}{\varepsilon^2} \right)$ for each $k \in [K]$.*

**Dependence on the size of $\mathcal{S}$ in our results.** The authors in [59] noted that the dependence on $\mathcal{S}$ in [19] is sub-optimal, in the worst case by a factor of $K \log n_k$, and that such a dependence could be removed. Here, we should add that this is only true because in [59] the learning bounds are globally uniform—no localization bounds were derived. Similarly, the bound from our Theorem 1 is also independent of the size of $\mathcal{S}$. Theorem 2, on the other hand, paints a more nuanced picture: the size of the sets $\mathcal{G}_k(r; \boldsymbol{f}^\star)$ from Eq. (11) depends on the size of $\mathcal{S}$ through a union: if all local neighborhoods of the $g_s$ are "similar," then $\mathcal{S}$ does not affect the bound at all. However, if the local neighborhoods are very different, then the union may be larger than any of the individual neighborhoods and hence the bound will grow with the size of $\mathcal{S}$; see the right side of Fig. 5. See also the discussion in Section 4.2. Nonetheless, if $\mathcal{S}$ is finite, Theorem 2 also yields the following bound.

**Corollary A.2.** *Let $\mathcal{S} \subseteq \mathcal{S}_{\text{lin}}$ be finite and let Assumptions 1 and 2 hold. Define*
$$\mathfrak{u}_k(s) := \inf \left\{ r \geq 0 : r^2 \geq \mathfrak{R}_{N_k}^k \left( r\mathcal{B}_{\|\cdot\|_k} \cap (\mathcal{G} - g_s) \right) \right\}.$$

*Then, if $\delta > 0$ is sufficiently small, the output $\{\widehat{g}_s : s \in \mathcal{S}\}$ from Algorithm 1 satisfies ($\mathcal{S}$-MOL) with probability $1 - \delta$ and $\varepsilon_s = s(\varepsilon_1, \ldots, \varepsilon_K)$, where*

$$\varepsilon_k \lesssim \widetilde{C}_k \left( \mathfrak{u}_k^2(s) + \mathfrak{l}_k^2 + (N_k^{-1} + n_k^{-1}) \log(4K |\mathcal{S}|/\delta) \right),$$

*with $\widetilde{C}_k = \widetilde{C}_k(s)$ from Eq. (14) where $\eta^2 = \eta^2(s) := \max\{1/\lambda_k : k \in [K], \lambda_k > 0\}$ for $s = s_{\boldsymbol{\lambda}}^{\text{lin}}$.*

*Proof of Corollary A.2.* Consider the setting where $\mathcal{S} = \{s_{\boldsymbol{\lambda}}^{\text{lin}}\}$ is a singleton. Because we only consider this one scalarization, we can make the following case distinction for each $k \in [K]$: either $\lambda_k = 0$, so we can ignore index $k$ completely, or $\lambda_k > 0$ and so ess sup $dP_X^k/d(\sum_{j=1}^K P_X^j) \leq 1/\lambda_k$. Hence, Assumption 4 is satisfied for $\mathcal{S} = \{s_{\boldsymbol{\lambda}}^{\text{lin}}\}$ with $\eta^2(s_{\boldsymbol{\lambda}}^{\text{lin}}) = \max\{1/\lambda_k : k \in [K], \lambda_k > 0\}$. The bound for $\mathcal{S} = \{s_{\boldsymbol{\lambda}}^{\text{lin}}\}$ follows from Theorem 2, and the corollary for a finite $\mathcal{S}$ follows from a union bound. $\qquad \square$

**Semi-supervised $\mathcal{S}$-MOL.** As far as we are aware, we are the first to study the $\mathcal{S}$-MOL problem in the general semi-supervised setting. The closest work to ours is [65], where the question of learning Pareto manifolds in high-dimensional Euclidean space was studied in a semi-supervised setting. They assume that 1) the ground-truths exhibit a sufficiently sparse structure and 2) the objectives have

a *benign parametrization* (their Assumptions 1,2, and 3): the paper considers parametric function classes, and the algorithm that achieves the bounds requires knowledge about a parameter $\theta_k \in \mathbb{R}^q$ so that $\mathcal{R}_k$ depends on distribution $P^k$ only through $\theta_k$. Estimating these parameters and then performing standard multi-objective optimization can enable learning in high dimensions. The resulting two-stage estimator is similar to our pseudo-labeling algorithm, and they can coincide, e.g., for linear regression with square loss. Moreover, in [65] the *necessity* of unlabeled data in high-dimensional linear regression is shown. While we borrow an idea for the stability argument in our Proposition D.1, our results apply to far more general settings.

**Comparison of label sample complexities.** In order to compare the label sample complexity of our results with prior work, we summarize the resulting bounds in Table 1 for VC (subgraph)-classes $\mathcal{G}$ and $\mathcal{H}_k = \mathcal{H}$ with VC dimensions $d_\mathcal{G}, d_\mathcal{H}$. Recall that in the ideal setting, the marginals are known.

Table 1: Label complexities up to logarithmic factors from this (gray) and prior work for VC (subgraph) classes. It holds that $d_\mathcal{H} \leq d_\mathcal{G}$ and potentially $d_\mathcal{H} \ll d_\mathcal{G}$. A definition of $d_\Theta$ is in Appendix B; both $d_\Theta \ll d_\mathcal{G}$ and $d_\Theta \gg d_\mathcal{G}$ are possible. Note that these results are not strictly comparable, as they depend on varying technical assumptions.

| | zero-one loss | | Bregman loss |
|---|---|---|---|
| **problem class** | upper bound | lower bound | upper bound |
| supervised MDL | $\frac{d_\mathcal{G}+K}{\varepsilon^2}$ [73, 51] | $\frac{d_\mathcal{G}+K}{\varepsilon^2}$ [27] | $\frac{d_\mathcal{G}+K}{\varepsilon^2}$ [73] |
| supervised $\mathcal{S}$-MOL | $\frac{Kd_\mathcal{G}}{\varepsilon^2}$ [59] / Cor. A.1 | $\frac{Kd_\mathcal{G}}{\varepsilon^2}$ Prop. 1 | $\frac{Kd_\mathcal{G}}{\varepsilon^2}$ [59] / Prop. A.1 |
| ideal semi-sup. $\mathcal{S}$-MOL | $\frac{Kd_\mathcal{G}}{\varepsilon^2}$ [59] / Cor. A.1 | $\frac{Kd_\mathcal{G}}{\varepsilon^2}$ Prop. 1 | $\frac{Kd_\mathcal{H}}{\varepsilon^4}$ Thm. 1 |
| ideal semi-sup. $\mathcal{S}$-MOL (with stronger assumptions) | $\frac{Kd_\Theta}{\varepsilon^4}$ Prop. B.1 | — | $\frac{Kd_\mathcal{H}}{\varepsilon}$ Thm. 2 |

## A.2 Adjacent related works

There are many works considering *multi-risk settings* in different contexts (not to be confused with the multiple competing risks in survival analysis, cf. [34]), for example, in fairness or insurance mathematics through the lens of multiple quantile risk measures [21, 37, 58]. Our work specifically is related to the fields of *ensembling, multi-task learning, and Pareto set learning*.

**Learning multiple models for one task.** Recall that in our Algorithm 1, we first learn multiple models (one per task), and combine them into a family of models that trade off the different risks. In comparison, there are many different ways in which combining multiple models can also help *on a single task*, usually by using some sort of ensembling. For instance:

- *Stacked generalization* combines multiple base models via a meta-model that takes their predictions as input features and outputs the final prediction [66, 46].
- *Mixture-of-Experts* models maintain a collection of expert predictors, and use a routing mechanism to select one or more experts based on the new input. This routing is often done through a direct soft gating or a weighted combination of the models [45].
- *Boosting* aggregates multiple weak learners to form a single strong predictor for one task, typically through sequential training where each model corrects the errors of the previous ensemble [25].

In contrast to any of these methods, our pseudo-labeling algorithm uses the predictions from individual models (in our work ERMs for simplicity) as training targets and fits a new model (or family of models) from scratch using the unlabeled inputs. This distinction is essential: unlike the aforementioned methods, our algorithm does not aggregate existing models to solve a single task, but instead leverages them as a supervisory signal to reduce the statistical cost of learning trade-offs in a richer function class. In particular, the described methods do not address the core challenge in MOL: the need to reconcile conflicting objectives within a single model. Our method explicitly constructs a family of joint predictors that trades off competing risks and can—or sometimes even must—deviate significantly from any of the base models.

**Learning multiple models for multiple tasks.** Multi-task learning (MTL), including semi-supervised MTL, is a problem that is related to MOL in that both are used in settings where multiple

learning problems need to be solved. However, in MTL, the aim is to learn multiple models, one per task, and exploit relatedness between tasks to improve sample complexity [39, 72]. As such, the problem of striking a trade-off, which is at the heart of MOL, is not present in MTL. For example, suppose a new instance $x \in \mathcal{X}$ is observed. In MTL, we can make multiple different predictions, one per task, in the hope that each prediction is good for the corresponding task. In MOL, on the other hand, we have to commit to one prediction for all tasks. Aside from these differences, as mentioned, if there is a relationship between the different learning problems, we could employ off-the-shelf MTL algorithms to adapt our pseudo-labeling algorithm by learning the task-specific models in the first part of the algorithm (Line 2 in Algorithm 1). Finally, from a technical perspective, it is worth mentioning that (localized) Rademacher complexities have been used for MTL in [67, 3, 41, 50].

**Learning for multi-objective optimization.** A recent line of research has introduced the so-called *Pareto set learning* (PSL) framework [54, 38, 48], which has found various applications, e.g., in finetuning language models on multiple objectives [64]. PSL is an approach to making learning algorithms such as Algorithms 1 and 2 computationally tractable: instead of producing a family of models, one for each trade-off, PSL approximates this family with one fixed function that takes both weights of the objectives and covariates as input (often called a *hypernetwork* [48]). However, importantly, there is no direct connection of PSL to the *learning* part of the MOL problem: it is actually *purely a computational technique*. Specifically, if one approximates the outputs of Algorithms 1 and 2 with PSL, then it inherits their statistical guarantees up to the approximation errors. The name Pareto set *learning* has its origin in the fact that to find such a PSL function, it is common to minimize some expected scalarization, where the expectation is taken with respect to weights of the objectives [71]. A standard way to make this tractable is to *sample the weights* [48]. Generalization is then usually discussed in terms of the number of sampled weights, not the data. See also [65] for a discussion. Finally, beyond hypernetworks, various other learning techniques have been deployed for multi-objective optimization when evaluating the objectives is expensive, such as active learning in [75, 74, 31].

### A.3 Proofs for ERM-MOL

*Proof of Proposition A.1.* The proof is analogous to the proof of [59, Theorem 2], additionally using Rademacher complexity and McDiarmid's bound to bound the supremum (denoted $C_N$ in [59]) and slightly different assumptions on the scalarizations. We repeat the proof here for completeness.

Fix $\delta > 0$. By a standard Rademacher bound (also see Appendix D.3), the following generalization guarantee holds for each task $k \in [K]$,

$$\mathbb{P}\left(\forall g \in \mathcal{G}: \quad \left|\widehat{\mathcal{R}}_k(g) - \mathcal{R}_k(g)\right| \leq \varepsilon_k/2\right) \geq 1 - \frac{\delta}{K}. \tag{16}$$

For any scalarization $s$, let $\widehat{\mathcal{T}}_s$ denote the empirical $s$-trade-off

$$\widehat{\mathcal{T}}_s(g) = s\big(\widehat{\mathcal{R}}_1(g), \ldots, \widehat{\mathcal{R}}_K(g)\big).$$

Then, $\mathcal{T}_s$ is well-approximated by $\widehat{\mathcal{T}}_s$. By Assumption 5, $\mathcal{R}_k = \mathcal{E}_k$, so that when the event Eq. (16) holds for all $k \in [K]$, and this occurs with probability at least $1 - \delta$ by a union bound, we obtain that:

$$\begin{aligned}
\sup_{g \in \mathcal{G}} \left|\widehat{\mathcal{T}}_s(g) - \mathcal{T}_s(g)\right| &= \sup_{g \in \mathcal{G}} \left|s\big(\widehat{\mathcal{R}}_1(g), \ldots, \widehat{\mathcal{R}}_K(g)\big) - s\big(\mathcal{R}_1(g), \ldots, \mathcal{R}_K(g)\big)\right| \\
&\leq \sup_{g \in \mathcal{G}} s\big(\big|\widehat{\mathcal{R}}_1(g) - \mathcal{R}_1(g)\big|, \ldots, \big|\widehat{\mathcal{R}}_K(g) - \mathcal{R}_K(g)\big|\big) \\
&\leq s\big(\varepsilon_1/2, \ldots, \varepsilon_K/2\big), \tag{17}
\end{aligned}$$

where the first inequality used the reverse triangle inequality, and the second inequality used the above claim and the monotonicity of the scalarization. In particular, this will allow us to bound the excess $s$-trade-off of $\widehat{g}_s$ as follows:

$$\begin{aligned}
\mathcal{T}_s(\widehat{g}_s) - \mathcal{T}_s(g_s) &= \underbrace{\mathcal{T}_s(\widehat{g}_s) - \widehat{\mathcal{T}}_s(\widehat{g}_s)}_{(a)} + \underbrace{\widehat{\mathcal{T}}_s(\widehat{g}_s) - \widehat{\mathcal{T}}_s(g_s)}_{(b)} + \underbrace{\widehat{\mathcal{T}}_s(g_s) - \mathcal{T}_s(g_s)}_{(c)} \\
&\leq s\big(\varepsilon_1/2, \ldots, \varepsilon_K/2\big) + s\big(\varepsilon_1/2, \ldots, \varepsilon_K/2\big) \\
&\leq s\big(\varepsilon_1, \ldots, \varepsilon_K\big),
\end{aligned}$$

where the (a) and (c) terms both contribute at most $s(\varepsilon_1/2, \ldots, \varepsilon_K/2)$ error from Equation (17), while the (b) term is non-positive, since $\widehat{g}_s$ minimizes the empirical $s$-trade-off $\widehat{\mathcal{T}}_s$. The last inequality follows from the positive homogeneity of the scalarizations. $\qquad\square$

*Proof of Corollary A.1.* From Proposition A.1 and the VC bound on Rademacher complexity (Lemma E.6), there exists a constant $C > 0$ such that for each $k \in [K]$ we have $\varepsilon_k \le \varepsilon$ for whenever $n_k$ is sufficiently large:

$$n_k \ge \frac{1}{\varepsilon^2} \left( 72 C^2 L_k^2 d_{\mathcal{G}} + 16 B^2 \log \frac{K}{\delta} \right).$$

The result follows, since we have that for any $s \in \mathcal{S}_{\mathrm{lin}}$:

$$s(\varepsilon_1, \ldots, \varepsilon_K) \le s(\varepsilon, \ldots, \varepsilon) = \varepsilon,$$

which concludes the proof. $\qquad\square$

# B   Beyond Bregman losses: Pseudo-labeling for the zero-one loss

Recall that Proposition 1 is a worst-case negative result that rules out any benefit of unlabeled data for MOL, at least in the absence of any additional structure. Indeed, our main results focus on overcoming this hardness for more structured losses—namely, Bregman losses. But, there are other natural forms of structure to consider; in this section, we take an alternative approach.

To help motivate this next approach, let us revisit the reason that Bregman losses are amenable to the semi-supervised approach. As we discuss in Section 3.2, the crux is Lemma 1. It expresses the excess risk functionals $\mathcal{E}_k$ in terms of a discrepancy operator $d_k(\cdot\,;\cdot)$ and the Bayes-optimal model $f_k^\star$, as follows:

$$\mathcal{E}_k(\cdot) = d_k(\cdot; f_k^\star). \tag{18}$$

Estimating the discrepancy operator over $\mathcal{G}$ only makes use of unlabeled data. And even though learning the Bayes-optimal model requires labeled data, its statistical cost is mitigated by the knowledge that $\mathcal{H}_k$ contains the optimal model. Algorithm 1 precisely constructs estimators of $\mathcal{E}_k$ in this way, before solving for the $s$-trade-offs over the learned approximations. The close relationship between $\mathcal{E}_k$ and $f_k^\star$ described by Eq. (18) is specific to Bregman losses, see [62, 15, 28]. Nevertheless, the excess risk functional for other losses may have decompositions that are similar in spirit.

**Excess risk decomposition for the zero-one loss.**   To see another instance of excess-risk decomposition, let us revisit the setting of the worst-case examples in Proposition 1: the multi-objective binary classification setting with the zero-one loss (Definition 3). In this case, it is a standard result that the excess risk can be expressed in terms of the conditional mean of the labels $\theta_k(x) := \mathbb{E}[Y^k | X^k = x]$, as in [20, Section 2.1]. In particular, the excess risks for the zero-one loss is given by

$$\mathcal{E}_k(f) = \mathbb{E}\left[ \left| 2\theta_k(X^k) - 1 \right| \cdot \left| f(X^k) - \mathbf{1}\left\{ \theta_k(X^k) \ge 1/2 \right\} \right| \right]. \tag{19}$$

Thus, for the zero-one loss, the form of its corresponding excess risk functional is $\mathcal{E}_k(\cdot) = d_k^{0/1}\left( \cdot\,; \theta_k \right)$, where let $d_k^{0/1}(\cdot\,;\cdot)$ be the "zero-one discrepancy" operator, given by

$$d_k^{0/1}(f; \theta) = \mathbb{E}\left[ \left| 2\theta(X^k) - 1 \right| \cdot \left| f(X^k) - \mathbf{1}\left\{ \theta(X^k) \ge 1/2 \right\} \right| \right],$$

where $\theta : \mathcal{X} \to [0, 1]$ is a conditional mean. Notice that the operator $d_k^{0/1}(\cdot\,;\cdot)$ depends only the marginal distribution $P_X^k$, as was the case for Bregman losses (cf. Eq. (18)). This suggests that a semi-supervised approach analogous to Algorithm 1 becomes possible if the *regression problem* of learning $\theta_k$ is easy. Note that this is potentially much harder than solving the individual classification tasks, and hence different from the original premise of this work.

We now formalize this intuition. For each $k \in [K]$, let $\Theta_k$ be a class of functions $\theta : \mathcal{X} \to [0, 1]$. In lieu of assuming that $f_k^\star \in \mathcal{H}_k$, we now assume that the true conditional mean $\theta_k$ is contained in $\Theta_k$ (we do not assume that $\Theta_k \subseteq \mathcal{G}$, especially as $\Theta_k$ consists of regression functions and $\mathcal{G}$

consists of classifiers). Define the empirical regression map $\widehat{\theta}_k(\cdot)$ and empirical zero-one discrepancy $\widehat{d}_k^{0/1}(g;\widehat{\theta}_k)$ as follows:

$$\widehat{\theta}_k(\cdot) := \arg\min_{\theta \in \Theta_k} \frac{1}{n_k} \sum_{i=1}^{n_k} (Y_k - \theta(X^k))^2$$

$$\widehat{d}_k^{0/1}(g;\widehat{\theta}_k) := \frac{1}{N_k} \sum_{i=1}^{N_k} \left[ \left| 2\widehat{\theta}_k(\widetilde{X}_i^k) - 1 \right| \cdot \left| g(\widetilde{X}_i^k) - 1\left\{ \widehat{\theta}_k(\widetilde{X}_i^k) \geq 1/2 \right\} \right| \right].$$

And finally, define the analogous $s$-scalarization as $\widehat{d}_s^{0/1}(g;\widehat{\boldsymbol{\theta}}) := s\big(\widehat{d}_1^{0/1}(g;\widehat{\theta}_1), \ldots, \widehat{d}_K^{0/1}(g;\widehat{\theta}_K)\big)$.

**Proposition B.1.** *In the multi-objective binary classification setting (Definition 3), let $\mathcal{S}$ be a class of scalarizations satisfying the reverse triangle inequality and positive-homogeneity (6). Assume that $\mathbb{E}[Y^k|X=\cdot] \in \Theta_k$. Then, $\{\widehat{g}_s \in \arg\min_{g \in \mathcal{G}} \widehat{d}_s^{0/1}(g;\boldsymbol{\theta}) : s \in \mathcal{S}\}$ satisfies ($\mathcal{S}$-MOL) with probability $1 - \delta$ and $\varepsilon_s = s(\varepsilon_1, \ldots, \varepsilon_k)$, where*

$$\varepsilon_k = 12\left( \mathfrak{R}_{N_k}^k(\mathcal{G}) + \sqrt{\frac{\log(2K/\delta)}{N_k}} + \sqrt{\mathfrak{R}_{n_k}^k(\Theta_k) + \left(\frac{\log(2K/\delta)}{n_k}\right)^{1/2}} \right).$$

The proof of Proposition B.1 is in Appendix B.1 and is analogous to that of Theorem 1. Here, we can also give a data-independent sample complexity bound. Let $d_{\Theta_k}$ denote the VC subgraph dimension (a.k.a. pseudo dimension) of $\Theta_k$ and $d_{\mathcal{G}}$ the VC dimension of $\mathcal{G}$. Then, Proposition B.1 implies a label sample complexity on the order of $O(Kd_{\Theta_k}/\varepsilon^4)$ and an unlabeled sample complexity of $O(Kd_{\mathcal{G}}/\varepsilon^2)$ (see Lemma E.6).

In summary, the hardness result Proposition 1 shows that, in the worst-case, the Bayes classifiers $f_k^\star$ are uninformative for making appropriate trade-offs over zero-one losses. Instead, Eq. (19) suggests that the relevant information is actually captured by the Bayes regressors $\theta_k$. Proposition B.1 makes this intuition rigorous: if we have additional structure, given here in the form of $\Theta_k$, we can expect benefits of semi-supervision even for the zero-one loss, as long as the Rademacher complexity of $\Theta_k$ is manageable.

## B.1 Proof of Proposition B.1

For this proof, define $f_\theta(x,g) = |2\theta(x) - 1| \cdot |g(x) - 1\{\theta(x) \geq 1/2\}|$. Then

$$|f_\theta(x,g) - f_{\theta'}(x,g)| \leq \left||2\theta(x) - 1| - |2\theta'(x) - 1|\right|$$
$$+ |2\theta'(x) - 1| \cdot |1\{\theta'(x) \geq 1/2\} - 1\{\theta(x) \geq 1/2\}|$$
$$\overset{(a)}{\leq} 2|\theta(x) - \theta'(x)| + 2|\theta(x) - \theta'(x)| \leq 4|\theta(x) - \theta'(x)|.$$

where in $(a)$ we used that the indicators can only disagree if $|\theta'(x) - \theta(x)| \geq |\theta(x) - 1/2|$. It follows that for any $g \in \mathcal{G}$,

$$\left| \mathcal{E}_k(g) - \widehat{d}_k^{0/1}(g;\widehat{\theta}_k) \right| \leq \left| \mathcal{E}_k(g) - d_k^{0/1}(g;\widehat{\theta}_k) \right| + \left| d_k^{0/1}(g;\widehat{\theta}_k) - \widehat{d}_k^{0/1}(g;\widehat{\theta}_k) \right|$$
$$= \left| \mathbb{E}\left[ f_{\theta_k}(X^k,g) - f_{\widehat{\theta}_k}(X^k,g) \right] \right| + \left| d_k^{0/1}(g;\widehat{\theta}_k) - \widehat{d}_k^{0/1}(g;\widehat{\theta}_k) \right|$$
$$\leq 4\mathbb{E}\left[ \left| \theta_k(X^k) - \widehat{\theta}_k(X^k) \right| \right] + \left| d_k^{0/1}(g;\widehat{\theta}_k) - \widehat{d}_k^{0/1}(g;\widehat{\theta}_k) \right|.$$

We bound each term separately. First, analogous to Eq. (30) in the proof of Theorem 1, since the square loss is 2-Lipschitz on $[0,1]$, with probability at least $1 - \delta/(2K)$

$$\mathbb{E}\left[ \left| \theta_k(X^k) - \widehat{\theta}_k(X^k) \right| \right] \leq \sqrt{\mathbb{E}\left[ \left( \theta_k(X^k) - \widehat{\theta}_k(X^k) \right)^2 \right]} \leq \sqrt{24\mathfrak{R}_{n_k}^k(\Theta_k) + 2\left(\frac{2\log(2K/\delta)}{n_k}\right)^{1/2}}.$$

For the second term, we use that $g \mapsto f_\theta(x,g)$ is 1-Lipschitz continuous. Let $c(x) = 1\{\theta(x) \geq 1/2\}$; then

$$|f_\theta(x,g) - f_\theta(x,g')| \leq |2\theta(x) - 1| \, ||g(x) - c(x)| - |g'(x) - c(x)|| \leq |g(x) - g'(x)|.$$

Hence, using contraction again, we get the analogous bound to Eq. (29) with probability $1 - \delta$

$$\sup_{g \in \mathcal{G}} \left| d_k^{0/1}(g; \widehat{\theta}_k) - \widehat{d}_k^{0/1}(g; \widehat{\theta}_k) \right| = \sup_{g \in \mathcal{G}} \left| \frac{1}{N_k} \sum_{i=1}^{N_k} f_{\widehat{\theta}_k}(\widetilde{X}_i^k, g) - \mathbb{E}\left[ f_{\widehat{\theta}_k}(X^k, g) \right] \right|$$

$$\leq 2\mathfrak{R}_{N_k}^k(\{x \mapsto f_{\widehat{\theta}_k}(x, g) : g \in \mathcal{G}\}) + \sqrt{\frac{2 \log(1/\delta)}{N_k}}$$

$$\leq 6\mathfrak{R}_{N_k}^k(\mathcal{G}) + \sqrt{\frac{2 \log(1/\delta)}{N_k}}.$$

In conclusion, we have proved that with probability $1 - \delta$, for all $k \in [K]$, we have

$$\sup_{g \in \mathcal{G}} \left| \mathcal{E}_k(g) - \widehat{d}_k^{0/1}(g; \widehat{\theta}_k) \right| \leq \sqrt{24\mathfrak{R}_{n_k}^k(\Theta_k) + 2\left(\frac{2\log(2K/\delta)}{n_k}\right)^{1/2}} + 6\mathfrak{R}_{N_k}^k(\mathcal{G}) + \sqrt{\frac{2\log(2K/\delta)}{N_k}}. \tag{20}$$

The rest of the proof then follows analogously to the proof of Theorem 1 provided in Appendix D.3 by replacing the "claim" with the uniform bound in Equation (20).

## C  More examples

In this section, we discuss two more examples: classification with logistic loss and another example applying Theorem 2 to linear regression.

### C.1  Binary classification

Denote for any $q \in [1, \infty]$ the norm balls $\mathcal{B}_q^d = \{w \in \mathbb{R}^d : \|w\|_q \leq 1\}$. Suppose that the covariates lie in the space $\mathcal{X} = \mathcal{B}_\infty^d \subset \mathbb{R}^d$, and that the labels in $\mathcal{Y} = [0, 1]$ for each task follow the Bernoulli distribution

$$Y^k | X^k = x \sim \mathrm{Ber}(\sigma(\langle x, w_k^\star \rangle))$$

where $\sigma(x) = 1/(1 + \exp(-x))$ denotes the sigmoid function and we assume that $w_k^\star \in \mathcal{B}_1^d$. This is the standard logistic regression setup. The Bayes-optimal models with respect to the logistic loss $\ell(y, \widehat{y}) = -(y \log(\widehat{y}) + (1 - y) \log(1 - \widehat{y}))$ are given by $f_k^\star(\cdot) = \sigma(\langle \cdot, w_k^\star \rangle) \in \mathcal{H} = \{h(x) = \sigma(\langle x, w \rangle) : w \in \mathcal{B}_1^d\}$. However, striking a good trade-off between the tasks within $\mathcal{H}$ can be impossible (see Fig. 3 for a simple example). To circumvent this issue, we may want to use some feature map $\Phi : \mathcal{B}_\infty^d \to \mathcal{B}_\infty^p$ with $p \gg d$, and then learn in the larger function class $\mathcal{G} = \{g(x) = \sigma(\langle \Phi(x), w \rangle) : w \in \mathcal{B}_1^p\}$. For example, $p = O(d^\kappa)$ and $\mathcal{H} \subseteq \mathcal{G}$ whenever $\Phi$ maps to the set of all polynomial features up to degree $\kappa$. In this setting, Algorithm 1 effectively exploits unlabeled data to achieve good trade-offs in the larger function class $\mathcal{G}$, as we show in Corollary C.1. This straightforwardly follows from Theorem 1; the proof is given in Appendix C.3.

**Corollary C.1** (Logistic regression)**.** *In the setting described above, let $\mathcal{S}$ be some class of scalarizations satisfying reverse triangle inequality and positive homogeneity. Suppose that $\min_{k \in [K]} N_k \geq \log(p + K)$ and $\min_{k \in [K]} n_k \geq \log(d + K)$. Then, the output of Algorithm 1 $\{\widehat{g}_s : s \in \mathcal{S}\}$ satisfies ($\mathcal{S}$-MOL) with probability at least $0.99$ and $\varepsilon_s = s(\varepsilon_1, \ldots, \varepsilon_K)$ where $\varepsilon_k \lesssim \left(\log(dK)/n_k\right)^{1/4} + \left(\log(pK)/N_k\right)^{1/2}$.*

We can also empirically observe the benefits of the semi-supervised method, Algorithm 1 (PL-MOL), over purely supervised approaches—namely, over running Algorithm 2 to learn models from either $\mathcal{H}$ (ERM-MOL linear) or $\mathcal{G}$ (ERM-MOL polynomial). Fig. 3 visualizes a toy classification problem with linear scalarization, and it compares the resulting decision boundaries, Pareto fronts, and excess $s$-trade-off across the different approaches.

Specifically, consider the data from Fig. 3a: the support of task 2 is completely contained within the support of task 1, and in particular, there is an area where the labels of the two tasks disagree (the bottom left "striped rectangle"). Both tasks are optimally solvable by linear models, but trying to solve both tasks at the same time is impossible, even in $\mathcal{F}_{\mathrm{all}}$. Meanwhile, better trade-offs still become available using, e.g., polynomial features.

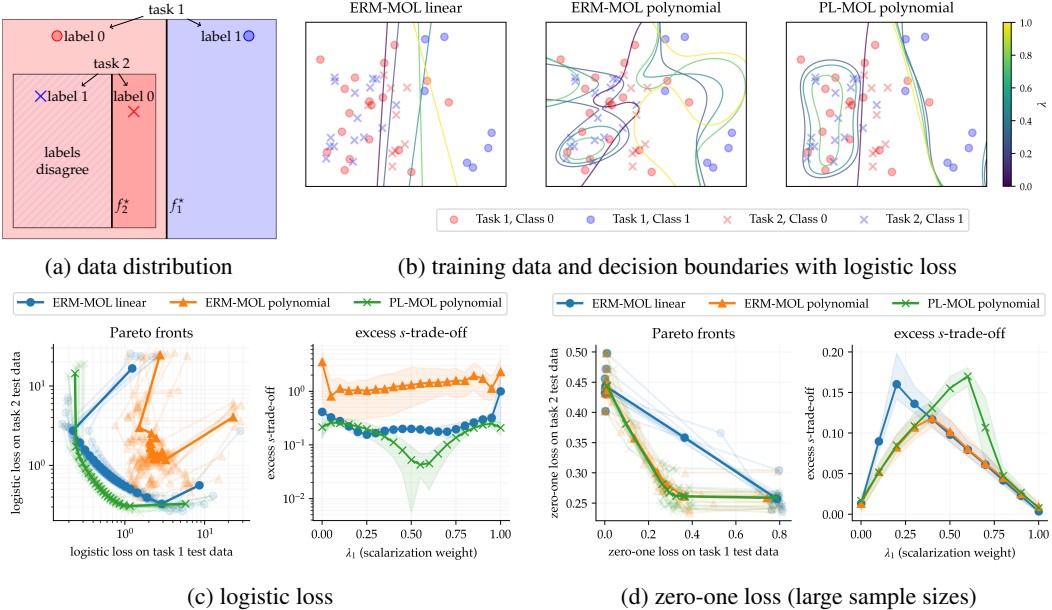

(a) data distribution

(b) training data and decision boundaries with logistic loss

(c) logistic loss

(d) zero-one loss (large sample sizes)

Figure 3: Learning trade-offs in the classification problem visualized in Fig. 3a. We show 1) supervised linear models, 2) supervised polynomial kernels, and 3) the mixture through the semi-supervised PL-MOL algorithm. (b) The training data and decision boundaries of the three methods, with a score threshold of $1/2$, for varying trade-off parameters $\lambda_1$. (c) The Pareto fronts for logistic loss and the $s$-trade-off as a function of the parameter $\lambda_1$ in the linear scalarization. (d) The Pareto fronts for zero-one loss and the $s$-trade-off as a function of the parameter $\lambda_1$. We repeat the experiment 10 times and show corresponding interquartile ranges.

We sample $n_1 = n_2 = 25$ data points uniformly from the regions in Fig. 3a and, in Figs. 3b and 3c, label them according to the linear logistic model $Y^k | X^k = x \sim \text{Ber}(f_k^\star(x))$, that is, with noise and in accordance with Eq. (3). Again, we run the three different algorithms on the logistic loss using linear scalarization: ERM-MOL (Algorithm 2) on the function class $\mathcal{H}$ of linear models, ERM-MOL on the function class $\mathcal{G}$ of linear models on polynomial features up to degree 5, and PL-MOL (Algorithm 1) using $\mathcal{H}$ in the first stage for all tasks, and $\mathcal{G}$ in the second stage with an additional number of $N_1 = N_2 = 300$ unlabeled data points. PL-MOL fits linear models to the labeled data and uses these to predict (soft) pseudo-labels for the unlabeled data, resulting in Fig. 4. Some resulting decision boundaries of each method are shown in Fig. 3b, and the Pareto fronts (on the test data) as well as excess $s$-trade-offs are shown in Fig. 3c.

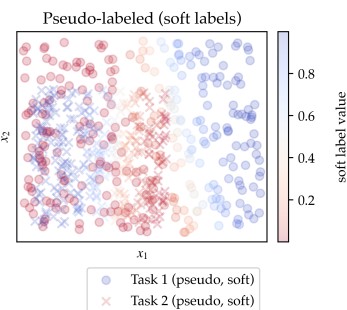

Figure 4: Pseudo-labeled data using PL-MOL with the logistic loss.

The expected bias-variance trade-off arises here. In this case, the individual tasks can be perfectly solved over the family of linear classifiers $\mathcal{H}$. However, ERM-MOL over $\mathcal{H}$ necessarily fails to find good trade-offs, as this model class is insufficiently expressive for the multi-objective learning problem—it has large bias. On the other hand, the ERM-MOL over $\mathcal{G}$ yields large estimation error, since there is not enough labeled data to solve for trade-offs over the much larger family of polynomial classifiers—the learned trade-offs have high variance. In contrast, the PL-MOL algorithm reduces this variance using only additional unlabeled data.

In this experiment, we can also corroborate the importance of the loss function. Fig. 3d shows that PL-MOL can be inconsistent when the losses are not Bregman divergences. While the Pareto front found by PL-MOL dominates the other methods, it incorrectly weighs the different objectives per linear scalarization, resulting in a sub-optimal excess $s$-trade-off. To amplify this effect, in Fig. 3d

we generate the labels in task 1 without any noise, and in task 2 according to this model:

$$Y^{(2)}|X^{(2)} = x \sim \begin{cases} 0 & x \text{ is in the "red region" of task 2,} \\ \text{Ber}(0.65) & x \text{ is in the "blue region" of task 2,} \end{cases}$$

where recall the different regions from Fig. 3a. Merely changing the loss to the zero-one loss then breaks Algorithm 1. Specifically, in Fig. 3d, we show how the same algorithms perform in a large sample regime ($n_1 = n_2 = 400$). PL-MOL does not attain the best-possible trade-off within $\mathcal{G}$, even when it recovers the Pareto front of $\mathcal{G}$.

## C.2 $\ell_2$-constrained linear regression

We now discuss another example where the localization can yield much tighter results than Theorem 1. To that end, we consider the following problem of constrained linear regression with squared loss.

Let $\mathcal{X} = \mathcal{B}_2^d$, $\mathcal{Y} = [-1, 1]$, and $\ell_k$ be the squared loss. For $R \in [0, 1]$, define the hypothesis spaces $\mathcal{H} = \{h(x) = \langle x, w \rangle : \|w\|_2 \le R\}$ and $\mathcal{G} = \{g(x) = \langle x, w \rangle : \|w\|_2 \le 1\}$. We consider distributions that satisfy $\mathbb{E}[Y^k|X^k = x] = \langle w_k^\star, x \rangle$, that is, a (possibly heteroscedastic) zero-mean noise model

$$Y^k = \langle w_k^\star, X^k \rangle + \xi^k \qquad \text{with} \qquad \forall x \in \mathcal{X} : \mathbb{E}[\xi^k|X^k = x] = 0,$$

where $f_k^\star = \langle w_k^\star, \cdot \rangle \in \mathcal{H} \subset \mathcal{F}_{\text{all}}$ for all $k$. Suppose that the covariance matrices of $X^k$ have smallest eigenvalue bounded from below by $\kappa \in [R, 1]$ (which is easily satisfied). Theorem 2 then yields the following corollary, proven in Appendix C.3.

**Corollary C.2** ($\ell_2$-constrained linear regression). *In the setting described above, the output of Algorithm 1 satisfies (S-MOL) with probability 0.99 and $\varepsilon_s = s(\varepsilon_1, \ldots, \varepsilon_K)$ for all $s \in \mathcal{S}_{\text{lin}}$ (Eq. (5)), where*

$$\varepsilon_k \lesssim \min\left\{\frac{1}{\kappa n_k}, \frac{2R}{\sqrt{n_k}}\right\} + \min\left\{\frac{1}{\kappa N_k}, \frac{2}{\sqrt{N_k}}\right\}.$$

Here $1/n_k$ and $1/N_k$ are the localized rates, where Theorem 1 would yield $1/\sqrt{n_k}$ and $1/\sqrt{N_k}$ instead. Notice that if $\mathcal{H}$ is very small (i.e., $R < 1/(2\kappa n_k)$), then the first term is small due to the smaller complexity of $\mathcal{H}$, while the second term may only become small due to larger unlabeled sample size $N_k$.

## C.3 Proofs for the Examples

In this section we provide the proofs of Corollaries 1, C.1 and C.2.

*Proof of Corollary 1.* We verify the assumptions of Theorem 2: Eq. (3) holds by definition of the data generating model. Assumption 1 and the smoothness from Assumption 2 hold, because the square loss $\ell$ is 2-Lipschitz and 1-bounded on $\mathcal{Y} = [0, 1]$, and induced by $\phi(y) = y^2$ which is 1-strongly convex, and $\max\{\phi'', \phi'''\} \le 2$. The other parts of Assumption 2 holds because the function classes $\mathcal{G}$ and $\mathcal{H}$ are convex, and the strong convexity holds with $\gamma_s = 1$: For every $s \equiv s_{\boldsymbol{\lambda}}^{\text{lin}} \in \mathcal{S}_{\text{lin}}$ a quick calculation shows that

$$d_s(g; \boldsymbol{h}) = \int_0^1 (g(x) - h_1(x))^2 \lambda_1 p_1(x) + (g(x) - h_2(x))^2 \lambda_2 p_2(x) \, dx$$

$$\|g\|_s^2 = \int_0^1 g^2(x)(\lambda_1 p_1(x) + \lambda_2 p_2(x)) \, dx$$

which implies that

$$d_s(g; \boldsymbol{h}) - \|g\|_s^2 = \int_0^1 \left((g(x) - h_1(x))^2 - g^2(x)\right)\lambda_1 p_1(x) + \left((g(x) - h_2(x))^2 - g^2(x)\right)\lambda_2 p_2(x) \, dx$$

and hence the strong convexity follows from the convexity of $g \mapsto (g - a)^2 - g^2$ for any $a \in \mathbb{R}$. Assumption 3 holds because the functions from Eq. (9) are assumed to be $L_{\mathcal{G}}$-Lipschitz and hence by Lemma 2 the global minimizers are contained in $\mathcal{G}$.

To apply Theorem 2, denote the space of $2L_{\mathcal{G}}$-Lipschitz functions $[0,1] \to [-1,1]$ as $\widetilde{\mathcal{G}}$, and note that for any function $g \in \mathcal{G}$ we have that $\mathcal{G} - g \subset \widetilde{\mathcal{G}}$. Hence, we can see that

$$\mathcal{G}_k(r; \boldsymbol{h}) = r\mathcal{B}_{\|\cdot\|_k} \cap \bigcup_{s \in \mathcal{S}} (\mathcal{G} - g_s^{\boldsymbol{h}}) \subset \left\{ g \in \widetilde{\mathcal{G}} : \|g\|_k \leq r \right\}.$$

Denote by $N(t, \mathcal{A}, \|\cdot\|)$ the covering number of a set $\mathcal{A}$ with norm $\|\cdot\|$ at radius $t > 0$, see e.g., [63, Chapter 5] for a definition. It is a standard fact [63, Example 5.10] that the metric entropy of $\{g \in \widetilde{\mathcal{G}} : \|g\|_k \leq r\}$ is bounded as

$$\forall 0 < t \leq r : \qquad \log N\left(t, \left\{ g \in \widetilde{\mathcal{G}} : \|g\|_k \leq r \right\}, \|\cdot\|_k\right) \leq \frac{8L_{\mathcal{G}}}{t}.$$

Hence, using standard bounds with Dudley's entropy integral [22], we can bound the Rademacher complexity of this function class by

$$\mathfrak{R}_{N_k}^k\left(\left\{ g \in \widetilde{\mathcal{G}} : \|g\|_k \leq r \right\}\right) \lesssim \frac{1}{\sqrt{N_k}} \int_0^r \sqrt{\log N\left(t, \left\{ g \in \widetilde{\mathcal{G}} : \|g\|_k \leq r \right\}, \|\cdot\|_k\right)} \, dt$$

$$\lesssim \frac{1}{\sqrt{N_k}} \int_0^r (L_{\mathcal{G}})^{1/2} t^{-1/2} \, dt$$

$$\lesssim \sqrt{\frac{L_{\mathcal{G}} r}{N_k}}$$

and, similarly for $\mathcal{H}_k(r)$ we get that

$$\mathfrak{R}_{n_k}^k\left(\mathcal{H}_k(r)\right) \lesssim \sqrt{\frac{L_{\mathcal{H}} r}{n_k}}.$$

Solving the corresponding inequalities $r^2 \geq \sqrt{\frac{L_{\mathcal{H}} r}{n_k}}$ and $r^2 \geq \sqrt{\frac{L_{\mathcal{G}} r}{N_k}}$ yields

$$\mathfrak{l}_k^2 \lesssim L_{\mathcal{H}}^{2/3} n_k^{-2/3} \qquad \text{and} \qquad \bar{\mathfrak{u}}_k^2 \lesssim L_{\mathcal{G}}^{2/3} N_k^{-2/3}.$$

Plugging this into Eq. (13) from Theorem 2 and noting that 1) for any fixed confidence $1 - \delta$ (such as 0.99) the confidence term goes to zero faster than the main terms, and 2) the constants $C_k$ are universal constants in this example, yields the result. $\qquad\square$

*Proof of Corollary C.1.* We apply Theorem 1 to the setting. First, note that $f_k^\star(x) = \mathbb{E}\left[Y^k | X^k = x\right] = \sigma(\langle x, w_k^\star \rangle)$ is contained in $\mathcal{H}$, so that Eq. (3) holds. Also note that the loss $\ell(y, \hat{y}) = -(y \log(\hat{y}) + (1 - y) \log(1 - \hat{y}))$ is a Bregman loss (Definition 4) induced by the potential $\phi(y) = y \log y + (1 - y) \log(1 - y)$.

Moreover, for all $g \in \mathcal{G}$ we have that $g(\mathcal{X}) \subset [\sigma(-1), \sigma(1)]$, since for all $w \in \mathcal{B}_1^p$ and $\Phi(x) \in \mathcal{B}_\infty^p$ we have that $|\langle w, \Phi(x) \rangle| \leq \|w\|_1 \|\Phi(x)\|_\infty \leq 1$.

We can then check Assumption 1 (using the remark that only the range of $\mathcal{G}$ needs to be considered):

1. Because $\frac{d^2}{dy^2}\phi(y) = 1/(y(1-y)) \geq 4$ for all $y \in [0,1]$, we have that $\phi$ is 4-strongly convex.

2. Making use of the fact that the range of functions in $\mathcal{G}$ lies in $[\sigma(-1), \sigma(1)]$, we get that $\ell$ is $L$-Lipschitz in both arguments with $L = \frac{3}{2\sigma(-1)\sigma(1)}$. To see that, employ Lemma E.1 with $\text{diam}_{|\cdot|}(\mathcal{Y}) = 1$ and $\frac{d^2}{dy^2}\phi(y) \leq \frac{1}{\sigma(-1)\sigma(1)}$ on the range of $\mathcal{G}$.

3. Similarly, because the range of functions in $\mathcal{G}$ lies in $[\sigma(-1), \sigma(1)]$, the loss is bounded by $\ell \leq B = -\log(\sigma(-1))$.

Hence, we may apply Theorem 1. Standard bounds on the Rademacher complexities yield

$$\mathfrak{R}_{n_k}^k(\mathcal{H}) \leq \frac{1}{4}\sqrt{\frac{2 \log 2d}{n_k}} \quad \text{and} \quad \mathfrak{R}_{N_k}^k(\mathcal{G}) \leq \frac{1}{4}\sqrt{\frac{2 \log 2p}{N_k}}.$$

This can be proven using Lipschitz contraction with respect to the sigmoid (which is $1/4$-Lipschitz continuous). For both bounds, there exist distributions so that the bound is tight. Plugging this into Theorem 1 yields (for some fixed high probability, such as 0.99)

$$\varepsilon_k \lesssim \left(\frac{\log p}{N_k}\right)^{1/2} + \left(\frac{\log K}{N_k}\right)^{1/2} + \sqrt{\left(\frac{\log d}{n_k}\right)^{1/2} + \left(\frac{\log K}{n_k}\right)^{1/2}}$$

$$\lesssim \left(\frac{\log dK}{n_k}\right)^{1/4} + \left(\frac{\log pK}{N_k}\right)^{1/2}$$

where the last inequality holds if $\max\left\{\frac{\log p}{N_k}, \frac{\log K}{N_k}, \frac{\log d}{n_k}, \frac{\log K}{n_k}\right\} \le 1$, which we assumed. $\qquad\square$

*Proof of Corollary C.2.* Denote $\Sigma_k = \mathbb{E}\left[X^k(X^k)^\top\right]$, $s \equiv s_{\boldsymbol{\lambda}}^{\mathrm{lin}}$ and $g_w = \langle\cdot, w\rangle \in \mathcal{G}$, so that for any $w, w' \in \mathbb{R}^d$

$$\|g_w - g_{w'}\|_k^2 = \mathbb{E}\left[\left(g_w(X^k) - g_{w'}(X^k)\right)^2\right] = \mathbb{E}\left[\left(\langle w - w', X^k\rangle\right)^2\right] = (w - w')^\top \Sigma_k (w - w'),$$

and by an identical argument $d_k(g_w; g_{w'}) = (w - w')^\top \Sigma_k (w - w') = \|g_w - g_{w'}\|_k^2$. It follows that

$$d_s(g_w; \boldsymbol{f}^\star) = \sum_{k=1}^K \lambda_k \left((w - w_k^\star)^\top \Sigma_k (w - w_k^\star)\right)$$

$$= (w - w_{\boldsymbol{\lambda}})^\top \left(\sum_{k=1}^K \lambda_k \Sigma_k\right)(w - w_{\boldsymbol{\lambda}}) + d_s(g_{w_{\boldsymbol{\lambda}}}; \boldsymbol{f}^\star) \qquad (21)$$

where we defined the minimizers

$$g_s \equiv g_{w_{\boldsymbol{\lambda}}} \quad \text{with} \quad w_{\boldsymbol{\lambda}} = \underset{\|w\|_2 \le 1}{\arg\min}\, d_s(g_w; \boldsymbol{f}^\star) = \left(\sum_{k=1}^K \lambda_k \Sigma_k\right)^{-1}\left(\sum_{k=1}^K \lambda_k \Sigma_k w_k^\star\right).$$

This holds because the *unconstrained* minimizer coincides with the constrained one, ensured by the bounded norms $\|w_k^\star\|_2 \le R \le \kappa$ and bounded smallest eigenvalue $\mu_{\min}(\Sigma_k) \ge \kappa$—note that because $\|X^k\|_2 \le 1$ we have that $\mu_{\max}(\Sigma_k) \le 1$—which implies

$$\left\|\left(\sum_{k=1}^K \lambda_k \Sigma_k\right)^{-1}\left(\sum_{k=1}^K \lambda_k \Sigma_k w_k\right)\right\|_2 \le \frac{\sum_{k=1}^K \lambda_k \mu_{\max}(\Sigma_k)\|w_k^\star\|_2}{\sum_{k=1}^K \lambda_k \mu_{\min}(\Sigma_k)} \le \frac{R}{\kappa} \le 1.$$

We verify the assumptions of Theorem 2: Eq. (3) holds by definition of the data generating model. Assumption 1 and the smoothness from Assumption 2 hold, because $\ell$ is 4-Lipschitz and 4-bounded on $\mathcal{Y} = [-1, 1]$, and induced by $\phi(y) = y^2$ which is 1-strongly convex, and $\max\{\phi'', \phi'''\} \le 2$. The convexity of $d_s(g; \boldsymbol{h}) - \|g\|_s^2$ in Assumption 2 holds with constants $\gamma_s = 1$ by inspecting Eq. (21), and $\mathcal{G}$ and $\mathcal{H}$ are clearly convex.

We now bound the critical radius $r_R := \inf\left\{r \ge 0 : r^2 \ge \mathfrak{R}_n^k(\mathcal{F}_R(r))\right\}$ of the following function class $\mathcal{F}_R(r) := \left\{\langle\cdot, w\rangle : \|w\|_2 \le 2R, w^\top \Sigma_k w \le r^2\right\}$. Note that because $\mu_{\max}(\Sigma_k^{-1}) = 1/\mu_{\min}(\Sigma_k) \le 1/\kappa$ and $X_i^k \in \mathcal{B}_2^d$, it holds that

$$\mathbb{E}\left\|\sum_{i=1}^n \sigma_i X_i^k\right\|_{\Sigma_k^{-1}}^2 = \mathbb{E}\sum_{i,j=1}^n \sigma_i \sigma_j (X_i^k)^\top \Sigma_k^{-1} X_j^k = \mathbb{E}\sum_{i=1}^n (X_i^k)^\top \Sigma_k^{-1} X_i^k \le \frac{n}{\kappa},$$

and thus we get by Jensen's inequality that for any $R, r \ge 0$

$$\mathfrak{R}_n^k(\mathcal{F}_R(r)) \le \frac{1}{n}\mathbb{E}\left[\sup_{w^\top \Sigma_k w \le r^2}\left\langle w, \sum_{i=1}^n \sigma_i X_i^k\right\rangle\right] \le \frac{r}{n}\mathbb{E}\left\|\sum_{i=1}^n \sigma_i X_i^k\right\|_{\Sigma_k^{-1}} \le \frac{r}{\sqrt{\kappa n}}.$$

and, by standard Rademacher complexity bounds (applying Cauchy-Schwartz and Jensen's inequality),

$$\mathfrak{R}_n^k(\mathcal{F}_R(r)) \leq \frac{1}{n}\mathbb{E}\left[\sup_{\|w\|_2 \leq 2R}\left\langle w, \sum_{i=1}^n \sigma_i X_i^k\right\rangle\right] \leq \frac{2R}{\sqrt{n}}.$$

Hence, we can solve $r^2 \geq r/\sqrt{\kappa n}$ and $r^2 \geq 2R/\sqrt{n}$ to get, taking the minimum of the two,

$$r_R^2 \leq \min\left\{\frac{1}{\kappa n}, \frac{2R}{\sqrt{n}}\right\}.$$

We are now ready to apply Theorem 2. Noting that

$$\mathcal{H}_k(r) = r\mathcal{B}_{\|\cdot\|_k} \cap (\mathcal{H}_k - f_k^\star) \subset \left\{\langle\cdot, w\rangle : \|w\|_2 \leq 2R, w^\top \Sigma_k w \leq r^2\right\} = \mathcal{F}_R(r)$$

and that the set $\mathcal{G}_k(r; \boldsymbol{f}^\star)$ is included in $\mathcal{F}_1(r)$;

$$\begin{aligned}
\mathcal{G}_k(r; \boldsymbol{f}^\star) &= r\mathcal{B}_{\|\cdot\|_k} \cap \bigcup_{s \in \mathcal{S}}(\mathcal{G} - g_s)\\
&= \left\{\langle\cdot, w\rangle : w^\top \Sigma_k w \leq r^2\right\} \cap \left\{\langle\cdot, w - w_{\boldsymbol{\lambda}}\rangle : \|w\|_2 \leq 1, \boldsymbol{\lambda} \in \Delta^{K-1}\right\}\\
&\subset \left\{\langle\cdot, w\rangle : w^\top \Sigma_k w \leq r^2, \|w\|_2 \leq 2\right\} = \mathcal{F}_1(r),
\end{aligned}$$

we can apply the previous bound on the critical radius and get that

$$\mathfrak{l}_k^2 \leq \min\left\{\frac{1}{\kappa n_k}, \frac{2R}{\sqrt{n_k}}\right\} \qquad \text{and} \qquad \bar{\mathfrak{u}}_k^2 \leq \min\left\{\frac{1}{\kappa N_k}, \frac{2}{\sqrt{N_k}}\right\}.$$

Plugging this into Theorem 2 yields the result. $\qquad\square$

# D   Proofs of main results

## D.1   Proof of Proposition 1

To prove a sample complexity lower bound, we show a reduction from a statistical estimation problem to the semi-supervised multi-objective binary classification problem.

We start by constructing a statistical estimation problem, defining a family of distributions parametrized by the set of Boolean vectors $\boldsymbol{\sigma} \in \{0,1\}^d$. We aim to use samples from a distribution to estimate its associated parameter; the distributions will be designed so that any estimator given insufficiently many samples will fail to estimate the underlying parameter well for some $\boldsymbol{\sigma}$. Then, we show that any learner that solves the multi-objective learning problem ($\mathcal{S}$-MOL) with $\varepsilon_s = \varepsilon$ and $\delta \geq 5/6$ can be used to solve the parametric estimation problem, implying a sample complexity lower bound for $\mathcal{S}$-MOL. For convenience, we reproduce the PAC version of $\mathcal{S}$-MOL here:

$$\forall (P^1, \ldots, P^K) \in \mathcal{P}^K, \qquad \mathbb{P}\left(\forall s \in \mathcal{S}_{\text{lin}}, \quad \mathcal{T}_s(\widehat{g}_s) - \inf_{g \in \mathcal{G}}\mathcal{T}_s(g) \leq \varepsilon\right) \geq 5/6, \tag{22}$$

where $\mathcal{P}$ is the set of all distributions over $\mathcal{X} \times \mathcal{Y}$.

Let's consider the $K = 2$ case first. We show that $n \geq d/1024\varepsilon^2$ samples are necessary.

**Defining the statistical estimation problem.**   Let $\mathcal{X}_0 := \{x_1, \ldots, x_d\} \subset \mathcal{X}$ be a set shattered by $\mathcal{G}$. For each $\boldsymbol{\sigma} \in \{0,1\}^d$, define the distributions $P_{\boldsymbol{\sigma}}^1$ and $P_{\boldsymbol{\sigma}}^2$ over $\mathcal{X} \times \mathcal{Y}$ where (i) the marginal distributions on $\mathcal{X}$ is uniform over the shattered set $\{x_1, \ldots, x_d\}$, and (ii) their conditional distributions on $\mathcal{Y} = \{0,1\}$ given $x_i$ are Bernoulli distributions. Let $c = 4\varepsilon$ and define:

$$P_{\boldsymbol{\sigma}}^1 = \frac{1}{d}\sum_{i \in [d]}\delta_{x_i} \otimes \text{Ber}\left(\frac{1}{2} + c\sigma_i\right) \qquad \text{and} \qquad P_{\boldsymbol{\sigma}}^2 = \frac{1}{d}\sum_{i \in [d]}\delta_{x_i} \otimes \text{Ber}\left(\frac{1}{2} - c(1 - \sigma_i)\right).$$

Fix a sample size $n \in \mathbb{N}$. Define the family $\mathcal{Q} = \{\mathcal{Q}_{\boldsymbol{\sigma}} : \boldsymbol{\sigma} \in \{0,1\}^d\}$, where:

$$\mathcal{Q}_{\boldsymbol{\sigma}} = \left(P_{\boldsymbol{\sigma}}^1 \otimes P_{\boldsymbol{\sigma}}^2\right)^{\otimes n}.$$

Let $Z_{\boldsymbol{\sigma}} \sim Q_{\boldsymbol{\sigma}}$ consist of $n$ i.i.d. draws from $P_{\boldsymbol{\sigma}}^1$ and $P_{\boldsymbol{\sigma}}^2$ each. The statistical estimation problem will be to construct an estimator $\widehat{\boldsymbol{\sigma}}(Z_{\boldsymbol{\sigma}})$ for $\boldsymbol{\sigma}$ that recovers at least $3/4$ of the coordinates of $\boldsymbol{\sigma}$:

$$\max_{\boldsymbol{\sigma}} \mathbb{P}\left( \frac{\|\widehat{\boldsymbol{\sigma}}(Z_{\boldsymbol{\sigma}}) - \boldsymbol{\sigma}\|_1}{d} \leq \frac{1}{4} \right) \geq \frac{5}{6}. \tag{23}$$

**Reduction to multi-objective learning.** Suppose that a learner can solve the $\mathcal{S}$-MOL problem (22) for $K = 2$ using at most $n$ samples. The reduction from estimating $\boldsymbol{\sigma}$ is as follows:

1. Given any instance $\boldsymbol{\sigma} \in \{0,1\}^d$ of the above statistical estimation problem, have the learner solve the MOL problem over $(P_{\boldsymbol{\sigma}}^1, P_{\boldsymbol{\sigma}}^2)$ and linear scalarization $\mathcal{S}_{\text{lin}}$ using data $Z_{\boldsymbol{\sigma}} \sim Q_{\boldsymbol{\sigma}}$ and zero-one loss.

2. Query the learner for the solution to the linear scalarization $s_{1/2} \equiv s_{\boldsymbol{\lambda}}^{\text{lin}}$ with weights $\boldsymbol{\lambda} = (\frac{1}{2}, \frac{1}{2})$. Denote this solution by $\widehat{g}_{s_{1/2}}(\,\cdot\,; Z_{\boldsymbol{\sigma}}) \in \mathcal{G}$ and construct the estimator $\widehat{\boldsymbol{\sigma}}_{\text{MOL}}$ for $\boldsymbol{\sigma}$ as:

$$\widehat{\boldsymbol{\sigma}}_{\text{MOL}}(Z_{\boldsymbol{\sigma}})_i = \widehat{g}_{s_{1/2}}(x_i; Z_{\boldsymbol{\sigma}}).$$

*Correctness of reduction.* Before proving correctness, we make a few observations:

1. For $P_{\boldsymbol{\sigma}}^1$, the conditional label distribution associated to $x_i \in \mathcal{X}_0$ is either biased toward 1 or uniform over $\{0,1\}$. In either case, under the zero-one loss, the label 1 is Bayes optimal, and so the constant function $f_{1,\boldsymbol{\sigma}}^\star \equiv 1$ is a Bayes-optimal classifier for $P_{\boldsymbol{\sigma}}^1$. Likewise, $f_{2,\boldsymbol{\sigma}}^\star \equiv 0$ is Bayes optimal for $P_{\boldsymbol{\sigma}}^2$.

2. A function $g : \mathcal{X} \to \mathcal{Y}$ only incurs excess risk from an instance $x_i$ drawn from $P_{\boldsymbol{\sigma}}^1$ when $\sigma_i = 1$ and $g(x_i) = 0$. Similarly, it accumulates excess risk from instances $x_i$ from $P_{\boldsymbol{\sigma}}^2$ when $\sigma_i = 0$ and $g(x_i) = 1$. The total excess risks of $g$ is given by:

$$\mathcal{E}_1(g) = \frac{2c}{d} \sum_{i \in [d]} \sigma_i \mathbf{1}\{g(x_i) = 0\} \qquad \text{and} \qquad \mathcal{E}_2(g) = \frac{2c}{d} \sum_{i \in [d]} (1 - \sigma_i) \mathbf{1}\{g(x_i) = 1\}.$$

For the linear scalarization $s_{1/2}$, we have:

$$\mathcal{T}_{s_{1/2}}(g) = \frac{1}{2}\Big(\mathcal{E}_1(g) + \mathcal{E}_2(g)\Big) = \frac{c}{d} \sum_{i \in [d]} \mathbf{1}\{g(x_i) \neq \sigma_i\}.$$

3. Since $\mathcal{G}$ shatters $\mathcal{X}_0$, it contains a function $g_{\boldsymbol{\sigma}}$ that satisfies $g_{\boldsymbol{\sigma}}(x_i) = \sigma_i$ for all $i \in [d]$. Thus:

$$\inf_{g \in \mathcal{G}} \mathcal{T}_{s_{1/2}}(g) = \mathcal{T}_{s_{1/2}}(g_{\boldsymbol{\sigma}}) = 0.$$

4. Given $g : \mathcal{X} \to \mathcal{Y}$, define the Boolean vector $\boldsymbol{\sigma}_g \in \{0,1\}^d$ by $\sigma_{g,i} = g(x_i)$. Then, by our choice of $c$, the excess $s_{1/2}$-trade-off of $g$ is related to the Hamming distance between $\boldsymbol{\sigma}_g$ and $\boldsymbol{\sigma}$:

$$\mathcal{T}_{s_{1/2}}(g) - \inf_{g \in \mathcal{G}} \mathcal{T}_{s_{1/2}}(g) \leq \varepsilon \quad \Longleftrightarrow \quad \frac{\|\boldsymbol{\sigma}_g - \boldsymbol{\sigma}\|_1}{d} \leq \frac{1}{4}, \tag{24}$$

since $\mathcal{T}_{s_{1/2}}(g) = \frac{c}{d} \cdot \|\boldsymbol{\sigma}_g - \boldsymbol{\sigma}\|_1$.

This last point implies the correctness of the reduction. That is, if a learner can solve (22), then we can use it to construct $\widehat{\boldsymbol{\sigma}}_{\text{MOL}}$ that achieves (23). In particular, (24) shows that:

$$\mathbb{P}\left( \mathcal{T}_{s_{1/2}}\big(\widehat{g}_{s_{1/2}}(\cdot; Z_{\boldsymbol{\sigma}})\big) - \inf_{g \in \mathcal{G}} \mathcal{T}_{s_{1/2}}(g) \leq \varepsilon \right) = \mathbb{P}\left( \frac{\|\widehat{\boldsymbol{\sigma}}_{\text{MOL}}(Z_{\boldsymbol{\sigma}}) - \boldsymbol{\sigma}\|_1}{d} \leq \frac{1}{4} \right).$$

We now show that this statistical estimation problem requires at least $n \geq /1024\varepsilon^2$ samples across $P_{\boldsymbol{\sigma}}^1$ or $P_{\boldsymbol{\sigma}}^2$. This holds for any estimator including those that knows that $f_{1,\boldsymbol{\sigma}}^\star$ and $f_{2,\boldsymbol{\sigma}}^\star$ are Bayes-optimal classifiers and that the marginal distribution over $\mathcal{X}$ for both $P_{\boldsymbol{\sigma}}^1$ and $P_{\boldsymbol{\sigma}}^2$ are uniform over the shattered set. In particular, the lower bound applies to the semi-supervised MOL learner, which is given access to these Bayes-optimal classifiers and marginal distributions over $\mathcal{X}$.

**Minimax lower bound.** We now show that if $n < d/1024\varepsilon^2$, the following bound holds:

$$\max_{\widehat{\sigma}} \min_{\sigma} \mathbb{P}\left(\frac{\|\widehat{\sigma}(Z) - \sigma\|_1}{d} \leq \frac{1}{4}\right) < \frac{5}{6},$$

where $\widehat{\sigma} : (\mathcal{X} \times \mathcal{Y} \times \mathcal{X} \times \mathcal{Y})^n \to \{0,1\}^d$ ranges over all estimators using $n$ samples from $P_{\sigma}^1$ and $P_{\sigma}^2$ each.

For every $\widehat{\sigma}$ and $\sigma$ it follows from Markov's inequality that

$$
\begin{aligned}
\mathbb{P}\left(\frac{\|\widehat{\sigma}(Z_{\sigma}) - \sigma\|_1}{d} \leq \frac{1}{4}\right) &= \mathbb{P}\left(1 - \frac{\|\widehat{\sigma}(Z) - \sigma\|_1}{d} \geq \frac{3}{4}\right) \qquad (25) \\
&\leq \frac{4}{3}\left(1 - \mathbb{E}\left[\frac{\|\widehat{\sigma}(Z_{\sigma}) - \sigma\|_1}{d}\right]\right) \\
&= \frac{4}{3} - \frac{4}{3d} \cdot \mathbb{E}\left[\|\widehat{\sigma}(Z_{\sigma}) - \sigma\|_1\right].
\end{aligned}
$$

We lower bound $\max_{\sigma} \mathbb{E}\left[\|\widehat{\sigma}(Z_{\sigma}) - \sigma\|_1\right]$ for any estimator $\widehat{\sigma}$ using Assouad's lemma:

**Lemma D.1** (Assouad's lemma, [68]). *Let $d \geq 1$ be an integer and let $\mathcal{Q} = \{Q_{\sigma} : \sigma \in \{0,1\}^d\}$ contain $2^d$ probability measures. Given $\sigma, \sigma' \in \{0,1\}^d$, write $\sigma \sim \sigma'$ if they differ only in one coordinate. Let $\widehat{\sigma}$ be any estimator. Then*

$$\max_{\sigma} \mathbb{E}_{Z \sim Q_{\sigma}}\left[\|\widehat{\sigma}(Z) - \sigma\|_1\right] \geq \frac{d}{2} \cdot \min\left\{1 - \sqrt{\frac{1}{2}\mathrm{KL}(Q_{\sigma} \| Q_{\sigma'})} : \sigma \sim \sigma'\right\},$$

*where $\mathrm{KL}(\cdot \| \cdot)$ measures the Kullback-Leibler divergence between two distributions.*

When $\sigma$ and $\sigma'$ differ only in one coordinate, the KL divergence between $Q_{\sigma}$ and $Q_{\sigma'}$ is bounded:

$$
\begin{aligned}
\mathrm{KL}(Q_{\sigma} \| Q_{\sigma'}) &= n \cdot \mathrm{KL}(P_{\sigma}^1 \| P_{\sigma'}^1) + n \cdot \mathrm{KL}(P_{\sigma}^2 \| P_{\sigma'}^2) \\
&= n\left(\frac{1}{d}\sum_{i \in [d]} \mathrm{KL}\left(\frac{1}{2} + c\sigma_i \,\Big\|\, \frac{1}{2} + c\sigma_i'\right)\right) \\
&\quad + n\left(\frac{1}{d}\sum_{i \in [d]} \mathrm{KL}\left(\frac{1}{2} + c(1 - \sigma_i) \,\Big\|\, \frac{1}{2} + c(1 - \sigma_i')\right)\right) \leq \frac{8nc^2}{d},
\end{aligned}
$$

where the last inequality holds when $c = 4\varepsilon \leq 1/3$ by Lemma D.2. Indeed, we've assumed $\varepsilon < 1/12$.

By Assouad's lemma (Lemma D.1) and the above bound on the KL divergence, in the worst-case setting, any algorithm using $n$ samples will have expected error at least

$$\max_{\sigma} \mathbb{E}_{Z \sim Q_{\sigma}}\left[\|\widehat{\sigma}(Z) - \sigma\|_1\right] \geq \frac{d}{2} \cdot \left(1 - \sqrt{\frac{4nc^2}{d}}\right).$$

Plugging into Equation (25), we finally obtain:

$$\max_{\widehat{\sigma}} \min_{\sigma} \mathbb{P}\left(\frac{\|\widehat{\sigma} - \sigma\|_1}{d} \leq \frac{1}{4}\right) \leq \frac{2}{3} + \frac{2}{3}\sqrt{\frac{4nc^2}{d}} < \frac{5}{6},$$

where the last inequality holds whenever $\sqrt{4nc^2/d} \leq 1/4$, which holds when $n < d/1024\varepsilon^2$.

**Generalization to all $K > 1$.** The MOL problem with $K$ tasks is at least as hard as $M = \lfloor K/2 \rfloor$ separate MOL problems each with two tasks. This leads to a total sample complexity lower bound $M \cdot d/1024\varepsilon^2$. We obtain the lower bound $CKd/\varepsilon^2$ in the statement of the result by setting $C = 1/3072$ and using Lemma D.3, which shows that $\lfloor K/2 \rfloor \geq K/3$ for all $K > 1$.

More explicitly, we can reduce $M$ separate copies of the statistical estimation problem for $\sigma_1, \ldots, \sigma_M$ into a single MOL problem over the distributions:

$$\left(\ldots, P_{\sigma_k}^{2k-1}, P_{\sigma_k}^{2k}, \ldots\right),$$

where $k = 1, \ldots, M$. Define $s_{1/2}^k$ to be the linear scalarization that equally divides all weight across the $2k - 1$ and $2k$ components:

$$s_{1/2}^k(\boldsymbol{v}) = \frac{v_{2k-1} + v_{2k}}{2}.$$

Then, an estimator $\widehat{\boldsymbol{\sigma}}_k$ for $\boldsymbol{\sigma}_k$ can be obtained from the by defining as before:

$$\widehat{\boldsymbol{\sigma}}_{k,i} = \widehat{g}_{s_{1/2}^k}(x_i).$$

The analysis from the $K = 2$ setting now holds for each $k = 1, \ldots, M$. This implies that at least $d/1024\varepsilon^2$ samples must be drawn across each pair of the $2k - 1$ and $2k$th distributions. This concludes the proof.

**Lemma D.2** (KL-divergence bound, e.g. [16])**.** *Let $x \in (-1/3, 1/3)$. Then:*

$$\mathrm{KL}\Big(\frac{1}{2} \,\|\, \frac{1}{2} + x\Big) \leq 4x^2 \qquad and \qquad \mathrm{KL}\Big(\frac{1}{2} + x \,\|\, \frac{1}{2}\Big) \leq 4x^2.$$

*Proof.* By a direct computation, we have that whenever $4x^2 \leq 1/2$, which is satisfied when $x \in [-1/3, 1/3]$,

$$\mathrm{KL}\Big(\frac{1}{2} \,\|\, \frac{1}{2} + x\Big) = \frac{1}{2} \log \frac{1}{1 - 4x^2} \leq 4x^2,$$

where the last inequality holds from the fact that $\frac{1}{2} \log \frac{1}{1-z} \leq z$ whenever $z \in [0, 1/2]$.

For the second inequality, we show that the function $\phi(x) = \mathrm{KL}(1/2 + x \| 1/2)$ is $L$-smooth on $(-1/3, 1/3)$ where $L \leq 8$ and has zero derivative at $x = 0$. This implies that it is upper bounded by $\frac{L}{2}x^2$. In particular, the first and second derivatives are:

$$\phi'(x) = \log(1 + 2x) + \log(1 - 2x) \qquad and \qquad \phi''(x) = \frac{4}{1 - 4x^2},$$

so that $\phi'' \leq 8$ whenever $x^2 \leq 1/9$. $\qquad\square$

**Lemma D.3.** *Let $K > 1$ be a natural number. Then, $\lfloor K/2 \rfloor \geq K/3$.*

*Proof.* There are two cases:

- When $K$ is even, then $\lfloor K/2 \rfloor = K/2 \geq K/3$.

- When $K$ is odd, then $\lfloor K/2 \rfloor = (K - 1)/2 \geq K/3$, where the last inequality is equivalent to $3(K - 1) \geq 2K$, which is further equivalent to $K \geq 3$.

$\qquad\square$

### D.2 Proof of Lemma 1

Let $\ell_k$ be a Bregman loss associated with the potential $\phi_k$. The first part is proven in [7, Theorem 1]: for any $Y^k$ such that $\mathbb{E}[Y^k]$ and $\mathbb{E}[\phi_k(Y^k)]$ are finite, it holds that

$$f_k^\star = \underset{f \in \mathcal{F}_{\mathrm{all}}}{\arg\min} \, \mathbb{E}[\ell_k(Y^k, f(X^k))] = \mathbb{E}\left[Y^k | X^k = \cdot\right].$$

Then, by definition of Bregman divergences, we have the following generalized Pythagorean identity [8, Equation (26)]

$$\ell_k(y, x) = \ell_k(y, z) + \ell_k(z, x) - \langle y - z, \nabla\phi_k(x) - \nabla\phi_k(z) \rangle,$$

so that by the tower property (see also [7, Equation (1)])

$$\begin{aligned}
\mathcal{R}_k(f) &= \mathbb{E}[\ell_k(Y^k, f(X^k))] \\
&= \underbrace{\mathbb{E}\left[\ell_k(Y^k, \mathbb{E}\left[Y^k | X^k\right])\right]}_{\mathcal{R}_k(f_k^\star)} + \underbrace{\mathbb{E}\left[\ell_k(\mathbb{E}\left[Y^k | X^k\right], f(X^k))\right]}_{d_k(f; f_k^\star)} \\
&\quad - \underbrace{\left\langle \mathbb{E}\left[Y^k - \mathbb{E}\left[Y^k | X^k\right]\right], \nabla\phi_k(f(X^k)) - \nabla\phi_k(\mathbb{E}\left[Y^k | X^k\right]) \right\rangle}_{=0}.
\end{aligned}$$

Rearranging yields that $\mathcal{E}_k(f) = \mathcal{R}_k(f) - \mathcal{R}_k(f_k^\star) = d_k(f; f_k^\star)$, which is the second claim.

### D.3 Proof of Theorem 1

The proof of Theorem 1 relies on the following lemma on estimating the excess risk functionals $\mathcal{E}_k$ with the risk discrepancies $d_k(f; \widehat{h}_k)$ under Assumption 1.

**Lemma D.4** (Excess risk functional estimation). *Suppose that Assumption 1 holds and that a function $\widehat{h}_k$ achieves excess risk $\mathcal{E}_k(\widehat{h}_k)$. Let $c_k = L_k\sqrt{2/\mu_k}$. Then, the risk discrepancy functional $d_k(\cdot; \widehat{h}_k)$ defined in Equation (7) approximates $\mathcal{E}_k(\cdot)$ uniformly on $\mathcal{F}_{\mathrm{all}}$, that is,*

$$\sup_{f \in \mathcal{F}_{\mathrm{all}}} \big|d_k(f; \widehat{h}_k) - \mathcal{E}_k(f)\big| \le c_k\sqrt{\mathcal{E}_k(\widehat{h}_k)}.$$

*Proof.* Recall that $f_k^\star \in \mathcal{H}_k$ is the minimizer of $\mathcal{E}_k$ over $\mathcal{F}_{\mathrm{all}}$. Then for all $f \in \mathcal{F}_{\mathrm{all}}$:

$$
\begin{aligned}
\big|d_k(f; \widehat{h}_k) - \mathcal{E}_k(f)\big| &= \big|d_k(f; \widehat{h}_k) - d_k(f; f_k^\star)\big| & \text{(Lemma 1)}\\
&= \big|\mathbb{E}\big[\ell_k\big(\widehat{h}_k(X^k), f(X^k)\big) - \ell_k\big(f_k^\star(X^k), f(X^k)\big)\big]\big|\\
&\le L_k\mathbb{E}\big\|\widehat{h}_k(X^k) - f_k^\star(X^k)\big\|_2 & \text{(Lipschitz continuity from Assumption 1)}\\
&\le L_k\sqrt{\mathbb{E}\big\|\widehat{h}_k(X^k) - f_k^\star(X^k)\big\|_2^2} & \text{(Jensen's inequality)}\\
&\le L_k\sqrt{\frac{2}{\mu_k} \cdot \mathbb{E}\big[\ell_k\big(f_k^\star(X^k), \widehat{h}_k(X^k)\big)\big]}\\
& & \text{(strong convexity from Assumption 1)}\\
&= c_k\sqrt{\mathcal{E}_k(\widehat{h}_k)}, & \text{(Lemma 1)}
\end{aligned}
$$

which is the claim. $\qquad\square$

*Proof of Theorem 1.* For $k \in [K]$, let $\widehat{h}_k$ be the empirical risk minimizer obtained in Line 2 of Algorithm 1 for the $k$th objective. Let us recall that we use the empirical risk discrepancy $\widehat{d}_k(\cdot; \widehat{h}_k)$ as an estimate for the excess risk $\mathcal{E}_k(\cdot) = d_k(\cdot; f_k^\star)$, following the properties of Bregman losses in Lemma 1. We now prove the theorem assuming that the following claim holds.

**Claim.** With probability at least $1 - \delta$, each estimate $\widehat{d}_k(\cdot; \widehat{h}_k)$ approximates the population excess risk functional $\mathcal{E}_k$ up to error $\varepsilon_k/2$:

$$\forall k \in [K], \qquad \sup_{g \in \mathcal{G}} \big|\widehat{d}_k(g; \widehat{h}_k) - \underbrace{d_k(g; f_k^\star)}_{=\mathcal{E}_k(g)}\big| \le \varepsilon_k/2, \tag{26}$$

where $\varepsilon_k$ is bounded as in Eq. (10).

Then, for any scalarization $s$ that satisfies the reverse triangle inequality, the $s$-trade-off $\mathcal{T}_s$ is also well-approximated by empirical scalarized discrepancy $\widehat{d}_s(\cdot; \widehat{h})$. In particular, we obtain

$$
\begin{aligned}
\sup_{g \in \mathcal{G}} \big|\widehat{d}_s(g; \widehat{h}) - \mathcal{T}_s(g)\big| &= \sup_{g \in \mathcal{G}} \big|s\big(\widehat{d}_1(g; \widehat{h}_1), \ldots, \widehat{d}_K(g; \widehat{h}_K)\big) - s\big(d_1(g; f_k^\star), \ldots, d_K(g; f_k^\star)\big)\big|\\
&\le \sup_{g \in \mathcal{G}} s\big(\big|\widehat{d}_1(g; \widehat{h}_1) - d_1(g; f_k^\star)\big|, \ldots, \big|\widehat{d}_K(g; \widehat{h}_K) - d_K(g; f_k^\star)\big|\big)\\
&\le s\big(\varepsilon_1/2, \ldots, \varepsilon_K/2\big), \tag{27}
\end{aligned}
$$

where the first inequality used the reverse triangle inequality of $s$, and the second inequality used the monotonicity of $s$ and Eq. (26). In particular, this allows us to bound the excess $s$-trade-off of $\widehat{g}_s$, the minimizer of the empirical scalarized discrepancy in $\mathcal{G}$ obtained in Line 5 of Algorithm 1, as follows:

$$
\begin{aligned}
\mathcal{T}_s(\widehat{g}_s) - \mathcal{T}_s(g_s) &= \underbrace{\mathcal{T}_s(\widehat{g}_s) - \widehat{d}_s(\widehat{g}_s; \widehat{h})}_{(a)} + \underbrace{\widehat{d}_s(\widehat{g}_s; \widehat{h}) - \widehat{d}_s(g_s; \widehat{h})}_{(b)} + \underbrace{\widehat{d}_s(g_s; \widehat{h}) - \mathcal{T}_s(g_s)}_{(c)}\\
&\le s\big(\varepsilon_1/2, \ldots, \varepsilon_K/2\big) + s\big(\varepsilon_1/2, \ldots, \varepsilon_K/2\big)\\
&= s\big(\varepsilon_1, \ldots, \varepsilon_K\big),
\end{aligned}
$$

where the (a) and (c) terms both contribute at most $s(\varepsilon_1/2, \ldots, \varepsilon_k/2)$ error from Eq. (27), while the (b) term is non-positive, since $\widehat{g}_s$ minimizes the empirical scalarized discrepancy $\widehat{d}_s(\cdot; \boldsymbol{h})$. The last equality follows from positive homogeneity of the scalarization. Then, the result follows for all such scalarizations *simultaneously*. It remains to prove Eq. (26).

**Proof of claim.**    Recall that $\widehat{d}_k(g; \widehat{h}_k)$ is an empirical estimator of $d_k(g; \widehat{h}_k)$ based on the unlabeled samples:

$$\widehat{d}_k(g; \widehat{h}_k) = \frac{1}{N_k} \sum_{i=1}^{N_k} \ell_k\big(\widehat{h}_k(\widetilde{X}_i^k), g(\widetilde{X}_i^k)\big) \quad \text{and} \quad d_k(g; \widehat{h}_k) = \mathbb{E}\left[\ell_k\big(\widehat{h}_k(X^k), g(X^k)\big)\right],$$

where these were defined in Eq. (7). Moreover, $d_k(g; \widehat{h}_k)$ itself is an estimator of the excess risk functional $\mathcal{E}_k(g) = d_k(g; f_k^\star)$, where $f_k^\star$ is the Bayes optimal regression function (Lemma 1). Thus, we have the decomposition:

$$\left|\widehat{d}_k(g; \widehat{h}_k) - \mathcal{E}_k(g)\right| = \underbrace{\left|\widehat{d}_k(g; \widehat{h}_k) - d_k(g; \widehat{h}_k)\right|}_{T_{a,k}} + \underbrace{\left|d_k(g; \widehat{h}_k) - \mathcal{E}_k(g)\right|}_{T_{b,k}}. \tag{28}$$

We can bound $T_{a,k}$ and $T_{b,k}$ separately:

*(a)* For each $k \in [K]$, we condition on the labeled samples (i.e., on $\widehat{h}_k$) and employ a standard Rademacher bound on the function class:

$$\ell_{\widehat{h}_k} \circ \mathcal{G} = \left\{x \mapsto \ell_k\big(\widehat{h}_k(x), g(x)\big) : g \in \mathcal{G}\right\}.$$

With probability at least $1 - \delta/(2K)$,

$$T_{a,k} \leq \sup_{g \in \mathcal{G}} \left|\widehat{d}_k(g; \widehat{h}_k) - d_k(g; \widehat{h}_k)\right| \leq 2\mathfrak{R}_{N_k}^k(\ell_{\widehat{h}_k} \circ \mathcal{G}) + B_k \left(\frac{2\log(2K/\delta)}{N_k}\right)^{1/2}$$

$$\leq 6L_k \mathfrak{R}_{N_k}^k(\mathcal{G}) + B_k \left(\frac{2\log(2K/\delta)}{N_k}\right)^{1/2}, \tag{29}$$

where the first inequality applies symmetrization (Lemma E.5) and the bounded difference inequality (Lemma E.2), and the second inequality follows by contraction (Lemma E.4).

*(b)* For each $k \in [K]$, we apply Lemma D.4 to bound $T_{b,k}$ in terms of the excess risk of $\widehat{h}_k$, which is a minimizer of the empirical risk $\widehat{\mathcal{R}}_k(\cdot)$ defined in Eq. (2):

$$\widehat{\mathcal{R}}_k(h) = \frac{1}{n_k} \sum_{i=1}^{n_k} \ell_k\big(Y_i^k, h(X_i^k)\big).$$

In order to use the lemma, we need to show that the excess risk of $\widehat{h}_k$ is indeed upper bounded by $\varepsilon_k$. First, observe that the excess risk can be upper bounded as follows

$$\begin{aligned}
\mathcal{E}_k(\widehat{h}_k) &= \mathcal{R}_k(\widehat{h}_k) - \mathcal{R}_k(f_k^\star) \\
&= \mathcal{R}_k(\widehat{h}_k) - \widehat{\mathcal{R}}_k(\widehat{h}_k) + \widehat{\mathcal{R}}_k(\widehat{h}_k) - \widehat{\mathcal{R}}_k(f_k^\star) + \widehat{\mathcal{R}}_k(f_k^\star) - \mathcal{R}_k(f_k^\star) \\
&\leq 2 \sup_{h \in \mathcal{H}_k} \left|\widehat{\mathcal{R}}_k(h) - \mathcal{R}_k(h)\right|.
\end{aligned}$$

Again by symmetrization (Lemma E.5), bounded difference (Lemma E.2), and contraction (Lemma E.4) for the function class $\ell_k \circ \mathcal{H}_k := \{(x, y) \mapsto \ell_k(y, h(x)) : h \in \mathcal{H}_k\}$, we obtain that with probability at least $1 - \delta/(2K)$,

$$2 \sup_{h \in \mathcal{H}_k} \left|\widehat{\mathcal{R}}_k(h) - \mathcal{R}_k(h)\right| \leq 4\mathfrak{R}_{n_k}^k(\ell_k \circ \mathcal{H}_k) + 2B_k \left(\frac{2\log(2K/\delta)}{n_k}\right)^{1/2}$$

$$\leq 12L_k \mathfrak{R}_{n_k}^k(\mathcal{H}_k) + 2B_k \left(\frac{2\log(2K/\delta)}{n_k}\right)^{1/2}.$$

And so, by Lemma D.4, we obtain that with probability at least $1 - \delta/(2K)$,

$$T_{b,k} \le c_k \sqrt{12 L_k \mathfrak{R}_{n_k}^k \left(\mathcal{H}_k\right) + 2B_k \left(\frac{2\log(2K/\delta)}{n_k}\right)^{1/2}}, \tag{30}$$

where $c_k = L_k \sqrt{2/\mu_k}$.

The claim in Eq. (26) follows by a union bound. By combining Equations (28) to (30), we obtain that with probability at least $1 - \delta$, for all $k \in [K]$:

$$\sup_{g \in \mathcal{G}} \left| \widehat{d}_k(g; \widehat{h}_k) - \mathcal{E}_k(g) \right| \le \varepsilon_k/2,$$

where we can set $\varepsilon_k$ as:

$$\varepsilon_k/2 = 4 L_k \mathfrak{R}_{N_k}^k(\mathcal{G}) + B_k \left(\frac{2\log(2K/\delta)}{N_k}\right)^{1/2} + c_k \sqrt{12 L_k \mathfrak{R}_{n_k}\left(\mathcal{H}_k\right) + 2B_k \left(\frac{2\log(2K/\delta)}{n_k}\right)^{1/2}}.$$

This concludes the proof of Theorem 1. $\qquad \square$

## D.4  Proof of Theorem 2

In this section, we provide the proof of Theorem 2, but leave some of the technical details to auxiliary results that we prove after the main proof. See also Fig. 5 for a visualization of the proof.

### D.4.1  Preliminaries

Recall that $s$ is a linear scalarization. We begin by introducing the notation $\mu_k = P_X^k$ as well as the mixture distribution $\mu_s = s(\mu_1, \ldots, \mu_K)$. Recall the definitions of the (semi-)Hilbert norms

$$\|f\|_k := \sqrt{\mathbb{E}_{X^k \sim \mu_k} \|f(X^k)\|_2^2} \qquad \text{and} \qquad \|f\|_s := \sqrt{s\left(\|f\|_1^2, \ldots, \|f\|_K^2\right)}.$$

which correspond to the inner products (denoting $\langle \cdot, \cdot \rangle$ the inner product on $\mathbb{R}^q$)

$$\langle f, f' \rangle_k := \int_{\mathcal{X}} \langle f, f' \rangle \, d\mu_k \qquad \text{and} \qquad \langle \cdot, \cdot \rangle_s := s((\langle \cdot, \cdot \rangle_k)_{k \in [K]}).$$

We first verify that these are indeed (semi-)Hilbert norms and inner products.

**Lemma D.5.** *The functions $\langle \cdot, \cdot \rangle_k, \langle \cdot, \cdot \rangle_s$ and $\|\cdot\|_k, \|\cdot\|_s$ defined above are the inner products and norms of the $L^2(\mu_k)$ and $L^2(\mu_s)$-Bochner spaces of functions $\mathcal{X} \to (\mathbb{R}^q, \|\cdot\|_2)$.*

See [30, Definition 1.2.15] for a definition. We prove Lemma D.5 in Appendix D.4.3 and use it throughout without explicitly referring to it. Note that we have implicitly assumed that $\mathcal{F}_{\text{all}} \subseteq \bigcap_{k \in [K]} L^2(\mu_k)$. In order to use first-order calculus throughout the proof, we derive the gradient and smoothness of the map $g \mapsto d_s(g; \boldsymbol{h})$ below. We also prove Lemma D.6 in Appendix D.4.3.

**Lemma D.6** (Gradients and smoothness). *For any $\boldsymbol{h} \in \mathcal{H}_1 \times \cdots \times \mathcal{H}_K$ and linear scalarization $s$, denote by $\nabla_g d_s(g; \boldsymbol{h}) : \mathcal{X} \to \mathbb{R}$ the gradient of the map $g \mapsto d_s(g; \boldsymbol{h})$ induced by the Fréchet derivatives on $L^2(\mu_s)$ and the inner product $\langle \cdot, \cdot \rangle_s$. Then it holds that[5]*

$$\nabla_g d_s(g; \boldsymbol{h}) : x \mapsto \sum_{k=1}^K \lambda_k \frac{d\mu_k}{d\mu_s}(x) \nabla^2 \phi_k(g(x))(g(x) - h_k(x))).$$

*Moreover, if Assumption 2 holds and we denote $D = \text{diam}_{\|\cdot\|_2}(\mathcal{Y})$, then the map $g \mapsto d_s(g; \boldsymbol{h})$ is $C^{\text{sm}} := \nu(1 + D)$-smooth in $\|\cdot\|_s$, that is, the gradient from above is $C^{\text{sm}}$-Lipschitz continuous in $g$ with respect to $\|\cdot\|_s$. If additionally Assumption 3 holds, then for $g_s = \arg\min_{g \in \mathcal{G}} d_s(g; \boldsymbol{f}^\star)$ and all $g \in \mathcal{G}$*

$$d_s(g; \boldsymbol{f}^\star) - d_s(g_s; \boldsymbol{f}^\star) \le \frac{C^{\text{sm}}}{2} \|g - g_s\|_s^2. \tag{31}$$

---

[5]Note that whenever $\lambda_k > 0$, the Radon-Nikodym derivative $d\mu_k/d\mu_s$ is well-defined.

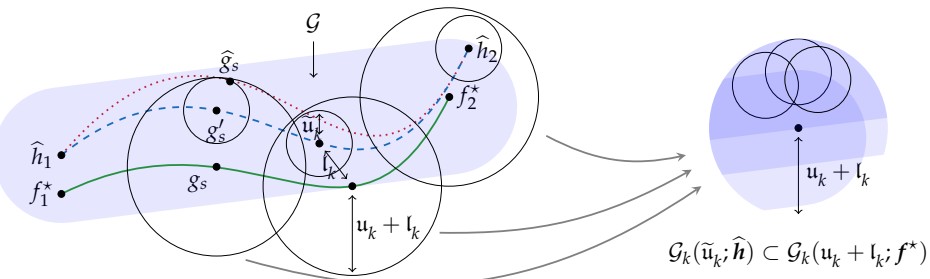

Figure 5: Informal visualization of the proof of Theorem 2. We first localize the set of estimators $\{\widehat{g}_s : s \in \mathcal{S}\}$ (dotted red line) around the *random* "helper" set $\{g'_s : s \in \mathcal{S}\}$ (dashed blue line) within the set $\mathcal{G}_k(\widetilde{\mathfrak{u}}_k; \widehat{h})$. We then expand a set $\mathcal{G}_k(\mathfrak{u}_k, \boldsymbol{f}^\star)$ centered at the "true" set $\{g_s : s \in \mathcal{S}\}$ (solid green line) to include the set $\mathcal{G}_k(\widetilde{\mathfrak{u}}_k; \widehat{h}) \subset \mathcal{G}_k(\mathfrak{u}_k + \mathfrak{l}_k; \boldsymbol{f}^\star)$ where $\mathfrak{l}_k$ bounds the maximal deviation of $g_s$ to $g'_s$. This way, we may bound the critical random critical radius $\mathfrak{u}$ of $\mathcal{G}_k(\widetilde{\mathfrak{u}}_k, \widehat{h})$ in terms of the deterministic critical radius $\mathfrak{u}$ of $\mathcal{G}_k(\mathfrak{u}; \boldsymbol{f}^\star)$ and $\mathfrak{l}_k$ as $\widetilde{\mathfrak{u}}_k^2 \lesssim \mathfrak{u}_k^2 + \mathfrak{l}_k^2$.

Lemma 2 is a direct consequence of the gradient characterization in Lemma D.6 together with Theorem 46 in [69]. Finally, we show that if Assumption 4 holds, the norms $\|\cdot\|_k$ and $\|\cdot\|_s$ are equivalent.

**Lemma D.7.** *Let $\mathcal{S} \subset \mathcal{S}_{\mathrm{lin}}$ be in the set of linear scalarizations* (5). *Then, for any $\eta \in [0, \infty)$*

$$\sup \left\{ \frac{\|f\|_k}{\|f\|_s} : k \in [K], s \in \mathcal{S}, f \in \mathcal{F}_{\mathrm{all}} \right\} \leq \eta \iff \forall k \in [K], s \in \mathcal{S} : \ \mathrm{ess} \sup \frac{d\mu_k}{d\mu_s} \leq \eta^2.$$

We also prove Lemma D.7 in Appendix D.4.3.

### D.4.2 Main proof of Theorem 2

Recall the empirical and population minimizers of the corresponding risk discrepancies from Eq. (7)

$$\forall s \in \mathcal{S} : \quad \widehat{g}_s \in \arg\min_{g \in \mathcal{G}} \widehat{d}_s(g; \widehat{h}) \quad \text{and} \quad g_s \in \arg\min_{g \in \mathcal{G}} d_s(g; \boldsymbol{f}^\star).$$

Our goal is to bound $\mathcal{T}_s(\widehat{g}_s) - \inf_{g \in \mathcal{G}} \mathcal{T}_s(g)$ simultaneously for all $s \in \mathcal{S}$. By Lemma 1, we have $\mathcal{T}_s(\widehat{g}_s) - \inf_{g \in \mathcal{G}} \mathcal{T}_s(g) = d_s(\widehat{g}_s; \boldsymbol{f}^\star) - d_s(g_s; \boldsymbol{f}^\star)$ so that we focus on bounding this expression.

The basic decomposition of our proof is a triangle inequality with a helper set of minimizers of the population risk discrepancy, defined with respect to pseudo-labeled data as

$$g'_s \in \arg\min_{g \in \mathcal{G}} d_s(g; \widehat{h}).$$

Specifically, by the smoothness from Lemma D.6, we can bound the excess trade-off as

$$d_s(\widehat{g}_s; \boldsymbol{f}^\star) - d_s(g_s; \boldsymbol{f}^\star) \leq \frac{C^{\mathrm{sm}}}{2} \|\widehat{g}_s - g_s\|_s^2 \leq C^{\mathrm{sm}} \Big( \underbrace{\|\widehat{g}_s - g'_s\|_s^2}_{=:T_s^{\mathrm{un}}} + \underbrace{\|g'_s - g_s\|_s^2}_{=:T_s^{\mathrm{lab}}} \Big). \tag{32}$$

Here $T_s^{\mathrm{lab}}$ quantifies the error from having a finite amount of labeled data to estimate $f_k^\star$ with $\widehat{h}_k$ and how that error propagates to $g'_s$, and $T_s^{\mathrm{un}}$ quantifies how close to $g'_s$ we can get with the finite amount of unlabeled data. Our goal will be to bound the terms $T_s^{\mathrm{lab}}$ and $T_s^{\mathrm{un}}$ using localization, simultaneously for all $s \in \mathcal{S}$. For the general proof technique of localization, we take inspiration from the approaches outlined in [63, 56, 35, 9, 11, 36].

We proceed in three main steps. See also Fig. 5.

1. To bound $T_s^{\mathrm{lab}}$, we first use standard localization bounds for the ERMs in each task separately, using uniform bound on the local sets $\mathcal{H}_k(r) = (\mathcal{H}_k - f_k^\star) \cap r\mathcal{B}_{\|\cdot\|_k}$ from Eq. (11). We then show how their errors translate to $g'_s$ through a deterministic stability argument.

2. To bound $T_s^{\mathrm{un}}$, we condition on $\widehat{\boldsymbol{h}}$ and simultaneously localize around the (random) functions $g_s'$ for all $s \in \mathcal{S}$, resulting in a uniform learning bound on local sets

$$\mathcal{G}_k(r; \widehat{\boldsymbol{h}}) = r\mathcal{B}_{\|\cdot\|_k} \cap \bigcup_{s \in \mathcal{S}} (\mathcal{G} - g_s') \tag{33}$$

that are "centered" at the helper set $\{g_s' : s \in \mathcal{S}\}$.

3. The resulting bound on $T_s^{\mathrm{un}}$ from the previous step is random, because $g_s'$ depends on $\widehat{\boldsymbol{h}}$, so we need to further bound it. We prove two ways of doing that, so that the bound takes the minimum of the two: the critical radius of $\mathcal{G}_k(r, \boldsymbol{f}^\star) = r\mathcal{B}_{\|\cdot\|_k} \cap \bigcup_{s \in \mathcal{S}}(\mathcal{G} - g_s)$ from Eq. (11) together with the bound on $T_s^{\mathrm{lab}}$, or a worst-case bound taking the supremum over such $\{g_s' : s \in \mathcal{S}\}$.

See also Fig. 5 for a visualization of the corresponding sets.

Throughout, we heavily use the following monotonicity property of the Rademacher complexity, analogous to the usual localization proofs. The proof can be found in Appendix D.4.4.

**Lemma D.8.** *Consider the sets from Eqs. (11) and (33). Under Assumption 2, the functions*

$$r \mapsto \frac{\mathfrak{R}_{n_k}^k\left(\mathcal{H}_k(r)\right)}{r} \qquad \text{and} \qquad r \mapsto \frac{\mathfrak{R}_{N_k}^k(\mathcal{G}_k(r; \boldsymbol{h}))}{r} \tag{34}$$

*are non-increasing on $(0, \infty)$ for all $\boldsymbol{h} \in \mathcal{H}_1 \times \cdots \times \mathcal{H}_K$.*

**Step 1: Localization for ERMs in $\mathcal{H}_k$ and bounding $T_s^{\mathrm{lab}}$.** In this step, we first bound the error of learning $f_k^\star$ with the ERMs $\widehat{h}_k$ (or, in fact, any other estimator that satisfies the basic inequality $\widehat{\mathcal{R}}_k(\widehat{h}_k) \le \widehat{\mathcal{R}}_k(f_k^\star)$). Recall the definition $\mathcal{H}_k(r) = (\mathcal{H}_k - f_k^\star) \cap r\mathcal{B}_{\|\cdot\|_k}$ from Eq. (11), and the corresponding critical radii $\mathfrak{l}_k = \inf\left\{r \ge 0 : r^2 \ge \mathfrak{R}_{n_k}^k(\mathcal{H}_k(r))\right\}$. Using the non-increasing property from Lemma D.8, we can summarize the bound in the following Lemma.

**Lemma D.9.** *Under Assumptions 1 and 2, and if $\delta > 0$ is sufficiently small, we have that $\mathbb{P}(E_\delta^{\mathrm{lab}}) \ge 1 - \delta$, where we define the event*

$$E_\delta^{\mathrm{lab}} := \left\{\forall k \in [K] : \quad \left\|\widehat{h}_k - f_k^\star\right\|_k^2 \lesssim \frac{L_k^2}{\mu_k^2}\mathfrak{l}_k^2 + \left(\frac{B_k}{\mu_k} + \frac{L_k^2}{\mu_k^2}\right)\frac{\log(K/\delta)}{n_k} =: \zeta_k^2\right\}. \tag{35}$$

The proof of Lemma D.9 can be found in Appendix D.4.5, and it essentially follows the localization technique from [9]: We bound the suprema of the empirical process over $\mathcal{H}_k(r)$ using Talagrand's inequality (Lemma E.3) in terms of the Rademacher complexity and variance. Using Lemma D.8 and a peeling argument, we get the bound in terms of the critical radius.

Next, we show that the bound from Eq. (35) directly translates into a bound on the helper set $\{g_s' : s \in \mathcal{S}\}$ with respect to labels from $\widehat{\boldsymbol{h}}$ but known covariate distributions. To do so, we prove the following *stability* result. Effectively, it removes the square-root from Lemma D.4 that appears in Theorem 1.

**Proposition D.1** (Quadratic stability of minimizers). *Denote $g_s^{\boldsymbol{h}} = \arg\min_{g \in \mathcal{G}} d_s(g; \boldsymbol{h})$ and $C^{\mathrm{st}} := \nu^2/4\gamma^2$. Under Assumptions 1 and 2, we have for any $\boldsymbol{h}, \boldsymbol{h}' \in \mathcal{H}_1 \times \cdots \times \mathcal{H}_K$, any $s = s_{\boldsymbol{\lambda}}^{\mathrm{lin}} \in \mathcal{S}$, that*

$$\left\|g_s^{\boldsymbol{h}} - g_s^{\boldsymbol{h}'}\right\|_s^2 \le C^{\mathrm{st}} \sum_{k=1}^K \lambda_k \left\|h_k - h_k'\right\|_k^2.$$

We prove Proposition D.1 in Appendix D.4.6. Note that a *linear* bound would directly follow from Lipschitz continuity of the losses. However, this would yield much slower statistical rates than the stability argument from Proposition D.1. Recalling the definition of $\zeta_k$ from (35), we can now use Proposition D.1 with $g_s^{\widehat{\boldsymbol{h}}} = g_s'$, $g_s^{\boldsymbol{f}^\star} = g_s$, to conclude that on $E_\delta^{\mathrm{lab}}$, it holds that for all $s \in \mathcal{S}$,

$$T_s^{\mathrm{lab}} = \|g_s' - g_s\|_s^2 \lesssim C^{\mathrm{st}} \cdot s\left(\zeta_1^2, \ldots, \zeta_K^2\right). \tag{36}$$

Eq. (36) describes how well our estimators would approximate the true set $\{g_s : s \in \mathcal{S}\}$ if we had an infinite amount of unlabeled data. In that sense, this can be seen as an intermediate result in the ideal semi-supervised setting by combining Eqs. (32) and (36).

**Step 2: Localization along helper Pareto set in $\mathcal{G}$ to bound $T_s^{\mathrm{un}}$.** We now need to take into account the finite sample effect of having only $N_k$ unlabeled samples to estimate the risk discrepancies.

To perform localization around the helper set, we again rely on Talagrand's concentration inequality (Lemma E.3). The benefit of Talagrand's inequality in standard localization usually comes from the fact that it accounts for the *variance* of the losses when centered at *the ground truth*, which can usually be controlled by its radius of the local function class. We also used this in Step 1. Now, however, we need to *simultaneously* localize for all scalarizations $s \in \mathcal{S}$. Hence, recall $\mathcal{G}_k(r; \widehat{\boldsymbol{h}})$ from Eq. (33) where, intuitively, $r$ uniformly controls the deviations $\widehat{g}_s - g_s'$ for all $s \in \mathcal{S}$. To keep track of which $g_s'$ any $g \in \mathcal{G}_k(r; \widehat{\boldsymbol{h}})$ "belongs to", we also define the set

$$\mathcal{M}_k(r) = \left\{ (s, g) : s \in \mathcal{S}, g - g_s' \in \mathcal{G}_k(r, \widehat{\boldsymbol{h}}) \right\}. \tag{37}$$

Lifting the set $\mathcal{G}_k(r; \widehat{\boldsymbol{h}})$ to $\mathcal{S} \times \mathcal{G}$ is inspired by a similar trick from multi-objective optimization, where the Pareto set is often lifted to this larger space to obtain its manifold structure, cf. [53].

We apply Talagrand's concentration inequality on $\mathcal{M}_k(r)$ and use a localization argument, summarized in the following lemma. Define the radii $\widetilde{\mathfrak{u}}_k = \inf\left\{ r \geq 0 : r^2 \geq \mathfrak{R}_{N_k}^k(\mathcal{G}_k(r; \widehat{\boldsymbol{h}})) \right\}$, and note that these radii are deterministic with respect to the unlabeled data, but are *random* with respect to the labeled data through the ERMs, a point revisited in the next section. Recall that $\widehat{g}_s$ is the minimizer of $\widehat{d}_s(g; \widehat{\boldsymbol{h}})$ and $g_s'$ is the minimizer of $d_s(g; \widehat{\boldsymbol{h}})$ (Eq. (7)). The next proposition bounds $T_s^{\mathrm{un}} = \|\widehat{g}_s - g_s'\|_s^2$ (or, in fact, the deviation of any estimator satisfying the basic inequality $\widehat{d}_s(\widehat{g}_s; \widehat{\boldsymbol{h}}) \leq \widehat{d}_s(g_s'; \widehat{\boldsymbol{h}})$).

**Proposition D.2** (Localization along helper Pareto set). *Under Assumptions 1 and 2, and for sufficiently small $\delta > 0$, we have that $\mathbb{P}(E_\delta^{\mathrm{un}}) \geq 1 - \delta$, where we define the event*

$$E_\delta^{\mathrm{un}} := \left\{ \forall s = s_{\boldsymbol{\lambda}}^{\mathrm{lin}} \in \mathcal{S} : \quad \|\widehat{g}_s - g_s'\|_s^2 \lesssim \sum_{k=1}^{K} \lambda_k \left( \frac{L_k^2}{\gamma^2} \widetilde{\mathfrak{u}}_k^2 + \left( \frac{L_k^2}{\gamma^2} + \frac{B_k}{\gamma} \right) \frac{\log(2K/\delta)}{N_k} \right) \right\}. \tag{38}$$

The proof of Proposition D.2 can be found in Appendix D.4.7.

**Step 3: Bounding the random critical radii $\widetilde{\mathfrak{u}}_k$.** Recall that $\widetilde{\mathfrak{u}}_k$ is deterministic with respect to the unlabeled data, but random with respect to the labeled data. To make the bound fully deterministic, we prove two bounds, so that their minimum appears in Theorem 2.

*Option 1* is taking the trivial approach: recall from Eq. (11) that we define for the function $g_s^{\boldsymbol{h}} = \arg\min_{g \in \mathcal{G}} d_s(g; \boldsymbol{h})$ the set

$$\mathcal{G}_k(r; \boldsymbol{h}) := \bigcup_{s \in \mathcal{S}} (\mathcal{G} - g_s^{\boldsymbol{h}}) \cap r\mathcal{B}_{\|\cdot\|_k}.$$

Then the following *deterministic worst-case* localized radii

$$\bar{\mathfrak{u}}_k := \sup_{\boldsymbol{h} \in \mathcal{H}_1 \times \cdots \times \mathcal{H}_K} \inf\left\{ r \geq 0 : r^2 \geq \mathfrak{R}_{N_k}^k(\mathcal{G}_k(r; \boldsymbol{h})) \right\}$$

bounds $\widetilde{\mathfrak{u}}_k^2 \leq \bar{\mathfrak{u}}_k^2$ (and $\mathfrak{u}_k^2 \leq \bar{\mathfrak{u}}_k^2$) deterministically (i.e., also almost surely).

*Option 2* is more nuanced: If Assumption 4 holds, we can combine Eq. (36) with an expansion argument to bound $\widetilde{\mathfrak{u}}_k$ in terms of the $\mathfrak{l}_k$ and the $\mathfrak{u}_k$. To relate them, we employ the following key proposition.

**Proposition D.3** (Critical radius shift). *Let $\mathcal{G}$ be any class of functions that is convex (Assumption 2), and let $n \in \mathbb{N}$. Let $\|\cdot\|$ be any norm on $\mathcal{F}_{\mathrm{all}}$ and let $\mathcal{B} = \{f \in \mathcal{F}_{\mathrm{all}} : \|f\| \leq 1\}$ be its unit ball. Define*

$$\mathcal{G}(r) = \bigcup_{s \in \mathcal{S}} (\mathcal{G} - g_s) \cap r\mathcal{B} \qquad and \qquad \mathcal{G}'(r) = \bigcup_{s \in \mathcal{S}} (\mathcal{G} - g_s') \cap r\mathcal{B}$$

*as well as the critical radii (for $\mathfrak{R}_n$ defined w.r.t. an arbitrary distribution)*

$$\mathfrak{u} := \inf\left\{ r \geq 0 : \mathfrak{R}_n(\mathcal{G}(r)) \leq r^2 \right\} \qquad and \qquad \widetilde{\mathfrak{u}} := \inf\left\{ r \geq 0 : \mathfrak{R}_n(\mathcal{G}'(r)) \leq r^2 \right\}.$$

*Let $\Delta = \sup_{s \in \mathcal{S}} \|g_s - g_s'\|$. Then it holds that $\widetilde{\mathfrak{u}} \leq 5(\mathfrak{u} + \Delta)$.*

The proof of Proposition D.3 can be found in Appendix D.4.8. We can apply Proposition D.3 to our setting: recall the definitions of $\mathcal{G}'_k(r)$ from Eq. (33) and $\mathcal{G}_k(r)$ from Eq. (11), and the definitions $\widetilde{\mathfrak{u}}_k = \inf\left\{r \geq 0 : r^2 \geq \mathcal{G}_k(r; \widehat{\boldsymbol{h}})\right\}$ and $\mathfrak{u}_k = \inf\left\{r \geq 0 : r^2 \geq \mathfrak{R}^k_{N_k}(\mathcal{G}_k(r; \boldsymbol{f}^\star))\right\}$. From Assumption 4, Lemma D.7, and Eq. (36), we know that on $E^{\mathrm{lab}}_\delta$ from Eq. (35), for $\zeta^2_{\mathcal{S}} = \sup_{s \in \mathcal{S}} s\left(\zeta^2_1, \ldots, \zeta^2_K\right)$

$$\sup_{s \in \mathcal{S}} \|g'_s - g_s\|^2_k \leq \sup_{s \in \mathcal{S}} \eta^2 \|g'_s - g_s\|^2_s \lesssim \sup_{s \in \mathcal{S}} \eta^2 C^{\mathrm{st}} \cdot s\left(\zeta^2_1, \ldots, \zeta^2_K\right) \leq \eta^2 C^{\mathrm{st}} \cdot \zeta^2_{\mathcal{S}} =: \Delta^2.$$

Employing Proposition D.3 with this $\Delta$ yields $\widetilde{\mathfrak{u}}^2_k \lesssim \mathfrak{u}^2_k + \eta^2 C^{\mathrm{st}} \cdot \zeta^2_{\mathcal{S}}$.

We define

$$C^{\mathrm{add}} := \max_{k \in [K]} \left(\frac{B_k}{\mu_k} + \frac{L^2_k}{\mu^2_k}\right), \tag{39}$$

$\mathfrak{l}^2_{\mathcal{S}} := \sup_{s \in \mathcal{S}} s(\mathfrak{l}^2_1, \ldots, \mathfrak{l}^2_K)$, and $n_{\mathcal{S}} = 1/\sup_{s \in \mathcal{S}} s(1/n_1, \ldots, 1/n_K)$. We can bound

$$\widetilde{\mathfrak{u}}^2_k \lesssim \mathfrak{u}^2_k + \eta^2 C^{\mathrm{st}} \cdot \zeta^2_{\mathcal{S}}$$

$$= \mathfrak{u}^2_k + \eta^2 C^{\mathrm{st}} \sup_{s \in \mathcal{S}} s\left(\left(\frac{L^2_k}{\mu^2_k} \mathfrak{l}^2_k + \left(\frac{B_k}{\mu_k} + \frac{L^2_k}{\mu^2_k}\right) \frac{\log(4K/\delta)}{n_k}\right)^K_{k=1}\right)$$

$$\leq \mathfrak{u}^2_k + \eta^2 C^{\mathrm{st}} C^{\mathrm{add}} \left(\mathfrak{l}^2_{\mathcal{S}} + \frac{\log(4K/\delta)}{n_{\mathcal{S}}}\right)$$

$$\leq \eta^2 C^{\mathrm{st}} C^{\mathrm{add}} \left(\mathfrak{u}^2_k + \mathfrak{l}^2_{\mathcal{S}} + \frac{\log(4K/\delta)}{n_{\mathcal{S}}}\right) \tag{40}$$

Note that in general, either bound can be tighter. For practical purposes, it may be easier to bound $\bar{\mathfrak{u}}_k$ anyways, so the detour through $\mathfrak{u}_k$ may be unnecessary.

**Putting everything together.** From Eq. (32), we see that on $E^{\mathrm{lab}}_{\delta/2} \cap E^{\mathrm{un}}_{\delta/2}$, which holds with probability at least $1 - \delta$ by union bound, for all $s \in \mathcal{S}$, the excess $s$-trade-off $\mathcal{T}_s(\widehat{g}_s) - \inf_{g \in \mathcal{G}} \mathcal{T}_s(g)$ is bounded by

$$C^{\mathrm{sm}}\left(T^{\mathrm{un}}_s + T^{\mathrm{lab}}_s\right)$$

$$\lesssim C^{\mathrm{sm}} \sum_{k=1}^K \lambda_k \left(\frac{L^2_k}{\gamma^2} \widetilde{\mathfrak{u}}^2_k + \left(\frac{L^2_k}{\gamma^2} + \frac{B_k}{\gamma}\right) \frac{\log(4K/\delta)}{N_k} + C^{\mathrm{st}} \left(\frac{L^2_k}{\mu^2_k} \mathfrak{l}^2_k + \left(\frac{B_k}{\mu_k} + \frac{L^2_k}{\mu^2_k}\right) \frac{\log(4K/\delta)}{n_k}\right)\right)$$

$$\text{(from Eqs. (36) and (38))}$$

$$\leq \sum_{k=1}^K \lambda_k \underbrace{C^{\mathrm{sm}} C^{\mathrm{st}} \max\left\{\frac{L^2_k}{\gamma^2} + \frac{B_k}{\gamma}, \frac{B_k}{\mu_k} + \frac{L^2_k}{\mu^2_k}\right\}}_{=:C_k} \left(\widetilde{\mathfrak{u}}^2_k + \mathfrak{l}^2_k + \left(N^{-1}_k + n^{-1}_k\right) \log(4K/\delta)\right),$$

where $\lesssim$ only hides universal constants. From the two options of bounding $\widetilde{\mathfrak{u}}_k$ we obtain:

1. The first bound, valid without Assumption 4: Recalling $C^{\mathrm{sm}} = \nu(1 + D)$, $C^{\mathrm{st}} = \frac{\nu^2}{4\gamma^2}$

$$\mathcal{T}_s(\widehat{g}_s) - \inf_{g \in \mathcal{G}} \mathcal{T}_s(g) \lesssim \sum_{k=1}^K \lambda_k C_k \left(\bar{\mathfrak{u}}^2_k + \mathfrak{l}^2_k + \left(N^{-1}_k + n^{-1}_k\right) \log(4K/\delta)\right)$$

$$\text{where} \quad C_k \asymp \frac{\nu^3(1 + D)}{\gamma^2} \max\left\{\frac{L^2_k}{\gamma^2} + \frac{B_k}{\gamma}, \frac{B_k}{\mu_k} + \frac{L^2_k}{\mu^2_k}\right\}.$$

2. The second bound, valid under Assumption 4, by plugging in Eq. (40) and $C^{\mathrm{add}}$ from Eq. (39)

$$\mathcal{T}_s(\widehat{g}_s) - \inf_{g \in \mathcal{G}} \mathcal{T}_s(g) \lesssim \sum_{k=1}^K \lambda_k \widetilde{C}_k \left(\mathfrak{u}^2_k + \mathfrak{l}^2_{\mathcal{S}} + \left(N^{-1}_k + n^{-1}_{\mathcal{S}}\right) \log(4K/\delta)\right)$$

$$\text{where} \quad \widetilde{C}_k = C_k \cdot \eta^2 C^{\mathrm{st}} C^{\mathrm{add}} \asymp C_k \cdot \eta^2 \frac{\nu^2}{\gamma^2} \max_{k \in [K]} \left(\frac{B_k}{\mu_k} + \frac{L^2_k}{\mu^2_k}\right).$$

That concludes the proof of Theorem 2, with the proofs of the auxiliary results presented next.

### D.4.3 Proof of preliminary lemmata

*Proof of Lemma D.5.* The claim for $\langle \cdot, \cdot \rangle_k$, $\|\cdot\|_k$, $k \in [K]$ is true by definition, but also as a special case of the scalarized form: for any $s = s_\lambda^{\text{lin}} \in \mathcal{S}_{\text{lin}}$ and $f, f' \in L^2(\mu_s)$ we have

$$\langle f, f' \rangle_s = \sum_{k=1}^K \lambda_k \int \langle f, f' \rangle \, d\mu_k = \int \langle f, f' \rangle \, d\left(\sum_{k=1}^K \lambda_k \mu_k\right) = \int \langle f, f' \rangle \, d\mu_s$$

where $\langle \cdot, \cdot \rangle$ is the Euclidean inner product. This is exactly the inner product of the Bochner $L^2(\mu_s)$ space (e.g., [30]). Further, plugging in $f' = f$ we obtain directly that $\langle f, f \rangle_s = \|f\|_s^2$, verifying that the norm is induced by this inner product. $\qquad\square$

*Proof of Lemma D.6.* Recall that $\langle \cdot, \cdot \rangle_s$ denotes the inner product of the norm $\|\cdot\|_s^2 = \sum_{k=1}^K \lambda_k \|\cdot\|_k^2$. In this proof, we use *Fréchet derivatives* (denoted $D$) and the corresponding gradients $\nabla_g$ induced by the inner product $\langle \cdot, \cdot \rangle_s$. Background on Fréchet derivatives can be found in [2, 12]. From Lemma D.5 we know that $\|\cdot\|_s$ actually is the (semi-)Hilbert norm that corresponds to the Bochner $L^2(\mu_s)$ space with respect to the space $(\mathbb{R}^q, \|\cdot\|_2)$, where recall that $\mu_s$ denotes the mixture distribution

$$\mu_s = \sum_{k=1}^K \lambda_k \mu_k \qquad \text{where} \qquad \mu_k = P_X^k.$$

It is then easily shown that the gradient of $\sum_{k=1}^K \lambda_k \mathbb{E}\left[Q_k(g(X^k))\right]$ for any differentiable functions $Q_k : \mathbb{R}^q \supset \mathcal{Y} \to \mathbb{R}$ with $\sup_{y \in \mathcal{Y}} \|\nabla Q(y)\|_2 \le M < \infty$, induced by $\langle \cdot, \cdot \rangle_s$, is given by

$$\nabla_g \sum_{k=1}^K \lambda_k \mathbb{E}\left[Q_k(g(X^k))\right] : \mathcal{X} \to \mathbb{R}^q, \qquad x \mapsto \sum_{k=1}^K \lambda_k \frac{d\mu_k}{d\mu_s}(x) \nabla Q_k(g(x)).$$

Indeed, for any $f \in \mathcal{F}_{\text{all}} \subset L^2(\mu_s)$, we can write the Fréchet derivative as the limit

$$D\left(\sum_{k=1}^K \lambda_k \mathbb{E}\left[Q_k(g(X^k))\right]\right)[f] = \sum_{k=1}^K \lambda_k \lim_{\varepsilon \to 0} \frac{\mathbb{E}\left[Q_k(g(X^k) + \varepsilon f(X^k))\right] - \mathbb{E}\left[Q_k(g(X^k))\right]}{\varepsilon}$$

$$= \sum_{k=1}^K \lambda_k \mathbb{E}\left[\langle \nabla Q_k(g(X^k)), f(X^k) \rangle\right] \quad \text{(dominated convergence)}$$

$$= \sum_{k=1}^K \lambda_k \int \langle (\nabla Q_k(g(x)), f(x) \rangle \frac{d\mu_k}{d\mu_s}(x) d\mu_s(x)$$

$$= \int \left\langle \sum_{k=1}^K \lambda_k \frac{d\mu_k}{d\mu_s}(x) \nabla Q_k(g(x)), f(x) \right\rangle d\mu_s(x)$$

$$= \left\langle \sum_{k=1}^K \lambda_k \frac{d\mu_k}{d\mu_s}(\nabla Q_k \circ g), f \right\rangle_s$$

where we could use dominated convergence thanks to $\sup_{y \in \mathcal{Y}} \|\nabla Q(y)\|_2 \le M < \infty$. This implies the claimed form of the gradient.

Since $\ell_k$ is Lipschitz and differentiable, its gradient in $g$ is bounded. Further,

$$\nabla_g d_s(g; \boldsymbol{h}) = \nabla_g \sum_{k=1}^K \lambda_k d_k(g; h_k) = \nabla_g \sum_{k=1}^K \lambda_k \mathbb{E}\left[\ell_k(h_k(X^k), g(X^k))\right]$$

and the gradient of a Bregman divergence in its second argument is given by

$$\nabla_y D_\phi(x, y) = \nabla_y \left(\phi(x) - \phi(y) - \langle \nabla\phi(y), x - y \rangle\right)$$
$$= -\nabla\phi(y) - \nabla^2\phi(y)x + \nabla^2\phi(y)y + \nabla\phi(y)$$
$$= \nabla^2\phi(y)(y - x),$$

so that the previous derivations imply for the Bregman losses that

$$\nabla_g d_s(g; \boldsymbol{h}) : x \mapsto \sum_{k=1}^{K} \lambda_k \frac{d\mu_k}{d\mu_s}(x) \nabla^2 \phi_k(g(x))(g(x) - h_k(x))),$$

which is the first claim of the lemma.

Note that $\mu_s$-almost surely $\sum_{k=1}^{K} \lambda_k \frac{d\mu_k}{d\mu_s} = 1$. Hence, for every fixed $\boldsymbol{h}$, and $g, g'$,

$$\left\| \nabla_g d_s(g; \boldsymbol{h}) - \nabla_g d_s(g'; \boldsymbol{h}) \right\|_s^2$$

$$= \int \left\| \sum_{k=1}^{K} \lambda_k \frac{d\mu_k}{d\mu_s} \left[ \nabla^2 \phi_k(g)(h_k - g) - \nabla^2 \phi_k(g')(h_k - g') \right] \right\|_2^2 d\mu_s$$

$$\leq \int \sum_{k=1}^{K} \lambda_k \frac{d\mu_k}{d\mu_s} \left\| \nabla^2 \phi_k(g)(h_k - g) - \nabla^2 \phi_k(g')(h_k - g') \right\|_2^2 d\mu_s \qquad \text{(Jensen's inequality)}$$

$$\leq \int \sum_{k=1}^{K} \lambda_k \frac{d\mu_k}{d\mu_s} \left( \left\| \nabla^2 \phi_k(g)(g - g') \right\|_2 + \left\| (\nabla^2 \phi_k(g) - \nabla^2 \phi_k(g'))(h_k - g') \right\|_2 \right)^2 d\mu_s. \qquad (41)$$

To bound the first term, we use from Assumption 2 that the $\ell_2$-operator norm of $\nabla^2 \phi(g(x))$ is bounded by $\nu > 0$, so that

$$\left\| \nabla^2 \phi_k(g)(g - g') \right\|_2 \leq \nu \left\| g - g' \right\|_2.$$

To bound the second term, we use that $\|h_k - g'\|_2 \leq \mathrm{diam}_{\|\cdot\|_2}(\mathcal{Y}) =: D$, and so

$$\left\| (\nabla^2 \phi_k(g) - \nabla^2 \phi_k(g'))(h_k - g') \right\|_2 \leq D \left\| \nabla^2 \phi_k(g) - \nabla^2 \phi_k(g') \right\|_2,$$

which together with the smoothness from Assumption 2 implies

$$\left\| \nabla^2 \phi_k(g) - \nabla^2 \phi_k(g') \right\|_2 = \left\| \int_0^1 (\nabla^3 \phi_k(g + t(g' - g))(g - g')dt \right\|_2 \leq \nu \left\| g - g' \right\|_2$$

Plugging both into Eq. (41) yields

$$\left\| \nabla_g d_s(g; \boldsymbol{h}) - \nabla_g d_s(g'; \boldsymbol{h}) \right\|_s^2 \leq \int \sum_{k=1}^{K} \lambda_k \frac{d\mu_k}{d\mu_s} \nu^2 (1+D)^2 \left\| g - g' \right\|_2^2 d\mu_s = \nu^2 (1+D)^2 \left\| g - g' \right\|_s^2.$$

Hence, by equivalent characterizations of smoothness (e.g., [12, Corollary 18.14]) it follows that

$$d_s(g; \boldsymbol{h}) - d_s(g'; \boldsymbol{h}) - \langle \nabla_g d_s(g'; \boldsymbol{h}), g - g' \rangle_s \leq \frac{\nu(1+D)}{2} \left\| g - g' \right\|_s^2.$$

By Assumption 3, we know that the *global* minimizer of $f \mapsto d_s(f; \boldsymbol{f}^\star)$ is contained in $\mathcal{G}$. By definition, therefore $g_s$ coincides with it, and hence for $g' = g_s$ and $\boldsymbol{h} = \boldsymbol{f}^\star$, we know that $\langle \nabla d_s(g_s; \boldsymbol{f}^\star), g - g_s \rangle_s = 0$ and obtain the bound

$$d_s(g; \boldsymbol{f}^\star) - d_s(g_s; \boldsymbol{f}^\star) \leq \frac{\nu(1+D)}{2} \left\| g - g_s \right\|_s^2.$$

This concludes the proof. $\qquad\qquad\square$

*Proof of Lemma D.7.* Denote $\mu_k = P_X^k$ and $\mu_s = \sum_k \lambda_k \mu_k$. Recall the definition of the essential supremum of a function $f : \mathcal{X} \to \mathbb{R}$ (with respect to $\mu_s$):

$$\mathrm{ess\,sup}\, f = \inf \left\{ a \in \mathbb{R} : \mu_s(f^{-1}(a, \infty)) = 0 \right\}.$$

We start with "$\Leftarrow$": Since $\mu_s(\{x \in \mathcal{X} : d\mu_k/d\mu_s(x) \geq \eta^2\}) = 0$, for any $k \in [K], s \in \mathcal{S}$ and $f \in \mathcal{F}_{\mathrm{all}}$,

$$\|f\|_k^2 = \int \|f\|_2^2 \frac{d\mu_k}{d\mu_s} d\mu_s \leq \eta^2 \|f\|_s^2 \implies \|f\|_k \leq \eta \|f\|_s.$$

Now we show "⇒": Choose an arbitrary $y \neq 0 \in \mathcal{Y}$ and measurable $A \subset \mathcal{X}$, and let $f = (y/\|y\|_2)\mathbb{1}_A$. Note that $\|f\|_2^2 = \mathbb{1}_A$. Then for all $s = s_{\boldsymbol{\lambda}}^{\lin} \in \mathcal{S}$

$$\mu_k(A) = \int \|f\|_2^2 \, \mu_k = \|f\|_k^2 \leq \eta^2 \|f\|_s^2 = \eta^2 \sum_{j=1}^{K} \lambda_j \|f\|_j^2 = \eta^2 \sum_{j=1}^{K} \lambda_j \mu_j(A) = \eta^2 \mu_s(A).$$

This implies the bound $\alpha := \operatorname{ess\,sup} d\mu_k/d\mu_s \leq \eta^2$, since for any $\varepsilon > 0$ we can choose the measurable event $A_\varepsilon := \{x : d\mu_k/d\mu_s(x) \geq \alpha - \varepsilon\}$ which satisfies (by definition) $\mu_s(A_\varepsilon) > 0$ and so

$$\eta^2 \geq \frac{\mu_k(A_\varepsilon)}{\mu_s(A_\varepsilon)} = \frac{1}{\mu_s(A_\varepsilon)} \int_{A_\varepsilon} \frac{d\mu_k}{d\mu_s} d\mu_s \geq \alpha - \varepsilon.$$

Taking $\varepsilon \to 0$ concludes the proof. $\qquad\square$

### D.4.4 Proof of Lemma D.8

For the first function, the argument is standard, we repeat it here for completeness. Let $0 < r < r'$ and consider some $h \in \mathcal{H}_k(r')$. Then $\|h\|_k \leq r'$ and hence $\|(r/r')h\|_k \leq r$, so that $(r/r')h \in \mathcal{H}_k(r)$ by the star-shape of $\mathcal{H}_k$ from Assumption 2. Therefore, we have that

$$\frac{r}{r'}\mathfrak{R}_{n_k}^k(\mathcal{H}_k(r')) = \mathbb{E}\left[\sup_{h \in \mathcal{H}(r')} \frac{1}{n_k} \sum_{i=1}^{n_k} \sum_{j=1}^{q} \sigma_{ij} \frac{r}{r'} h_j(X_i)\right] \leq \mathfrak{R}_{n_k}^k(\mathcal{H}_k(r))$$

which is the claim.

For the other function the proof is identical once we realize that the convexity of $\mathcal{G}$ from Assumption 2 implies that $\mathcal{G}_k(r; \boldsymbol{h})$ is star-shaped around the origin. Indeed, for any $\boldsymbol{h} \in \mathcal{H}_1 \times \cdots \times \mathcal{H}_K$ and $g_s^{\boldsymbol{h}} = \arg\min_{g \in \mathcal{G}} d_s(g; \boldsymbol{h})$, since $(\mathcal{G} - g_s^{\boldsymbol{h}}) \cap r\mathcal{B}_k$ is convex and contains the origin,

$$g \in \bigcup_{s \in \mathcal{S}} (\mathcal{G} - g_s^{\boldsymbol{h}}) \cap r\mathcal{B}_k \implies \forall \alpha \in [0,1], \ \alpha g \in \bigcup_{s \in \mathcal{S}} (\mathcal{G} - g_s^{\boldsymbol{h}}) \cap r\mathcal{B}_k.$$

We require this star-shapedness for all $\boldsymbol{h} \in \mathcal{H}_1 \times \cdots \times \mathcal{H}_K$, because we also localize around $g_s' = g_s^{\widehat{\boldsymbol{h}}}$ that are random elements and may be anywhere in $\mathcal{G}$.

### D.4.5 Proof of Lemma D.9

The proof of this Lemma is a mixture of Corollary 5.3 in [9] and Theorem 14.20 in [63]; see also [35] for an exposition. We repeat it here for completeness and because we make slightly different assumptions from [9, 63], see Remark 1 below. Recall the definition of the sets for any $r \geq 0$,

$$\mathcal{H}_k(r) := (\mathcal{H}_k - f_k^\star) \cap r\mathcal{B}_{\|\cdot\|_k}$$

and the random variables

$$T_k(r) = \sup_{h - f_k^\star \in \mathcal{H}_k(r)} \left|(\mathcal{R}_k(h) - \mathcal{R}_k(f_k^\star)) - (\widehat{\mathcal{R}}_k(h) - \widehat{\mathcal{R}}_k(f_k^\star))\right|$$

which are the suprema of empirical processes indexed by the function classes defined as

$$\{(x,y) \mapsto \ell_k(y, h(x)) - \ell_k(y, f_k^\star(x)) : h - f_k^\star \in \mathcal{H}_k(r)\}.$$

By Assumption 1, these function classes are uniformly bounded by $B_k \geq 0$. Hence, by Talagrand's concentration inequality (Lemma E.3), for any choice of *deterministic* radii $r_1, \ldots, r_K \geq 0$, the event

$$Q_\delta^{\lab}(r_1, \ldots, r_K) := \left\{\forall k \in [K] : T_k(r_k) \leq 2\mathbb{E}[T_k(r_k)] + \sqrt{2}\sqrt{\frac{\tau_k^2(r_k)\log(K/\delta)}{n_k}} + 3\frac{B\log(K/\delta)}{n_k}\right\}$$

holds with probability at least $1 - \delta$. Here $\tau_k^2(r)$ is a short-hand for the variance proxy from Lemma E.3, defined as

$$\tau_k^2(r) = \sup_{h - f_k^\star \in \mathcal{H}_k(r)} \operatorname{Var}\left[\ell_k(Y^k, h(X^k)) - \ell_k(Y^k, f_k^\star(X^k))\right].$$

We now bound $\mathbb{E}\left[T_k(r_k)\right]$ and $\tau_k^2(r_k)$. Using symmetrization (Lemma E.5) and vector contraction (Lemma E.4), recalling that $\ell_k$ is $L_k$-Lipschitz w.r.t. the $\ell_2$-norm in its second argument, we can bound

$$\mathbb{E}\left[T_k(r)\right] \le 6L_k \mathfrak{R}_{n_k}^k(\mathcal{H}_k(r)) \qquad \text{and} \qquad \tau_k^2(r) \le L_k^2 r^2.$$

Therefore, we get on the event $Q_\delta^{\text{lab}}(r_1, \ldots, r_K)$ that for all $k \in [K]$

$$T_k(r_k) \le 12 L_k \mathfrak{R}_{n_k}^k(\mathcal{H}_k(r_k)) + \sqrt{2} L_k r_k \sqrt{\frac{\log(K/\delta)}{n_k}} + 3B \frac{\log(K/\delta)}{n_k}.$$

Now recall the definition

$$\mathfrak{l}_k := \inf\left\{r \ge 0 : r^2 \ge \mathfrak{R}_{n_k}^k(\mathcal{H}_k(r))\right\}.$$

By (34), we get that for any $r \ge \mathfrak{l}_k$

$$\frac{\mathfrak{R}_{n_k}^k(\mathcal{H}_k(r))}{r} \le \frac{\mathfrak{R}_{n_k}^k(\mathcal{H}_k(\mathfrak{l}_k))}{\mathfrak{l}_k} \le \mathfrak{l}_k$$

and therefore, if $r_k \ge \mathfrak{l}_k$ for all $k$, on the event $Q_\delta^{\text{lab}}(r_1, \ldots, r_K)$ (which holds with probability at least $1 - \delta$), it holds that

$$T_k(r_k) \le 12 L_k r_k \mathfrak{l}_k + \sqrt{2} L_k r_k \sqrt{\frac{\log(K/\delta)}{n_k}} + 3B \frac{\log(K/\delta)}{n_k} =: \Phi_k(r_k, \delta).$$

We now choose $r_k := \left\|\widehat{h}_k - f_k^\star\right\|_k$, which are *random* radii, so we have to perform a peeling argument. Define the event

$$W_\delta^{\text{lab}} := \left\{\exists k \in [K],\, h \in \mathcal{H}_k : \|h - f_k^\star\|_k \ge \mathfrak{l}_k \text{ and } T_k(\|h - f_k^\star\|_k) \ge 3\Phi_k(\|h - f_k^\star\|_k, \delta)\right\}.$$

Because $\|h - f_k^\star\|_k \le \operatorname{diam}_{\|\cdot\|_2}(\mathcal{Y}) =: D$, we know that for any $M$ satisfying $2^M \mathfrak{l}_k \ge D \iff M \ge \log(D/\mathfrak{l}_k)/\log(2)$, for any $\|h - f_k^\star\|_k \ge \mathfrak{l}_k$ there must be at least one $0 \le m \le M$ so that $2^{m-1}\mathfrak{l}_k \le \|h - f_k^\star\|_k \le 2^m \mathfrak{l}_k$. Moreover, a calculation shows that the functions $\Phi_k$ satisfy

$$\forall m \le M : \quad 3\Phi_k(2^{m-1}\mathfrak{l}_k, \delta) \ge \Phi_k(2^m \mathfrak{l}_k, \delta/2^m).$$

for sufficiently small $\delta$, and so $\mathbb{P}(W_\delta^{\text{lab}})$ is bounded by

$$\mathbb{P}\left( \bigcup_{m \in [M]} \left\{\exists k \in [K], h \in \mathcal{H}_k : 2^{m-1}\mathfrak{l}_k \le \|h - f_k^\star\|_k \le 2^m \mathfrak{l}_k \right.\right.$$

$$\left.\left. \text{and } T_k(\|h - f_k^\star\|_k) \ge 3\Phi_k(\|h - f_k^\star\|_k, \delta)\right\}\right)$$

$$\le \sum_{m \in [M]} \mathbb{P}\left(\exists k \in [K] : T_k(2^m \mathfrak{l}_k) \ge 3\Phi_k(2^{m-1}\mathfrak{l}_k, \delta)\right)$$

$$\le \sum_{m \in [M]} \mathbb{P}\left(\exists k \in [K] : T_k(2^m \mathfrak{l}_k) \ge \Phi_k(2^m \mathfrak{l}_k, \delta/2^m)\right) \overset{(a)}{\le} \sum_{m \in [M]} \frac{\delta}{2^m} \le \delta.$$

where in $(a)$ we used that $\mathbb{P}(Q_{\delta/2^m}^{\text{lab}}(2^m \mathfrak{l}_1, \ldots, 2^m \mathfrak{l}_K)) \ge 1 - \delta/2^m$.

Now, by the standard risk decomposition, we have that

$$\mathcal{R}_k(\widehat{h}_k) - \mathcal{R}_k(f_k^\star) = \mathcal{R}_k(\widehat{h}_k) - \widehat{\mathcal{R}}_k(\widehat{h}_k) + \underbrace{\widehat{\mathcal{R}}_k(\widehat{h}_k) - \widehat{\mathcal{R}}_k(f_k^\star)}_{\le 0} + \widehat{\mathcal{R}}_k(f_k^\star) - \mathcal{R}_k(f_k^\star)$$

$$\le 2T_k\left(\left\|\widehat{h}_k - f_k^\star\right\|_k\right),$$

and we can make a case distinction.

*Remark* 1. Many localization proofs for general loss functions only assume strong convexity and Lipschitz continuity (see, e.g., Section 14.3 in [63]), and therefore one needs to handle the case where the $L^2$-radius is bounded but the excess loss is not (tightly) bounded, which would occur in the first case below. In our setting, by the smoothness (Lemma D.6), a bounded radius directly implies bounded excess risk, so this case cannot occur and no separate treatment is required.

Either $r_k = \left\|\widehat{h}_k - f_k^\star\right\|_k \leq \mathfrak{l}_k$ and we are done, or $r_k = \left\|\widehat{h}_k - f_k^\star\right\|_k > \mathfrak{l}_k$, and so, because $\mathbb{P}(W_\delta^{\text{lab}}) \leq \delta$, we have with probability at least $1 - \delta$

$$T_k(r_k) = T_k\left(\left\|\widehat{h}_k - f_k^\star\right\|_k\right) \leq 3\Phi_k\left(\left\|\widehat{h}_k - f_k^\star\right\|_k, \delta\right) = 3\Phi_k(r_k, \delta).$$

Recall that by Assumption 1, $\phi_k$ is $\mu_k$-strongly convex w.r.t. $\|\cdot\|_2$, so that $\ell_k(y, y') \geq \frac{\mu_k}{2}\|y - y'\|_2^2$. Hence, we have that

$$
\begin{aligned}
r_k^2 &= \mathbb{E}_{X^k \sim P_X^k} \left\|\widehat{h}_k(X^k) - f_k^\star(X^k)\right\|_2^2 \\
&\leq \frac{2}{\mu_k} \mathbb{E}_{X^k \sim P_X^k} \ell_k\left(f_k^\star(X^k), \widehat{h}_k\right) = \frac{2}{\mu_k}\left(\mathcal{R}_k(\widehat{h}_k) - \mathcal{R}_k(f_k^\star)\right) \leq \frac{4}{\mu_k}T_k(r_k) \leq \frac{12}{\mu_k}\Phi_k(r_k, \delta),
\end{aligned}
$$

where we used Lemma 1 in the second equality. Solving $r_k^2 \leq \frac{12}{\mu_k}\Phi_k(r_k, \delta)$ for $r_k$, we get that

$$r_k^2 \leq \frac{82944\, L_k^2}{\mu_k^2}\mathfrak{l}_k^2 + \frac{144}{\mu_k}\left(B_k + \frac{8\, L_k^2}{\mu_k}\right)\frac{\log(K/\delta)}{n_k}.$$

Hence, in either case we have

$$r_k^2 \lesssim \frac{L_k^2}{\mu_k^2}\mathfrak{l}_k^2 + \left(\frac{B_k}{\mu_k} + \frac{L_k^2}{\mu_k^2}\right)\frac{\log(K/\delta)}{n_k}.$$

Therefore, because $\mathbb{P}(W_\delta^{\text{lab}}) \leq \delta$, we have that $\mathbb{P}(E_\delta^{\text{lab}}) \geq 1 - \delta$, where

$$E_\delta^{\text{lab}} := \left\{\forall k \in [K]: \quad \left\|\widehat{h}_k - f_k^\star\right\|_k^2 \lesssim \frac{L_k^2}{\mu_k^2}\mathfrak{l}_k^2 + \left(\frac{B_k}{\mu_k} + \frac{L_k^2}{\mu_k^2}\right)\frac{\log(K/\delta)}{n_k}\right\}.$$

which concludes the proof for localization in $\mathcal{H}_k$.

### D.4.6 Proof of Proposition D.1

Recall the form of the gradient $\nabla_g d_s(g; \boldsymbol{h})$ from Lemma D.7. For every fixed $g$, and any $\boldsymbol{h}, \boldsymbol{h}'$,

$$
\begin{aligned}
&\left\|\nabla_g d_s(g; \boldsymbol{h}) - \nabla_g d_s(g; \boldsymbol{h}')\right\|_s^2 \\
&= \int \left\|\sum_{k=1}^K \lambda_k \frac{d\mu_k}{d\mu_s}(\nabla^2\phi_k(g)(h_k - g) - \nabla^2\phi_k(g)(h'_k - g))\right\|_2^2 d\mu_s \\
&\leq \int \sum_{k=1}^K \lambda_k \frac{d\mu_k}{d\mu_s}\left\|(\nabla^2\phi_k(g)(h_k - g) - \nabla^2\phi_k(g)(h'_k - g))\right\|_2^2 d\mu_s && \text{(Jensen's inequality)} \\
&\leq \nu^2 \sum_{k=1}^K \lambda_k \int \frac{d\mu_k}{d\mu_s}\|h_k - h'_k\|_2^2 d\mu_s \\
&= \nu^2 \sum_{k=1}^K \lambda_k \|h_k - h'_k\|_k^2. && (42)
\end{aligned}
$$

This is what we call "cross-smoothness".

Denote $g = g_s^{\boldsymbol{h}}$ and $g' = g_s^{\boldsymbol{h}'}$. We may now use a generalization of the stability argument used in the proof of Theorem 1 in [65], where the following argument was used in $\mathbb{R}^m$ and for unconstrained

optimization: By the convexity of $\mathcal{G}$ (Assumption 2), and the optimality of $g, g'$ we get these two variational inequalities

$$\langle \nabla_g d_s(g; \boldsymbol{h}), g' - g \rangle_s \geq 0 \quad \text{and} \quad \langle \nabla_g d_s(g'; \boldsymbol{h}'), g - g' \rangle_s \geq 0 \iff \langle \nabla_g d_s(g'; \boldsymbol{h}'), g' - g \rangle_s \leq 0,$$

see Lemma 2 and [69, Theorem 46]. Combining both, and subtracting $\langle \nabla_g d_s(g; \boldsymbol{h}'), g' - g \rangle$ on both sides we see that

$$\langle \nabla_g d_s(g; \boldsymbol{h}) - \nabla_g d_s(g; \boldsymbol{h}'), g' - g \rangle_s \geq \langle \nabla_g d_s(g'; \boldsymbol{h}') - \nabla_g d_s(g; \boldsymbol{h}'), g' - g \rangle_s. \tag{43}$$

From the second item in Assumption 2, and the main results in [49], we get that the right-hand side of (43) is lower bounded as

$$2\gamma \|g - g'\|_s^2 \leq \langle \nabla_g d_s(g'; \boldsymbol{h}') - \nabla_g d_s(g; \boldsymbol{h}'), g' - g \rangle_s,$$

and from the cross-smoothness in (42), we get that the left-hand side of (43) is upper bounded by

$$\langle \nabla_g d_s(g; \boldsymbol{h}) - \nabla_g d_s(g; \boldsymbol{h}'), g' - g \rangle_s \leq \left\| \nabla_g d_s(g; \boldsymbol{h}) - \nabla_g d_s(g; \boldsymbol{h}') \right\|_s \|g' - g\|_s$$

$$\leq \nu \|g' - g\|_s \sqrt{\sum_{k=1}^{K} \lambda_k \|h_k - h'_k\|_k^2}.$$

Combining the two, we can see that

$$2\gamma \|g - g'\|_s^2 \leq \nu \|g' - g\|_s \sqrt{\sum_{k=1}^{K} \lambda_k \|h_k - h'_k\|_k^2} \implies \|g - g'\|_s^2 \leq \frac{\nu^2}{4\gamma^2} \sum_{k=1}^{K} \lambda_k \|h_k - h'_k\|_k^2.$$

This is the claimed quadratic bound.

### D.4.7 Proof of Proposition D.2

Throughout this proof, condition on the $\widehat{h}_k$. In particular, all expectations and variances are conditioned on $\widehat{h}_k$. Recall from Eq. (33) that for any $r \geq 0$

$$\mathcal{G}_k(r; \widehat{\boldsymbol{h}}) = \bigcup_{s \in \mathcal{S}} (\mathcal{G} - g'_s) \cap r\mathcal{B}_{\|\cdot\|_k}$$

and from Eq. (37) that

$$\mathcal{M}_k(r) = \left\{ (s, g) : s \in \mathcal{S}, g - g'_s \in \mathcal{G}_k(r; \widehat{\boldsymbol{h}}) \right\}.$$

The first part of this proof is mostly standard and follows the same proof structure as Lemma D.9. Define the random variables

$$Z_k(r) := \sup_{(s,g) \in \mathcal{M}_k(r)} \left| (\widehat{d}_k(g; \widehat{h}_k) - \widehat{d}_k(g'_s; \widehat{h}_k)) - (d_k(g; \widehat{h}_k) - d_k(g'_s; \widehat{h}_k)) \right|$$

$$= \sup_{(s,g) \in \mathcal{M}_k(r)} \left| \frac{1}{N_k} \sum_{i=1}^{N_k} (\ell_k(\widehat{h}_k(\widetilde{X}_i^k), g(\widetilde{X}_i^k)) - \ell_k(\widehat{h}_k(\widetilde{X}_i^k), g'_s(\widetilde{X}_i^k))) \right.$$

$$\left. - \mathbb{E}(\ell_k(\widehat{h}_k(X^k), g(X^k)) - \ell_k(\widehat{h}_k(X^k), g'_s(X^k))) \right|$$

which are the suprema of an empirical processes over the function classes for $k \in [K]$

$$\left\{ x \mapsto \ell_k(\widehat{h}_k(x), g(x)) - \ell_k(\widehat{h}_k(x), g'_s(x)) : (s, g) \in \mathcal{M}_k(r) \right\}$$

By Assumption 1, these function classes are uniformly bounded by $B_k \geq 0$. Hence, by Talagrand's concentration inequality (Lemma E.3), for any choice of *deterministic* radii $r_1, \ldots, r_K \geq 0$, the event

$$Q_\delta^{\mathrm{un}}(r_1, \ldots, r_K) = \left\{ \forall k \in [K] : Z_k(r_k) \leq 2\mathbb{E}[Z_k(r_k)] + \sqrt{2}\sqrt{\frac{\sigma_k^2(r_k) \log(K/\delta)}{N_k}} + 3\frac{B_k \log(K/\delta)}{N_k} \right\}$$

holds with probability at least $1 - \delta$. Here $\sigma_k^2(r_k)$ is a short-hand for the variance proxy from Lemma E.3, defined in this section as

$$\sigma_k^2(r) = \sup_{(s,g) \in \mathcal{M}_k(r)} \mathrm{Var}\left[\ell_k(\widehat{h}_k(X^k), g(X^k)) - \ell_k(\widehat{h}_k(X^k), g_s'(X^k))\right].$$

We now bound $\mathbb{E}[Z_k(r)]$ and $\sigma_k^2(r)$. Using symmetrization (Lemma E.5) in addition to vector contraction (Lemma E.4), recalling that $\ell_k$ is $L_k$-Lipschitz w.r.t. $\ell_2$-norm in its second argument, we can bound

$$\mathbb{E}[Z_k(r)] \leq 6L_k \mathfrak{R}_{N_k}^k(\mathcal{G}_k(r; \widehat{\boldsymbol{h}})) \qquad \text{and} \qquad \sigma_k^2(r) \leq L_k^2 r^2.$$

Therefore, we get on the event $Q_\delta^{\mathrm{un}}(r_1, \dots, r_K)$ that

$$Z_k(r_k) \leq 12 L_k \mathfrak{R}_{N_k}^k(\mathcal{G}_k(r; \widehat{\boldsymbol{h}})) + \sqrt{2} L_k r_k \sqrt{\frac{\log(K/\delta)}{N_k}} + 6 B_k \frac{\log(K/\delta)}{N_k}.$$

Define

$$\widetilde{\mathfrak{u}}_k := \inf\left\{r \geq 0 : r^2 \geq \mathfrak{R}_{N_k}^k(\mathcal{G}_k(r; \widehat{\boldsymbol{h}}))\right\}.$$

By Lemma D.8, which holds under Assumption 2, we get that for any $r \geq \widetilde{\mathfrak{u}}_k$

$$\frac{\mathfrak{R}_{N_k}^k(\mathcal{G}_k(r; \widehat{\boldsymbol{h}}))}{r} \leq \frac{\mathfrak{R}_{N_k}^k(\mathcal{G}_k(\widetilde{\mathfrak{u}}_k; \widehat{\boldsymbol{h}}))}{\widetilde{\mathfrak{u}}_k} \leq \widetilde{\mathfrak{u}}_k.$$

Therefore, if $r_k \geq \widetilde{\mathfrak{u}}_k$ for all $k$, on the event $Q_\delta^{\mathrm{un}}(r_1, \dots, r_K)$ (which holds with probability at least $1 - \delta$), it holds that for all $k \in [K]$,

$$Z_k(r_k) \leq 12 L_k r_k \widetilde{\mathfrak{u}}_k + \sqrt{2} L_k r_k \sqrt{\frac{\log(K/\delta)}{N_k}} + 3 B_k \frac{\log(K/\delta)}{N_k} =: \Psi_k(r_k, \delta).$$

We now come to the part of the proof that is less standard. Consider the family of *random* radii

$$r_k^s := \|\widehat{g}_s - g_s'\|_k \qquad s \in \mathcal{S}.$$

We perform a peeling argument to bound the probabilities of the two events

$W_{\delta,0}^{\mathrm{un}} := \{\exists k \in [K] : Z_k(\widetilde{\mathfrak{u}}_k) \geq \Psi_k(\widetilde{\mathfrak{u}}_k, \delta)\}$

$W_{\delta,1}^{\mathrm{un}} := \{\exists k \in [K], s \in \mathcal{S}, g \in \mathcal{G} : \|g - g_s'\|_k \geq \widetilde{\mathfrak{u}}_k \text{ and } Z_k(\|g - g_s'\|_k) \geq 3\Psi_k(\|g - g_s'\|_k, \delta)\}.$

*Remark* 2. Contrary to Remark 1, here we include the case where the radii are small, because we have to control all $K$ radii simultaneously. One could also adapt the following proof without this case, but the resulting bound would be the same (up to constants).

By the previous derivations, $\mathbb{P}\left(W_{\delta,0}^{\mathrm{un}}\right) \leq \delta$, and for $W_{\delta,1}^{\mathrm{un}}$ we apply a peeling argument. Because $\|g - g_s'\|_k \leq \mathrm{diam}_{\|\cdot\|_2}(\mathcal{Y}) =: D$, we know that for any $M$ satisfying

$$2^M \widetilde{\mathfrak{u}}_k \geq D \iff M \geq \log(D/\widetilde{\mathfrak{u}}_k) / \log(2),$$

and for any $\|g - g_s'\|_k \geq \widetilde{\mathfrak{u}}_k$ there must be at least one $0 \leq m \leq M$ so that $2^{m-1} \widetilde{\mathfrak{u}}_k \leq \|g - g_s'\|_k \leq 2^m \widetilde{\mathfrak{u}}_k$. Moreover, a calculation shows that the functions $\Psi_k$ satisfy

$$\forall 0 \leq m \leq M : \quad 3\Psi_k(2^{m-1} \widetilde{\mathfrak{u}}_k, \delta) \geq \Psi_k(2^m \widetilde{\mathfrak{u}}_k, \delta/2^m)$$

for small enough $\delta$, which yields that

$$\mathbb{P}\left(W_{\delta,1}^{\mathrm{un}}\right) = \mathbb{P}\left(\bigcup_{m \in [M]} \{\exists k \in [K], s \in \mathcal{S}, g \in \mathcal{G} : 2^{m-1} \widetilde{\mathfrak{u}}_k \leq \|g - g_s'\|_k \leq 2^m \widetilde{\mathfrak{u}}_k \right.$$

$$\left. \text{and } Z_k(\|g - g_s'\|_k) \geq 3\Psi_k(\|g - g_s'\|_k, \delta)\}\right)$$

$$\leq \sum_{m \in [M]} \mathbb{P}\left(\exists k \in [K] : Z_k(2^m \widetilde{\mathfrak{u}}_k) \geq 3\Psi_k(2^{m-1} \widetilde{\mathfrak{u}}_k, \delta)\right)$$

$$\leq \sum_{m \in [M]} \mathbb{P}\left(\exists k \in [K] : Z_k(2^m \widetilde{\mathfrak{u}}_k) \geq \Psi_k(2^m \widetilde{\mathfrak{u}}_k, \delta/2^m)\right) \overset{(a)}{\leq} \sum_{m \in [M]} \frac{\delta}{2^m} \leq \delta.$$

where in $(a)$ we used that $\mathbb{P}(Q^{\mathrm{un}}_{\delta/2^m}(2^m \widetilde{\mathfrak{u}}_1, \ldots, 2^m \widetilde{\mathfrak{u}}_K)) \geq 1 - \delta/2^m$. Combining the two with a union bound yields $\mathbb{P}\left((W^{\mathrm{un}}_{\delta,0})^c \cap (W^{\mathrm{un}}_{\delta,1})^c\right) \geq 1 - 2\delta$. Condition on $(W^{\mathrm{un}}_{\delta,0})^c \cap (W^{\mathrm{un}}_{\delta,1})^c$.

By the standard risk decomposition, we get for all $s \in \mathcal{S}$

$$d_s(\widehat{g}_s; \widehat{\boldsymbol{h}}) - d_s(g'_s; \widehat{\boldsymbol{h}}) = d_s(\widehat{g}_s; \widehat{\boldsymbol{h}}) - \widehat{d}_s(\widehat{g}_s; \widehat{\boldsymbol{h}}) + \underbrace{\widehat{d}_s(\widehat{g}_s; \widehat{\boldsymbol{h}}) - \widehat{d}_s(g'_s; \widehat{\boldsymbol{h}})}_{\leq 0} + \widehat{d}_s(g'_s; \widehat{\boldsymbol{h}}) - d_s(g'_s; \widehat{\boldsymbol{h}})$$

$$\leq d_s(\widehat{g}_s; \widehat{\boldsymbol{h}}) - \widehat{d}_s(\widehat{g}_s; \widehat{\boldsymbol{h}}) + \widehat{d}_s(g'_s; \widehat{\boldsymbol{h}}) - d_s(g'_s; \widehat{\boldsymbol{h}})$$

$$= s\left(\left(d_k(\widehat{g}_s; \widehat{h}_k) - \widehat{d}_k(\widehat{g}_s; \widehat{h}_k) + \widehat{d}_k(g_s; \widehat{h}_k) - d_k(g; \widehat{h}_k)\right)_{k \in [K]}\right)$$

$$\leq s\left((Z_k(r^s_k))_{k \in [K]}\right) \tag{44}$$

Further, by the "multi-objective Bernstein condition" $\|\widehat{g}_s - g'_s\|^2_s \leq \frac{1}{\gamma}(d_s(\widehat{g}_s; \widehat{\boldsymbol{h}}) - d_s(g_s; \widehat{\boldsymbol{h}}))$, implied by the second item from Assumption 2, and Eq. (44),

$$r^2_s := s\left((r^s_1)^2, \ldots, (r^s_K)^2\right) = \|\widehat{g}_s - g'_s\|^2_s$$

$$\leq \frac{1}{\gamma}\left(d_s(\widehat{g}_s; \widehat{\boldsymbol{h}}) - d_s(g'_s; \widehat{\boldsymbol{h}})\right) \leq \frac{1}{\gamma}s\left((Z_k(r^s_k))_{k \in [K]}\right).$$

On the event $(W^{\mathrm{un}}_{\delta,0})^c \cap (W^{\mathrm{un}}_{\delta,1})^c$, we thus get for every $s = s^{\mathrm{lin}}_{\boldsymbol{\lambda}} \in \mathcal{S}$

$$r^2_s \leq \frac{1}{\gamma}\left(\sum_{k:\, r^s_k \leq \widetilde{\mathfrak{u}}_k} \lambda_k \Psi_k(\widetilde{\mathfrak{u}}_k, \delta) + \sum_{k:\, r^s_k > \widetilde{\mathfrak{u}}_k} \lambda_k 3\Psi_k(r^s_k, \delta)\right)$$

$$\leq \frac{1}{\gamma}\left(\sum_{k=1}^K \lambda_k \Psi_k(\widetilde{\mathfrak{u}}_k, \delta) + \sum_{k=1}^K \lambda_k 3\Psi_k(r^s_k, \delta)\right).$$

We can simplify the first term using $ab \leq \frac{1}{2}(a^2 + b^2)$ as

$$\frac{1}{\gamma}\sum_{k=1}^K \lambda_k \Psi_k(\widetilde{\mathfrak{u}}_k, \delta) = \frac{1}{\gamma}\sum_{k=1}^K \lambda_k \left(12 L_k \widetilde{\mathfrak{u}}^2_k + \sqrt{2} L_k \widetilde{\mathfrak{u}}_k \sqrt{\frac{\log(K/\delta)}{N_k}} + 3 B_k \frac{\log(K/\delta)}{N_k}\right)$$

$$\leq \frac{1}{\gamma}\sum_{k=1}^K \lambda_k \left(13 L_k \widetilde{\mathfrak{u}}^2_k + (L_k + 3 B_k)\frac{\log(K/\delta)}{N_k}\right) =: b_{s,1}.$$

Plugging this into the bound on $r^2_s$ yields

$$r^2_s \leq \frac{1}{\gamma}\sum_{k=1}^K \lambda_k \left(36 L_k r^s_k \widetilde{\mathfrak{u}}_k + 5 L_k r^s_k \sqrt{\frac{\log(K/\delta)}{N_k}} + 18 B_k \frac{\log(K/\delta)}{N_k}\right) + b_{s,1}$$

$$= \frac{1}{\gamma}\sum_{k=1}^K \left(\sqrt{\lambda_k} \cdot r^s_k\right)\left(\sqrt{\lambda_k}\underbrace{\left(36 L_k \widetilde{\mathfrak{u}}_k + 5 L_k \sqrt{\frac{\log(K/\delta)}{N_k}}\right)}_{=:a_k}\right) + 18\underbrace{\frac{1}{\gamma}\sum_{k=1}^K \lambda_k \frac{B_k \log(K/\delta)}{N_k}}_{=:b_{s,2}} + b_{s,1}$$

$$\leq \frac{1}{\gamma}\left(\sum_{k=1}^K \lambda_k (r^s_k)^2\right)^{1/2}\left(\sum_{k=1}^K \lambda_k a^2_k\right)^{1/2} + \underbrace{b_{s,1} + b_{s,2}}_{=:b_s}$$

$$= r_s \underbrace{\left(\sum_{k=1}^K \lambda_k \frac{a^2_k}{\gamma^2}\right)^{1/2}}_{=:a_s} + b_s$$

where the last inequality is Cauchy-Schwarz. Some algebra shows that $r_s^2 \le r_s a_s + b_s$ implies $r_s^2 \le 2(a_s^2 + b_s)$, and so

$$r_s^2 \le 2 \sum_{k=1}^{K} \lambda_k \left( \frac{1}{\gamma^2} \left( 36 L_k \widetilde{\mathfrak{u}}_k + 5 L_k \sqrt{\frac{\log(K/\delta)}{N_k}} \right)^2 + 13 \frac{L_k}{\gamma} \widetilde{\mathfrak{u}}_k^2 \right.$$
$$\left. + \frac{(L_k + 3 B_k)}{\gamma} \frac{\log(K/\delta)}{N_k} + \frac{18}{\gamma} \frac{B_k \log(K/\delta)}{N_k} \right)$$

$$\lesssim \sum_{k=1}^{K} \lambda_k \left( \left( \frac{L_k^2}{\gamma^2} + \frac{L_k}{\gamma} \right) \widetilde{\mathfrak{u}}_k^2 + \left( \frac{L_k^2}{\gamma^2} + \frac{L_k}{\gamma} + \frac{B_k}{\gamma} \right) \frac{\log(K/\delta)}{N_k} \right)$$

$$\lesssim \sum_{k=1}^{K} \lambda_k \left( \frac{L_k^2}{\gamma^2} \widetilde{\mathfrak{u}}_k^2 + \left( \frac{L_k^2}{\gamma^2} + \frac{B_k}{\gamma} \right) \frac{\log(K/\delta)}{N_k} \right)$$

where in the last line we used $L_k/\gamma \ge 1$. Therefore, because $\mathbb{P}\left( (W_{\delta,0}^{\mathrm{un}})^c \cap (W_{\delta,1}^{\mathrm{un}})^c \right) \ge 1 - 2\delta$, we have that $\mathbb{P}\left( E_\delta^{\mathrm{un}} \right) \ge 1 - \delta$, where

$$E_\delta^{\mathrm{un}} := \left\{ \forall s = s_{\boldsymbol{\lambda}}^{\mathrm{lin}} \in \mathcal{S} : \quad \|\widehat{g}_s - g_s'\|_s^2 \lesssim \sum_{k=1}^{K} \lambda_k \left( \frac{L_k^2}{\gamma^2} \widetilde{\mathfrak{u}}_k^2 + \left( \frac{L_k^2}{\gamma^2} + \frac{B_k}{\gamma} \right) \frac{\log(2K/\delta)}{N_k} \right) \right\},$$

which concludes the proof of this part.

### D.4.8 Proof of Proposition D.3

Recall that $\Delta = \sup_{s \in \mathcal{S}} \|g_s - g_s'\|$. For every $r \ge 0$, we have the following key inclusion
$$\mathcal{G}'(r) \subset \mathcal{G}(r + \Delta) + \{g_s - g_s' : s \in \mathcal{S}\}$$
To see that, let $h = g - g_s' \in \mathcal{G}'(r)$. Then $h + (g_s' - g_s) = g - g_s$ and $\|h + (g_s' - g_s)\| \le r + \Delta$.

Because Rademacher complexity is sub-additive, we get that for all $r \ge 0$
$$\mathfrak{R}_n\left( \mathcal{G}'(r) \right) \le \mathfrak{R}_n\left( \mathcal{G}(r + \Delta) + \{g_s - g_s' : s \in \mathcal{S}\} \right)$$
$$\le \mathfrak{R}_n\left( \mathcal{G}(r + \Delta) \right) + \mathfrak{R}_n\left( \{g_s - g_s' : s \in \mathcal{S}\} \right)$$
$$\le \mathfrak{R}_n\left( \mathcal{G}(r + \Delta) \right) + \mathfrak{R}_n\left( \mathcal{G}(\Delta) \right)$$
where in the last step we used that
$$-\{g_s - g_s' : s \in \mathcal{S}\} = \{g_s' - g_s : s \in \mathcal{S}\} \subset \mathcal{G}(\Delta).$$

Using that for all $r \ge \mathfrak{u}$ we have $\mathfrak{R}_n\left( \mathcal{G}(r) \right) \le r^2$, we get that
$$\mathfrak{R}_n\left( \mathcal{G}'(\mathfrak{u} + \Delta) \right) \le \mathfrak{R}_n\left( \mathcal{G}(\mathfrak{u} + 2\Delta) \right) + \mathfrak{R}_n\left( \mathcal{G}(\Delta) \right) \le (\mathfrak{u} + 2\Delta)^2 + \mathfrak{R}_n\left( \mathcal{G}(\Delta) \right)$$
and using that $r \mapsto \mathfrak{R}_n\left( \mathcal{G}(r) \right) / r$ is non-increasing (by Assumption 2 and Lemma D.8), we get that
$$\mathfrak{R}_n\left( \mathcal{G}(\Delta) \right) \le \frac{\Delta}{\mathfrak{u} + \Delta} \mathfrak{R}_n\left( \mathcal{G}(\mathfrak{u} + \Delta) \right) \le \Delta(\mathfrak{u} + \Delta),$$
which together yields
$$\mathfrak{R}_n\left( \mathcal{G}'(\mathfrak{u} + \Delta) \right) \le (\mathfrak{u} + 2\Delta)^2 + \Delta(\mathfrak{u} + \Delta).$$

Again, by the fact that $r \mapsto \mathfrak{R}_n\left( \mathcal{G}'(r) \right) / r$ is non-increasing (Lemma D.8), we get that for all $r \ge \mathfrak{u} + \Delta$
$$\mathfrak{R}_n\left( \mathcal{G}'(r) \right) \le \frac{r}{\mathfrak{u} + \Delta} \mathfrak{R}_n\left( \mathcal{G}'(\mathfrak{u} + \Delta) \right) \le \frac{r}{\mathfrak{u} + \Delta} \left( (\mathfrak{u} + 2\Delta)^2 + \Delta(\mathfrak{u} + \Delta) \right)$$

In particular, for the choice $r = 5(\mathfrak{u} + \Delta)$
$$\mathfrak{R}_n\left( \mathcal{G}'(5(\mathfrak{u} + \Delta)) \right) \le \frac{5(\mathfrak{u} + \Delta)}{\mathfrak{u} + \Delta} \left( (\mathfrak{u} + 2\Delta)^2 + \Delta(\mathfrak{u} + \Delta) \right)$$
$$= 5 \left( (\mathfrak{u} + \Delta)^2 + 2(\mathfrak{u} + \Delta)\Delta + \Delta^2 + \Delta(\mathfrak{u} + \Delta) \right)$$
$$\le 5(5(\mathfrak{u} + \Delta)^2)$$
$$= (5(\mathfrak{u} + \Delta))^2$$
which implies that $\widetilde{\mathfrak{u}} \le 5(\mathfrak{u} + \Delta)$, completing the proof.

# E  Auxiliary results

**Lemma E.1** (Lipschitz continuity of Bregman divergences). *Assume* $\mathrm{diam}_{\|\cdot\|}(\mathcal{Y}) = \sup_{y,y'\in\mathcal{Y}} \|y - y'\| < \infty$ *and that* $\phi$ *is* $\nu$-*smooth w.r.t.* $\|\cdot\|$ *and* $\|\nabla\phi(x) - \nabla\phi(z)\|_* \le M \|x - z\|$. *Then* $D_\phi$ *is Lipschitz continuous in both of its arguments separately, that is, for all* $x, y, z$ *we have*

$$|D_\phi(y, x) - D_\phi(z, x)| \le \left(\frac{\nu}{2} + M\right) \mathrm{diam}_{\|\cdot\|}(\mathcal{Y}) \|y - z\|,$$

$$|D_\phi(y, x) - D_\phi(y, z)| \le \left(\frac{\nu}{2} + M\right) \mathrm{diam}_{\|\cdot\|}(\mathcal{Y}) \|x - z\|.$$

*Proof.* This follows from the three-point identity: First,

$$
\begin{aligned}
|D_\phi(y, x) - D_\phi(z, x)| &= |D_\phi(y, z) - \langle y - z, \nabla\phi(x) - \nabla\phi(z)\rangle| \\
&\le \frac{\nu}{2} \|y - z\|^2 + \|y - z\| \|\nabla\phi(x) - \nabla\phi(z)\|_* \\
&\le \frac{\nu}{2} \|y - z\|^2 + 2M \|y - z\| \|x - z\| \\
&\le \left(\frac{\nu}{2} + M\right) \mathrm{diam}_{\|\cdot\|}(\mathcal{Y}) \|y - z\|
\end{aligned}
$$

and second, by the same argument,

$$
\begin{aligned}
|D_\phi(y, x) - D_\phi(y, z)| &= |D_\phi(z, x) - \langle y - z, \nabla\phi(x) - \nabla\phi(z)\rangle| \\
&\le \frac{\nu}{2} \|z - x\|^2 + \|y - z\| \|\nabla\phi(x) - \nabla\phi(z)\|_* \\
&\le \frac{\nu}{2} \|z - x\|^2 + 2M \|y - z\| \|x - z\| \\
&\le \left(\frac{\nu}{2} + M\right) \mathrm{diam}_{\|\cdot\|}(\mathcal{Y}) \|x - z\|
\end{aligned}
$$

which concludes the proof. $\qquad\square$

## E.1  Concentration inequalities

**Lemma E.2** (Consequence of McDiarmid's inequality [43]). *Let* $\mathcal{F}$ *be a function class of measurable functions* $\mathcal{X} \to \mathbb{R}$ *that is* $B$-*bounded,* $\sup_{x\in\mathcal{X}} |f(x)| \le B$ *for all* $f \in \mathcal{F}$, *and* $X, X_1, \ldots, X_n$ *be i.i.d. random elements in* $\mathcal{X}$. *Define*

$$Z = \sup_{f\in\mathcal{F}} \left|\frac{1}{n} \sum_{i=1}^{n} f(X_i) - \mathbb{E}\left[f(X_1)\right]\right|.$$

*Then it holds that*

$$\mathbb{P}\left(|Z - \mathbb{E}\left[Z\right]| \le B\sqrt{\frac{2\log(1/\delta)}{n}}\right) \ge 1 - 2\delta.$$

The proof can be found, for instance, in [63, 56]. A significant improvement over Lemma E.2 is Talagrand's concentration inequality, stated next.

**Lemma E.3** (Talagrand's concentration inequality [60]). *Let* $\mathcal{F}$ *be a countable function class of measurable functions* $\mathcal{X} \to \mathbb{R}$ *that is* $B/2$-*bounded,* $\sup_{x\in\mathcal{X}} |f(x)| \le B/2$ *for all* $f \in \mathcal{F}$, *and* $X_1, \ldots, X_n$ *be i.i.d. random elements in* $\mathcal{X}$. *Define*

$$Z = \sup_{f\in\mathcal{F}} \left|\frac{1}{n} \sum_{i=1}^{n} f(X_i) - \mathbb{E}\left[f(X_1)\right]\right| \quad and \quad \sigma^2(\mathcal{F}) = \sup_{f\in\mathcal{F}} \mathbb{E}\left[(f(X_1) - \mathbb{E}\left[f(X_1)\right])^2\right] \le B^2.$$

*Then it holds that*

$$\mathbb{P}\left(Z \le 2\mathbb{E}\left[Z\right] + \sqrt{2}\sqrt{\frac{\sigma^2(\mathcal{F})\log(1/\delta)}{n}} + 3\frac{B\log(1/\delta)}{n}\right) \ge 1 - \delta.$$

### E.2 Rademacher complexities

While there are multiple notions of Rademacher complexity for vector-valued functions [50], our choice of Rademacher complexity in this work is motivated by the following contraction inequality, which is used multiple times in our proofs.

**Lemma E.4** (Vector contraction, Theorem 3 in [42] adapted for Rademacher complexities with absolute values). *Let $\mathcal{H}$ be a class of functions $\mathcal{X} \to \mathcal{Y} \subset \mathbb{R}^q$. Assume that $\ell : \mathcal{Y} \times \mathcal{Y} \to \mathbb{R}$ is $L$-Lipschitz continuous in its second argument with $\ell_2$-norm in $\mathbb{R}^q$, that is,*

$$\forall y, y', y'' \in \mathcal{Y}: \qquad |\ell(y, y') - \ell(y, y'')| \leq L \, \|y' - y''\|_2 \,.$$

*Then it holds for $\ell \circ \mathcal{H} := \{(x, y) \mapsto \ell(y, h(x)) : h \in \mathcal{H}\}$ that*

$$\mathfrak{R}_n(\ell \circ \mathcal{H}) \leq 2\sqrt{2}L\mathfrak{R}_n(\mathcal{H}) \leq 3L\mathfrak{R}_n(\mathcal{H}) \,,$$

*where $\mathfrak{R}_n$ denotes the coordinate-wise Rademacher complexity.*

This contraction inequality crucially relies on the $\ell_2$-Lipschitz continuity. If the loss exhibits more favourable Lipschitz continuity, e.g., with respect to an $\ell_p$-norm with $p > 2$, then our results can readily be adapted to use other contraction inequalities [24].

We now state two more well-known results from learning theory appearing throughout the manuscript, solely for convenience purposes.

**Lemma E.5** (Symmetrization in expectation, e.g., Theorem 4.10 in [63]). *Let $\mathcal{F}$ be a class of functions $\mathcal{X} \to \mathbb{R}$ and $n \in \mathbb{N}$. Let $X_1, \ldots, X_n$ be i.i.d. samples in $\mathcal{X}$. Then*

$$\mathbb{E}\left[\sup_{f \in \mathcal{F}} \left| \frac{1}{n} \sum_{i=1}^{n} f(X_i) - \mathbb{E}[f(X_1)] \right| \right] \leq 2\mathfrak{R}_n(\mathcal{F}) \,.$$

**Lemma E.6** (VC Bounds, [10, 9]). *Suppose that $\mathcal{H}$ consists of functions $\mathcal{X} \to \{0, 1\}$ and that $\mathcal{H}$ has VC dimension $d_{\mathcal{H}} \in \mathbb{N}$. Let $n \geq d_{\mathcal{H}}$. Then there exists a constant $C > 0$ so that the Rademacher complexity of $\mathcal{H}$ with respect to any distribution on $\mathcal{X}$ is bounded as*

$$\mathfrak{R}_n(\mathcal{H}) \leq \min\left\{ \sqrt{\frac{2d_{\mathcal{H}} \log(en/d_{\mathcal{H}})}{n}}, \, C\sqrt{\frac{d_{\mathcal{H}}}{n}} \right\} \,.$$

*If $\mathcal{H}$ consists of functions $\mathcal{X} \to [-B, B]$ and has VC-subgraph dimension (a.k.a. pseudo-dimension) $d_{\mathcal{H}} \in \mathbb{N}$, then there exists a constant $C > 0$ such that the Rademacher complexity of $\mathcal{H}$ with respect to any distribution on $\mathcal{X}$ is bounded as*

$$\mathfrak{R}_n(\mathcal{H}) \leq \min\left\{ 2B\sqrt{\frac{2d_{\mathcal{H}} \log(en/d_{\mathcal{H}})}{n}}, \, CB\sqrt{\frac{d_{\mathcal{H}}}{n}} \right\} \,.$$

*Moreover, for the $L^2$-norm ball of functions with the same distribution $\mu$ as the Rademacher complexity, $\mathcal{B} = \left\{ f \in L^2(\mu) : \|f\|_{L^2(\mu)} \leq 1 \right\}$, let $\rho := \inf \left\{ r > 0 : r^2 \geq \mathfrak{R}_n(\mathcal{H} \cap r\mathcal{B}) \right\}$. Then there exists a constant $C > 0$ so that*

$$\rho^2 \leq C \frac{d_{\mathcal{H}} \log(en/d_{\mathcal{H}})}{n} \,.$$

# F   Table of Notations



Table 2: Notation



| Symbol | Definition |
|---|---|
| $\ell_k, D_\phi$ | loss / Bregman divergence: $\mathcal{Y} \times \mathcal{Y} \to \mathbb{R}$ |
| $\mathcal{R}_k(f)$ | risk: $\mathbb{E}[\ell_k(Y^k, f(X^k))]$ |
| $\widehat{\mathcal{R}}_k(f)$ | empirical risk: $\frac{1}{n_k} \sum_{i=1}^{n_k} \ell_k(Y_i^k, f(X_i^k))$ |
| $\mathcal{E}_k(f)$ | excess risk: $\mathcal{R}_k(f) - \inf_{f \in \mathcal{F}_{\text{all}}} \mathcal{R}_k(f)$ |
| $\mathcal{T}_s(f)$ | $s$-trade-off: $s(\mathcal{E}_1(f), \ldots, \mathcal{E}_K(f))$ |
| — | excess $s$-trade-off: $\mathcal{T}_s(f) - \inf_{g \in \mathcal{G}} \mathcal{T}_s(g)$ |
| $d_k(f; h)$ | risk discrepancy: $\mathbb{E}[\ell_k(h(X^k), f(X^k))]$ |
| $\widehat{d}_k(f; h)$ | empirical risk discrepancy: $\frac{1}{N_k} \sum_{i=1}^{N_k} \ell_k(h(\widetilde{X}_i^k), f(\widetilde{X}_i^k))$ |
| $d_s(f; \boldsymbol{h})$ | scalarized discrepancy: $s(d_1(f; h_1), \ldots, d_K(f; h_K))$ |
| $\widehat{d}_s(f; \boldsymbol{h})$ | empirical scalarized discrepancy: $s(\widehat{d}_1(f; h_1), \ldots, \widehat{d}_K(f; h_K))$ |
| $\boldsymbol{f}^\star = (f_k^\star)_{k \in [K]}$ | Bayes-optimal models: $f_k^\star = \arg\min_{f \in \mathcal{F}_{\text{all}}} \mathcal{R}_k(f)$ |
| $\widehat{\boldsymbol{h}} = (\widehat{h}_k)_{k \in [K]}$ | ERMs in $\mathcal{H}_k$: $\widehat{h}_k = \arg\min_{h \in \mathcal{H}_k} \widehat{\mathcal{R}}_k(h)$ |
| $g_s$ | Pareto set in $\mathcal{G}$: $\arg\min_{g \in \mathcal{G}} d_s(g; \boldsymbol{f}^\star)$ |
| $g_s'$ | helper Pareto set in $\mathcal{G}$: $\arg\min_{g \in \mathcal{G}} d_s(g; \widehat{\boldsymbol{h}})$ |
| $\widehat{g}_s$ | our estimator: $\arg\min_{g \in \mathcal{G}} \widehat{d}_s(g; \widehat{\boldsymbol{h}})$ |
| $s_{\boldsymbol{\lambda}}^{\text{lin}}$ | linear scalarization: $\sum_{k=1}^{K} \lambda_k v_k$ |
| $s_{\boldsymbol{\lambda}}^{\text{max}}$ | Tchebycheff scalarization: $\max_{k \in [K]} \lambda_k v_k$ |
| $\mathcal{B}_1^d$ | $\ell_1$-ball: $\{\boldsymbol{v} \in \mathbb{R}^d : \|\boldsymbol{v}\|_1 \leq 1\}$ |
| $\mathcal{B}_2^d$ | $\ell_2$-ball: $\{\boldsymbol{v} \in \mathbb{R}^d : \|\boldsymbol{v}\|_2 \leq 1\}$ |
| $\mathcal{B}_\infty^d$ | $\ell_\infty$-ball: $\{\boldsymbol{v} \in \mathbb{R}^d : \|\boldsymbol{v}\|_\infty \leq 1\}$ |
| $\text{diam}_{\|\cdot\|}(\mathcal{A})$ | diameter of the set $\mathcal{A} \subset \mathbb{R}^d$: $\sup\{\|x - y\| : x, y \in \mathcal{A}\}$ |
| $\mathfrak{R}_n^k$ | Rademacher complexity w.r.t. distribution $k$ and $n$ samples |
| $\mathfrak{l}_k, \mathfrak{u}_k$ | critial radii from Eq. (12) |

