# OpenReview forum: "On the sample complexity of semi-supervised multi-objective learning"
_NeurIPS.cc/2025/Conference — NeurIPS 2025 spotlight_

### Official Review · Reviewer_v5Nn · 2025-06-23

**Clarity:** 2
**Significance:** 3
**Originality:** 3
**Rating:** 5
**Confidence:** 2

**Summary:**

This work investigates when the statistical cost of multi-objective learning can be mitigated, where the role of unlabeled data and loss functions is elucidated. Specifically, the authors first show that it is still necessary to collect enough labeled data for each task separately to control the complexity of the model class $\mathcal{G}$ under the general loss, even if the Bayes optimal model for each task is known and there is infinite unlabeled data. The authors then focus on the Bregman loss and propose a simple pseudo-labeling algorithm with uniform and local upper bounds on sample complexity.

**Questions:**

1. Deriving results relies on many strict assumptions. Do they usually hold in practical applications? It is up to the authors to validate them using real-world datasets.
2. Please add a discussion on the impact of the proposed pseudo-labeling algorithm on the possible generation of labeling noise.
3. Please add the effect on the Pareto front of changes in the proportion of unlabeled data.

**Ethical Concerns:**

["NO or VERY MINOR ethics concerns only"]

**Final Justification:**

I have read the rebuttals of authors and comments of other reviewers. I will keep my original score. Thanks.

**Limitations:**

The derivations in this paper rely on several strict assumptions. It is not clear whether all the assumptions hold in practical applications. In addition, the validation of the benefits of the proposed algorithm in this paper lacks larger-scale real data validation.

**Paper Formatting Concerns:**

Not applicable.

**Quality:**

3

**Strengths And Weaknesses:**

Strengths:
1. The theoretical contribution of this paper is solid. This paper establishes a sample complexity framework for semi-supervised MOL for the first time, and explicitly states the necessary conditions for “when unlabeled data is useful”. It analyzes the complexity of labeled samples for multi-objective learning in terms of both lower and upper bounds.
2. The proposed solution is simple, practical, and easy to integrate.

Weaknesses:
1. The empirical validation of this paper is insufficient and lacks real-world scenario examples, such as multi-risk trade-offs in medical diagnosis, autonomous driving, and other large-scale evaluations on practical problems.
2. No empirical comparison is made with existing deep learning-based MOL methods.

---

> ### Author Rebuttal · Authors · 2025-07-31
>
> Thank you for taking the time to review and for the positive feedback. Hopefully, the following will help answer your questions and provide additional context to address any lingering concerns you might have.
>
>
> **Summary of contributions**
>
> First, on a high level, we would like to clarify the goal of this paper: our results shed light on the fundamental differences between single-objective and multi-objective learning (MOL), rather than proposing an algorithm to be used for MOL in practice. The results and proofs should help us gain valuable intuition for leveraging “structure” in multi-objective learning and connect it explicitly to common intuition in single-objective learning - something that has not yet been clearly established in previous works. While we agree that it would be very interesting to empirically and theoretically explore more general settings - including deep learning and larger scale data and experiments - we argue that our results under the current assumptions (standard in classical statistical learning theory) are a valuable first step in building missing intuition on MOL.
>
> In particular, we are able to gain first insights on the following questions: how many (unlabeled or labeled) samples are needed when we aim to simultaneously minimize risks on multiple distributions, and how do unlabeled samples help? How do the sample complexities depend on the function classes that “solve” individual learning problems (in our paper $\mathcal{H}_k$) and the function class that is “necessary” for achieving good trade-offs (in our paper $\mathcal{G}$), and how do they depend on the loss functions?
> We’d be happy to explore in future work whether the insights may carry over in a similar way to neural network training and large-scale experiments.
>
> **Implications for real-world settings.**
>
> A significant limitation of existing theory is that it doesn’t really help practitioners decide which architecture/model class to use (beyond the usual consideration that more complex model classes may require more data). For single-objective learning, the choice of architecture in practice often incorporates strong priors hard-earned from experience (e.g., specific domain knowledge, or more general knowledge that certain architectures work well for certain types of data). But which model class should the practitioner choose in the multi-objective setting, especially if there are distinct priors for each task? This might be one of the first questions a practitioner faces, yet existing theory cannot answer this.
>
> The simple, but important, point that we make is that in MOL, the choice of model class no longer only serves to incorporate our prior knowledge about individual learning tasks, but it must also enable good trade-offs across those tasks. This dual role is elided in the earlier formulations of MOL, and so these questions about which model classes should be used cannot be answered there. Thus, our work reopens, or at least refines, the MOL problem in a way that is crucial for practice. It enables us to ask whether and how prior/domain knowledge about the individual tasks can be leveraged for learning good trade-offs across all tasks.
>
> **Testing assumptions and additional experiments.**
>
> Most of the assumptions made in our work are easily verified in practice: Assumptions 1, 2, 4, 5, and 6 are completely verifiable through the design choices of the practitioner. However, testing the distributional assumptions (including Assumption 3) under which the pseudolabeling algorithm enjoys its guarantees is hard: even the most basic assumptions, such as independence and identical distribution of the data, are essentially impossible to test on real-world data (or, such tests require assumptions themselves). This is a caveat most learning theory faces, and indeed has sparked the whole field of testable learning.
>
> At the same time, we would like to mention that we have also performed additional experiments on classification and non-parametric regression problems that confirm our findings. Unfortunately, we may not post any anonymized links or figures here.
>
> **Impact of label noise.**
>
> The problem of label noise is central to our work, and incorporating the error of having imperfect labels constitutes one of the main technical contributions of our work. The algorithm that we analyze consists of two stages: first, learning a separate model per task, and then using these models to pseudolabel the unlabeled data, on which the models that trade-off the different tasks are trained. Because the models from the first stage are imperfect, its labels indeed introduce a *label shift*. One of the main contributions of the proofs of Theorems 1 and 2 is to show how this label shift affects the models trained in the second stage, which requires non-standard tools. In our final bounds, the effect of this label shift is represented by the Rademacher complexity of $\mathcal{H}_k$: the larger the complexity, the larger the (expected) label shift, and the worse the bound. Consequently, to be able to learn with vanishing trade-off errors, the prediction error of the task-specific models, and hence also of the label shift, needs to vanish - otherwise there will remain an irreducible error even with infinite unlabeled data.
>
> **Effect on the Pareto front as a function of unlabeled data.**
>
>  In general, having more unlabeled data allows the model class to be larger, because it reduces the variance of the MOL problem. Hence, the Pareto front will move more towards the optimal solutions. And in the limit, we recover what is known as the *ideal semi-supervised* setting. The effect of unlabeled data over purely supervised methods can be observed empirically, for example, in Figure 3. Moreover, we plan to include a new experiment in the updated manuscript where we keep the number of labeled datapoints fixed and vary the number of unlabeled datapoints, and plot the varying Pareto fronts of the pseudolabeling algorithm. Again, unfortunately, we may not post any anonymized links or figures here.
>
>
> As we were somewhat unsure whether the reviewer’s question was aimed in this direction, we would kindly ask the reviewer to clarify their question if any unclear points remain.

---

### Official Review · Reviewer_n4Vo · 2025-07-02

**Clarity:** 3
**Significance:** 2
**Originality:** 3
**Rating:** 4
**Confidence:** 3

**Summary:**

The authors prove that unlabeled data is powerless to cut the Θ(K d_G / ε²) label cost of learning all Pareto trade-offs under generic losses, but show that for the broad Bregman-loss family a simple pseudo-labeling routine lets unlabeled examples absorb the dependence on the large joint class, so the remaining label complexity scales only with the per-task classes instead of $\mathcal{G}$.

**Questions:**

If we were to consider some more recent generalization measures, such as [1], can this study be poentially enhanced to fit the literture of deep models and lead to some more informative and pratical results?

[1] Theoretical analysis of self-training with deep networks on unlabeled data.

**Ethical Concerns:**

["NO or VERY MINOR ethics concerns only"]

**Final Justification:**

My opinion for this paper remains the same - a theoretical solid paper considering a relatively new problem, but with very little practicability and relevance to cutting-edge reserach (even for DNNs). The hypothesis class being considered in this paper remains to be simple and the derived results cannot will reconcile with modern deep learning (a known problem, not specified to this paper).

**Limitations:**

N.A.

**Paper Formatting Concerns:**

N.A.

**Quality:**

3

**Strengths And Weaknesses:**

**Strengths:**

1. The paper offers a rigorous proof that clarifies exactly how unlabeled data can reduce labeled-sample complexity relative to fully supervised learning.

2. The derived results looks interesting - in worst case, unlabeled data cannot help to learn the Pareto-optimal decisions for generic losses, but can be useful for Bregman-loss family (CE, MSE, etc.)

3. The high-level proof sketch distills the core argument clearly, making the full derivation much easier to follow.

**Weaknesses:**

1. The bounds rely on classical complexity measures such as VC dimension and Rademacher complexity; for modern models these quantities are often vacuous, limiting practical relevance.

2. The relationship of the results in this paper between previous works should be high-lighted and discussed, it is known that there are many discussions on the sample complexity of SSL and multi-task learning respectively, but how related are the results in this paper comparing to those results?

3. I am generally happy with Assumption 1-5, but Assumption 6 appears to be too strong - this basically enforces hypothesis class must be linear - akin to W1, this weakness also limits the scope of this study to much simpler model and does not fit the scope of recent literatures.

4. Because Bregman losses already cover the vast majority of practical objectives, the negative result for generic losses has limited real-world impact..

5. The motivation emphasizes “complex model classes” and cites state-of-the-art deep architectures in related works, yet the main theorems do not extend to those settings.

Overall, this seems to be a technical solid paper addressing a rather new problem, and could be of interest to the learning theory community, but the practical significance of the derived results remains modest in my view.

---

> ### Author Rebuttal · Authors · 2025-07-31
>
> We thank reviewer n4Vo for the thoughtful review. We appreciate that the reviewer thought that our work makes a solid theoretical contribution to a novel problem that is of interest to the learning theory community.
>
> **Summary of contributions**
>
> First, on a high level, we would like to clarify the goal of this paper: our results shed light on the fundamental differences between single-objective and multi-objective learning (MOL), rather than proposing an algorithm to be used for MOL in practice. The results and proofs should help us gain valuable intuition for leveraging “structure” in multi-objective learning and connect it explicitly to common intuition in single-objective learning - something that has not yet been clearly established in previous works. While we agree that it would be very interesting to empirically and theoretically explore more general settings - including deep learning and larger scale data and experiments - we argue that our results under the current assumptions (standard in classical statistical learning theory) are a valuable first step in building missing intuition on MOL.
>
> In particular, we are able to gain first insights on the following questions: how many (unlabeled or labeled) samples are needed when we aim to simultaneously minimize risks on multiple distributions, and how do unlabeled samples help? How do the sample complexities depend on the function classes that “solve” individual learning problems (in our paper $\mathcal{H}_k$) and the function class that is “necessary” for achieving good trade-offs (in our paper $\mathcal{G}$), and how do they depend on the loss functions?
> We’d be happy to explore in future work whether the insights may carry over in a similar way to neural network training and large-scale experiments.
>
>
> **Beyond Rademacher complexities**
>
> We acknowledge the reviewer’s main reservation: indeed, as our analysis is developed using tools from classical learning, it necessarily inherits their limitations. Namely, current learning theory significantly lags behind and does not satisfactorily explain the practical successes of modern, over-parametrized models.
>
> We’d like to emphasize again that we are very careful in the paper **not** to claim that our results show that the pseudo-learning algorithm is effective in practice (where the assumptions may indeed be too restrictive). In particular, we deliberately opted not to include any neural network experiments, since our theory is not aimed at informing deep learning settings (for which bounds with Rademacher complexities can indeed be insufficient).
>
> We agree that it would be of great practical significance to obtain generalization bounds that meaningfully apply to these modern ML models, and particularly for the multi-objective learning setting. Even more modestly, from just a theoretical perspective, we think that the suggested direction is definitely worth pursuing: to develop our results based on alternative complexity measures, such as the all-layer margin from [D] and the suggested work [E]. We suspect it would lead to quite different tools and bring additional insight about the nature of MOL. On the other hand, we feel that not only is this alternate approach substantial enough to merit its own paper, but that our work as is already makes a significant and fundamental contribution to MOL.
>
> **Practial implications of our work**
>
> In our view, one important contribution is the following simple, almost obvious, insight: the choice of model class needs to play a dual role of fitting individual tasks and enabling trade-offs across tasks. However, as discussed within our paper’s introduction, this distinction is not captured in existing theoretical frameworks. And so, under the frameworks of earlier works, it is not possible to ask *how can priors for individual tasks be incorporated into the multi-objective learning problem*. This is a practically-motivated question; after all, we often have strong priors about which models work well for specific learning tasks in practice.
>
> Another important contribution is our two-fold discovery that (a) having strong priors for the individual tasks does not automatically imply benefits for MOL, even with infinite unlabeled data, but that nonetheless, (b) this fundamental barrier can be overcome if we measure performance via more “informative” losses. From here, we established the first results showing how to incorporate priors about individual tasks into the learning problem (via simpler model classes $\mathcal{H}_k$), and when/how these priors can significantly reduce labeled sample complexity in MOL.
>
>
> **Relationship to SSL and MTL.**
>
> Thank you for the question about related work on semi-supervised learning (SSL) and multi-task learning (MTL). Some of this is presented with a fair amount of detail in Appendix A.1, which provides a discussion and comparison of both lines of research. Please let us know if you see any blind spots or gaps we may have in covering related literature. The overview of different sample complexities in our response to reviewer eRZf may also be helpful. Let us also summarize the main differences between the SSL and multi-task learning settings and our multi-objective setting here.
>
> Whether unlabeled data can help in the single-objective semi-supervised learning setting is quite nuanced, and usually requires some notions of “clusterability” or “compatibility”, see for example [9,15,45]. Roughly speaking, when the marginal does not carry any information about the conditional labels, unlabeled data cannot help. Our results, on the other hand, hold regardless of whether such clusterability or compatibility conditions hold: in MOL, unlabeled data enables estimating the likelihood of a given sample under each task, helping the model to make desired trade-offs corresponding to the task preferences/weights, even when the covariate does not inform the labels at all. If the likelihood of a sample is higher under one task than another, a model with a good trade-off prioritizes that task.
>
> As discussed in Appendix A.1, the goal of MTL is fundamentally different from the multi-objective learning setting (albeit the names being misleadingly similar). MTL aims to learn multiple models, one model per task, and in doing so, exploit relatedness between tasks to improve sample complexity. Specifically, at inference time in MTL, we are allowed to predict a different label per task, whereas in MOL, we have to commit to one label for all tasks. This is related to the collaborative learning setting.
>
> Because of these fundamental differences, the sample complexities of MTL and SSL are incomparable to our $\mathcal{S}$-MOL problem setting. Nevertheless, from a methodological point of view, our pseudolabeling algorithm is inspired by the standard SSL pseudolabeling approach, and linked to other methods such as mixture-of-experts, discussed in more detail in Appendix A.1.
>
> **Assumption 6.**
>
> We would like to clarify that the assumption that the function class is star-shaped/convex does not imply a linear function class. For example, the class of Lipschitz continuous functions on $[0,1]$ discussed in Appendix B.2 is convex (and hence star-shaped around every element), but Lipschitz continuous functions need not be linear. It is also standard to assume in learning theory to obtain localization results (e.g. Section 13.2 in [46]).
>
> **Negative result.**
>
> In some sense, we want the negative result given by Proposition 1 to have as little relevance as possible in the real world: that would mean that we can hope to practically achieve MOL. The role of this lower bound (and probably of most lower bounds in general) is to illuminate where the fundamental barriers are, in this case, to multi-objective learning. In particular, Proposition 1 reveals the importance of the choice of loss function in MOL. If the loss is “uninformative” like the zero-one loss, then it can be impossible to leverage strong priors for the individual tasks for the multi-objective goal of making good trade-offs.
> This motivated our consideration of more “structured” losses, where we discovered that Bregman losses sidestep this barrier. Happily, and as the reviewer notes, this is a general class of losses, broadly used in practice. Note that this is only one way that additional “structure” in MOL allows us to leverage the benefits of unlabeled data. Such lower bounds help us know where to look for other types of structure for future research directions.
>
> We hope these discussions clarify any remaining concerns and encourage the reviewer to ask any further clarifying questions if necessary.
>
>
> [D] Wei, C., & Ma, T. (2019). Improved sample complexities for deep networks and robust classification via an all-layer margin.
>
> [E] Wei, C., Shen, K., Chen, Y., & Ma, T. (2020). Theoretical analysis of self-training with deep networks on unlabeled data.

---

> ### Comment · Reviewer_n4Vo · 2025-08-04
>
> Thank the authors for their comprehensive rebuttal, my previous concerns are mostly satisfied. Nevertheless, as authors have acknowledged - currently the theoretical contributions from this paper might have limited impacts to cutting-edge research - that being said, this is perhaps a common drawback.
>
> I will maintain my evaluation as positive and not consider further increase the score.

---

### Official Review · Reviewer_McKt · 2025-07-03

**Clarity:** 3
**Significance:** 3
**Originality:** 3
**Rating:** 4
**Confidence:** 2

**Summary:**

This paper investigates the sample complexity of semi-supervised multi-objective learning (MOL), where the goal is to simultaneously solve for multiple learning objectives, or "tasks," using a single model. The work addresses a key challenge: while each individual task might be simple to solve within a small model class ($\mathcal{H}_k$), finding a model that achieves a good trade-off among all tasks might require a much more complex model class ($\mathcal{G}$).

For "uninformative" losses like the zero-one loss, the paper establishes a sample complexity lower bound showing that the amount of labeled data required scales with the complexity of the large trade-off class $\mathcal{G}$. This limitation holds even if the learner has access to infinite unlabeled data and the optimal models for each individual task.

For the important class of Bregman losses (which includes squared error and cross-entropy), the authors show that unlabeled data can significantly alleviate the need for labeled data. They propose a simple pseudo-labeling algorithm (PL-MOL) and establish sample complexity upper bounds for it. The key result is that for this algorithm, the labeled sample complexity depends only on the complexity of the simple, individual-task classes ($\mathcal{H}_k$), while the complexity of the larger trade-off class ($\mathcal{G}$) is absorbed by the unlabeled sample complexity.

**Questions:**

1. The main theoretical result (Theorem 1) cleanly separates the sample complexity, with labeled data complexity depending on the simpler classes $\mathcal{H}_k$ and unlabeled data complexity depending on the more complex class $\mathcal{G}$. Is this separation fundamental to semi-supervised MOL, or is it a feature of the specific pseudo-labeling approach?

**Ethical Concerns:**

["NO or VERY MINOR ethics concerns only"]

**Limitations:**

yes

**Quality:**

3

**Strengths And Weaknesses:**

Strengths:

1. The paper is the first to formally study the sample complexity of multi-objective learning (MOL) in a semi-supervised setting. It addresses the practical question of when and how unlabeled data can reduce the need for labeled data when finding optimal trade-offs requires a more complex model class than solving individual tasks.
2. The paper provides a balanced perspective by establishing both a negative and a positive result. It first proves a strong hardness result, showing that for general losses like the zero-one loss, even infinite unlabeled data doesn't help reduce the labeled sample complexity. It then shows that for the important class of Bregman losses (e.g., squared error, cross-entropy), this barrier can be overcome.

---

> ### Author Rebuttal · Authors · 2025-07-31
>
> We thank reviewer McKt for the positive and helpful review. Let us address the question - for a clarification of our contributions, we refer to the *summary of contributions* section in the rebuttal of reviewer eRZf.
>
> **Summary of contributions**
>
> First, on a high level, we would like to clarify the goal of this paper: our results shed light on the fundamental differences between single-objective and multi-objective learning (MOL), rather than proposing an algorithm to be used for MOL in practice. The results and proofs should help us gain valuable intuition for leveraging “structure” in multi-objective learning and connect it explicitly to common intuition in single-objective learning - something that has not yet been clearly established in previous works. While we agree that it would be very interesting to empirically and theoretically explore more general settings - including deep learning and larger scale data and experiments - we argue that our results under the current assumptions (standard in classical statistical learning theory) are a valuable first step in building missing intuition on MOL.
>
> In particular, we are able to gain first insights on the following questions: how many (unlabeled or labeled) samples are needed when we aim to simultaneously minimize risks on multiple distributions, and how do unlabeled samples help? How do the sample complexities depend on the function classes that “solve” individual learning problems (in our paper $\mathcal{H}_k$) and the function class that is “necessary” for achieving good trade-offs (in our paper $\mathcal{G}$), and how do they depend on the loss functions?
> We’d be happy to explore in future work whether the insights may carry over in a similar way to neural network training and large-scale experiments.
>
>
> **Separation of labeled and unlabeled data.**
>
> The question of whether the separation of unlabeled and labeled data in Theorems 1 and 2 is fundamental to the problem or part of the PL-MOL algorithm is interesting and a good opportunity to reiterate the main point of our work:
>
> That the label complexity depends only on $\mathcal{H}_k$ and the unlabeled complexity only on $\mathcal{G}$ seems to be fundamental for the setting/assumptions we consider. Because, to recover the entire Pareto front, we need to be able to solve each individual learning task, the labeled sample complexity on $\mathcal{H}_k$ is *unremovable* (as we’ve assumed $\mathcal{H}_k$ is contained in $\mathcal{G}$). Then, the remaining question is whether we can reduce the dependence of unlabeled sample complexity on $\mathcal{G}$. In general, this is also not possible. For example, consider the case that we are given the Bayes optimal predictors for each task; then no labeled data is necessary, but there is still a need for unlabeled data to learn how to make trade-offs over $\mathcal{G}$. This necessity for unlabeled data was also exemplified in [47] (see Proposition 5).
>
> Also note that one viable approach may be to reuse the labeled data by discarding the labels and pooling it with the unlabeled data. This may help in practice, but it does not change the order of the unlabeled sample complexity.

---

### Official Review · Reviewer_eRZf · 2025-07-03

**Clarity:** 3
**Significance:** 3
**Originality:** 3
**Rating:** 5
**Confidence:** 4

**Summary:**

This paper studies semi-supervised multi-objective learning (S-MOL), where a learner must solve multiple prediction tasks (risks $R_k$) simultaneously using both labeled and unlabeled data. To address this problem, the authors investigate the sample complexity of learning theory pareto optimal models. To my knowledge, the authors derive the first sample-complexity results of this kind.  They prove a minimax lower bound (Proposition 1) showing that for general losses the labeled-sample complexity scales with the complexity of the full model class $G$ – even if the Bayes predictors for each task and infinite unlabeled data (equivalently, the marginal distribution of feature data of each task)  are available .  By contrast, they show that for Bregman losses (e.g. squared loss, cross-entropy) one can exploit unlabeled data via a simple pseudo-labeling algorithm.  Specifically, they obtain uniform-convergence upper bounds (Theorem 1) and stronger localized Rademacher-complexity bounds (Theorem 2) for this algorithm.  These bounds demonstrate that the labeled sample complexity depends only on each task’s class $H_k$ (where $H_k$ is a simpler model for which the individual prediction task $k$ can be done with negligible error), while the complexity of $G$ enters only through the unlabeled data requirements. Further the effect of complexity of G vanishes when we have infinite unlabeled data.  In effect, this shows when and how unlabeled data can significantly reduce the label complexity of learning multiple Pareto-optimal trade-offs.

**Questions:**

1. The results focus on Bregman losses and linear scalarizations. Could the authors discuss whether the analysis might extend to other loss families (e.g., hinge) or scalarizations (e.g.,
2.  A short paragraph or table comparing sample complexity to supervised MOL, multi‑distribution learning, and classical SSL would help highlight what is truly new.
3. As mentioned in weaknesses, Assumption 2 requires each loss to be a bounded, Lipschitz, strongly-convex Bregman loss. This is not the case even for KL loss, unless we consider strong convexity with respect to generalized norm such as $\ell_1$ norm for KL. Is it possible to extend the analytical argument to support the strong convexity in the generalized form?

**Ethical Concerns:**

["NO or VERY MINOR ethics concerns only"]

**Final Justification:**

My assessment is the same as before. This paper is a solid theoretical work. I will keep my positive score.

**Limitations:**

Yes

**Paper Formatting Concerns:**

Nothing!

**Quality:**

4

**Strengths And Weaknesses:**

# Strengths
* **Strong theoretical contributions.**  The hardness result (Proposition 1) is an interesting result: it shows that in general S-MOL has no advantage over supervised MOL in the worst case.  Even when the Bayes models $f^*_k$ and the marginal distributions are known, any algorithm needs $\Omega(K,d_G/\epsilon^2)$ labeled examples to achieve $\epsilon$-optimality for all trade-offs. In particular, the proof shows that even in the case of two tasks, the sample complexity can be large, even when the optimal models for individual tasks is known.   This lower bound is non-trivial (proved via reduction to multiple Hypothesis testing problem and applying Assouad’s lemma) and highlights the fundamental difficulty of S-MOL without further structure.
* **Effective pseudo-labeling algorithm and upper bounds.**  The paper’s key positive result is that for Bregman losses, a simple pseudo-labeling approach yields much lower label complexity.  By first learning each task’s Bayes predictor $h_k$ in a simpler model $H_k$ (which is good enough for learning task $k$) from labels, and then using these to pseudo-label unlabeled samples, the algorithm reduces the multi-task problem to solving scalarized objectives on unlabeled data .  Theorem 1 gives a uniform learning bound via standard Rademacher analysis.  For VC-type classes  $\tilde O(Kd_{H}/\epsilon^4)$ labels and $\tilde O(Kd_G/\epsilon^2)$ unlabeled samples are needed.  Importantly, the result shows large complexity of $G$ can be absorbed by unlabeled data, so the label cost is driven by the (smaller) $H_k$’s.
* **Advanced analytical techniques.**  Beyond the uniform bound, the authors derive localized fast-rate bounds (Theorem 2) using finer Rademacher-complexity tools .  By carefully analyzing the interaction between the estimated regressors and the true Pareto-optimal models (using a stability argument from [47]), they avoid the “square-root” penalty in the labeled terms.  Theorem 2 shows that if the critical radii of $H_k$ are small, the convergence can be much faster (e.g. improving from $n^{-1/4}$ to $n^{-1/2}$ in rate).  The technical argument is well-explained (with a helpful figure) and leverages known results [6,22] for multi-task generalization.  These theoretical tools are non-trivial and represent a solid extension of localized Rademacher analysis to the multi-objective SSL context.
# Weaknesses
*  **Strong assumptions.**  The positive results rely on several restrictive assumptions.  Assumption 1 requires each task class $H_k$ to contain the true Bayes predictor and be well-specified.  Assumption 2 requires each loss to be a bounded, Lipschitz, strongly-convex Bregman loss. This is not the case even for KL loss, unless we consider strong convexity with respect to generalized norm such as $\ell_1$ norm for KL. Further, for theorem 2 we have 4 more assumptions. These conditions are mathematically convenient but limit scope.  In practice, tasks may be mis-specified or losses may not decompose in this way.  The paper acknowledges that without such structure, Proposition 1 shows no benefit to SSL.  Thus the usefulness of the method depends on how often real problems satisfy these conditions.
* **Clarity of exposition (minor).**  The paper is dense and notation-heavy.  Although definitions are given, a non-expert reader may struggle.  For example, the metric assumptions for Theorem 2 (Assumptions 3–6) are quite technical and deferred to the appendix, which could be better summarized in prose.  Also, the connection between the pseudo-label algorithm and classical co-training or multi-task methods is not fully explored, which might help intuition.  These are not serious flaws but areas where the writing could be smoothed or given more intuition (beyond the formal statements).

---

> ### Author Rebuttal · Authors · 2025-07-31
>
> We thank reviewer eRZf for the detailed, positive review and the valuable feedback. Thank you also for the suggestions on how to improve clarity; we will continue to work on improving our exposition. We’d like to address the questions and clarify our contributions.
>
> **Summary of our contributions.**
>
> First, on a high level, we would like to clarify the goal of this paper: our results shed light on the fundamental differences between single-objective and multi-objective learning (MOL), rather than proposing an algorithm to be used for MOL in practice. The results and proofs should help us gain valuable intuition for leveraging “structure” in multi-objective learning and connect it explicitly to common intuition in single-objective learning - something that has not yet been clearly established in previous works. While we agree that it would be very interesting to empirically and theoretically explore more general settings - including deep learning and larger scale data and experiments - we argue that our results under the current assumptions (standard in classical statistical learning theory) are a valuable first step in building missing intuition on MOL.
>
> In particular, we are able to gain first insights on the following questions: how many (unlabeled or labeled) samples are needed when we aim to simultaneously minimize risks on multiple distributions, and how do unlabeled samples help? How do the sample complexities depend on the function classes that “solve” individual learning problems (in our paper $\mathcal{H}_k$) and the function class that is “necessary” for achieving good trade-offs (in our paper $\mathcal{G}$), and how do they depend on the loss functions?
> We’d be happy to explore in future work whether the insights may carry over in a similar way to neural network training and large-scale experiments.
>
> **On our assumptions.**
>
> Let us clarify our assumptions and how they might be extended. In general, we tried to formulate our results as general as possible, and our assumptions each play an important role.
>
> - **Assumption 1:** The focus of our paper is the juxtaposition of easy individual tasks and potentially hard to learn trade-offs. We argue that Assumption 1 is a natural way to model this setting, and can also be viewed the other way around: given the classes $\mathcal{H}_1,\dots,\mathcal{H}_K$, how large can we choose $\mathcal{G}$? Moreover, the assumption can easily be extended to a misspecified setting using irreducible approximation errors.
>
> - **Assumption 2, strong convexity and KL divergence:** The assumption that the potentials are strongly convex immediately holds for “generalized” strong convexity (w.r.t. other norms besides $\ell_2$) whenever they are norm-equivalent to $\ell_2$ (and from functional analysis, all norms are indeed equivalent in finite-dimensional space). For example, the potential associated with the KL divergence $\phi(y) = \sum y_i \log y_i$ is not only strongly convex with respect to the $\ell_1$-norm, but also the $\ell_2$-norm on the simplex (this follows, for instance, from Pinsker’s inequality [B]). Perhaps the more subtle issue is that for other potentials, the norm equivalence constant may depend on the dimension of the label space, and so an interesting question is whether it is possible to prove bounds that are independent of the label space dimension for strongly convex potentials under such norms.
>
> - **Assumptions 4-6:** These assumptions are standard and (essentially) necessary for localization, see, for instance, Section 14.3 in [46]. If they do not hold, usually, other assumptions need to replace them to achieve fast rates, e.g. [21].
>
> We’ll add a discussion on which losses satisfy these assumptions to clarify the full extent of our results.
>
> **Scalarizations.**
>
> We’d like to stress that, while our second main result, Theorem 2 (fast rates under stronger assumptions), is restricted to linear scalarization, our first main result, *Theorem 1 (slow rate but more general), applies to broad classes of scalarizations*, importantly including the Tchebycheff scalarization. It would be extremely interesting, but quite non-trivial, to extend Theorem 2 to general, non-linear scalarizations.
>
> **Beyond Bregman losses.**
>
> While Bregman losses encompass a large family of important losses, we agree that an important and very interesting next direction is to study non-Bregman losses; especially losses, such as the hinge loss, that are commonly used in practice. And indeed, we have discovered settings where there are potential benefits to semi-supervision for non-Bregman losses, but so far they’ve required substantially stronger assumptions and more specialized algorithms. In fact, in some sense, Algorithm 1 works *if and only if* the losses are Bregman, due to Theorem 12 in [C].
>
> **Overview of sample complexities and existing works.**
>
> For readability, let us summarize the *label* complexities from prior work and derived in our work for VC (subgraph)-classes with dimensions $d_{\mathcal{G}},d_{\mathcal{H}}$ of $\mathcal{G}$ and $\mathcal{H}_k=\mathcal{H}$. But note that our results are much more general.
>
> In the supervised MDL setting, the sample complexity is of order $(d_{\mathcal{G}}+K)/\epsilon^2$ under adaptive sampling (see related works). For non-adaptive sampling, however, it is $Kd_{\mathcal{G}}/\epsilon^2$, see [52]. In the supervised $\mathcal{S}$-MOL setting the sample complexity also is $Kd_{\mathcal{G}}/\epsilon^2$ (see discussion before Prop. 1 and App. A.2).
> The single-objective semi-supervised learning setting is more nuanced because the degree to which unlabeled data can improve upon just using the labeled data usually requires some notions of “clusterability” or “compatibility” (e.g., margins, or disagreement in co-training), see [9,15,45] and [A] for an overview. But in the worst case, its label complexity is $d_{\mathcal{G}}/\epsilon^2$ (i.e., unlabeled data cannot help). Similarly, the sample complexity of multi-task learning is lower than that of learning each task separately only under additional assumptions.
>
> Our work establishes that in the semi-supervised $\mathcal{S}$-MOL setting with zero-one loss, the label complexity remains $Kd_{\mathcal{G}}/\epsilon^2$ in the worst case, whereas for Bregman losses it is $Kd_{\mathcal{H}}/\epsilon^4$ or $Kd_{\mathcal{H}}/\epsilon$ depending on the strength of the assumptions.
> We will include a table of sample complexities in the updated paper.
>
> [A] Balcan, M. F., & Blum, A. (2006). An augmented PAC model for semi-supervised learning.
> [B] Yu, Y. L. (2013). The strong convexity of von Neumann’s entropy. Unpublished note, 15.
> [C] T. Heskes. Bias-variance decompositions: the exclusive privilege of Bregman divergences.

---

> > ### Comment · Reviewer_eRZf · 2025-08-05
> >
> > Thanks for the detailed responses, which have addressed my previous concerns. I have one final question and a minor comment.
> >
> > ---
> >
> > ### Question
> >
> > Regarding the strong convexity of **Kullback–Leibler (KL) divergence**:
> > The authors are absolutely right in pointing out that KL divergence is  also strongly convex with respect to the \ell_2-norm. As the authors mentioned, however, the constant depends on the dimension, d. While the constant for the L_1-L_2 equivalence is 1 in the worst case (i.e., $\|.\|_1 \ge \|.\|_2$), for a large d, we have $\frac{\| .\|_1}{\|.\|_2} = \Theta(\sqrt{d})$ with a high probability. In this scenario, one might anticipate a better bound than the one based on Pinsker's inequality.
> > Thus I'm interested in understanding how the proposed theoretical framework depends on using the **Euclidean norm**. Could the authors discuss the challenges and possibilities for generalizing the approach to other norms (e.g., $L_p$ norms)? To what extent does the reliance on the Euclidean norm present a fundamental limitation or "bottleneck" for the broader applicability of this work?
> >
> > ---
> >
> > ### Minor Comment
> > It appears that the Pareto fronts of different function classes contradict **Assumption 1**, which states that $\mathcal{G}$ contains all $\mathcal{H}_k$. Specifically, the front of $\mathcal{G}$ should be below the front of each $\mathcal{H}_k$.
> >
> > ---
> >
> > Overall, I appreciate the concrete theoretical work presented in this paper and will maintain my positive score.

---

> > > ### Author Response · Authors · 2025-08-05
> > >
> > > Thank you for the interesting clarifying question and minor comment.
> > >
> > > **Answer to question on  strong convexity**
> > >
> > > First, just to make sure this is what the reviewer meant, we’d like to clarify that the strong convexity constant with respect to $\ell_2$-norm of the entropy potential, which induces the KL divergence,  is _dimension-independent_, since from Pinsker's inequality we get
> > > $$
> > > \operatorname{KL}(x,y) := \phi(x) - \phi(y)-\nabla\phi(y)^\top (x-y) \geq \frac{1}{2}\lVert x-y \rVert_1^2 \geq \frac{1}{2}\lVert x-y \rVert_2^2.
> > > $$
> > > We also emphasize again that the dimension here refers to the label space.
> > >
> > > Now, if we understand the reviewer correctly, the question is this: not using the _stronger_ inequality from above could be _wasteful_, as the $\ell_1$ norm can be much larger than the $\ell_2$ norm.
> > >
> > > _And indeed, it is possible,_ and actually straightforward, to extend our Theorem 1 to make use of this effect: Whenever the potential is strongly convex with respect to a norm $\lVert\cdot\rVert$, we can replace the assumption of _Lipschitz continuity in the first argument_ of the loss to hold in the same norm $\lVert\cdot\rVert$, which is then a _weaker_ assumption and the overall bound may benefit from a favourable dependence on this Lipschitz constant (which may in turn depend on the label space dimension).
> > > In fact, the only place where the specific norm (of both the potential and the Lipschitz continuity in the first argument) matter is in the proof of the auxiliary Lemma C.2; nothing else changes in the proof. We intend to implement this in our updated manuscript.
> > >
> > > For Theorem 2 the question is more nuanced, because the proof heavily relies on the Hilbert space structure of the $\ell_2$-norm. Consequently, it is not straightfoward to adapt the proof in the same way one can adapt Theorem 1. We suspect that it is not possible (up to the trivial case of Mahalanobis norms) without a substantially different proof technique, but we will certainly investigate this further.
> > >
> > > **Comment on Figure 1a**
> > >
> > > Thank you for pointing this out - indeed, the Pareto front of $\mathcal{G}$ in Figure 1a should have the endpoints at the same points that the $\mathcal{H}_k$ have theirs. We have also noticed this minor error and adapted the figure.

---

### Decision · Program_Chairs · 2025-09-17

**Decision:**

Accept (spotlight)

**Comment:**

This paper appears to be the first to formally study the sample complexity of multi-objective learning (MOL) in a semi-supervised setting. All the reviewers agree that the paper addresses a practical and interesting question, and provides balanced results on theory and simulation. During the rebuttal period, the authors and the reviewers were actively engaged, which led to a favorable decision of this paper.